# On the Saturation Effects of Spectral Algorithms in Large Dimensions

**Weihao Lu**
Department of Statistics and Data Science
Tsinghua University
Beijing, China 100084
luwh19@mails.tsinghua.edu.cn

**Haobo Zhang**
Department of Statistics and Data Science
Tsinghua University
Beijing, China 100084
zhang-hb21@mails.tsinghua.edu.cn

**Yicheng Li**
Department of Statistics and Data Science
Tsinghua University
Beijing, China 100084
liyc22@mails.tsinghua.edu.cn

**Qian Lin**[*]
Department of Statistics and Data Science
Tsinghua University
Beijing, China 100084
qianlin@tsinghua.edu.cn

## Abstract

The saturation effects, which originally refer to the fact that kernel ridge regression (KRR) fails to achieve the information-theoretical lower bound when the regression function is over-smooth, have been observed for almost 20 years and were rigorously proved recently for kernel ridge regression and some other spectral algorithms over a fixed dimensional domain. The main focus of this paper is to explore the saturation effects for a large class of spectral algorithms (including the KRR, gradient descent, etc.) in large dimensional settings where $n \asymp d^\gamma$. More precisely, we first propose an improved minimax lower bound for the kernel regression problem in large dimensional settings and show that the gradient flow with early stopping strategy will result in an estimator achieving this lower bound (up to a logarithmic factor). Similar to the results in KRR, we can further determine the exact convergence rates (both upper and lower bounds) of a large class of (optimal tuned) spectral algorithms with different qualification $\tau$'s. In particular, we find that these exact rate curves (varying along $\gamma$) exhibit the periodic plateau behavior and the polynomial approximation barrier. Consequently, we can fully depict the saturation effects of the spectral algorithms and reveal a new phenomenon in large dimensional settings (i.e., the saturation effect occurs in large dimensional setting as long as the source condition $s > \tau$ while it occurs in fixed dimensional setting as long as $s > 2\tau$).

## 1   Introduction

Let's assume we have $n$ i.i.d. samples $(x_i, y_i)$ from a joint distribution supported on $\mathbb{R}^d \times \mathbb{R}$. The regression problem, one of the most fundamental problems in statistics, aims to find a function $\hat{f}$ based on these samples such that the *excess risk*, $\|\hat{f} - f_\star\|_{L^2}^2 = \mathbb{E}_x[(f_\star(x) - \hat{f}(x))^2]$, is small, where $f_\star(x) = \mathbb{E}[Y|x]$ is the *regression function*. Many non-parametric regression methods are proposed to solve the regression problem by assuming that $f_\star$ falls into certain function classes, including polynomial splines Stone (1994), local polynomials Cleveland (1979); Stone (1977), the spectral algorithms Caponnetto (2006); Caponnetto and De Vito (2007); Caponnetto and Yao (2010), etc.

---

[*]Corresponding author.

38th Conference on Neural Information Processing Systems (NeurIPS 2024).

Spectral algorithms, as a classical topic, have been studied since the 1990s. Early works treated certain types of spectral algorithms in their theoretical analysis (Caponnetto (2006); Caponnetto and De Vito (2007); Raskutti et al. (2014); Lin et al. (2020)). These works often consider $d$ as a fixed constant and impose the polynomial eigenvalue decay assumption under a kernel (i.e., there exist constants $0 < \mathfrak{c} \leq \mathfrak{C} < \infty$, such that the eigenvalues of the kernel satisfy $\mathfrak{c}j^{-\beta} \leq \lambda_j \leq \mathfrak{C}j^{-\beta}, j \geq 1$ for certain $\beta > 1$ depending on the fixed $d$). They further assume that $f_\star$ belongs to the reproducing kernel Hilbert space (RKHS) $\mathcal{H}$ associated with the kernel. Under the above assumptions, they then showed that the minimax rate of the excess risk of regression over the corresponding RKHS is lower bounded by $n^{-\beta/(\beta+1)}$ and that some (regularized) spectral algorithms, e.g., the kernel ridge regression (KRR) and the kernel gradient flow, can produce estimators achieving this minimax optimal rate.

However, subsequent studies have revealed that when higher regularity (or smoothness) of $f_\star$ is assumed, KRR fails to achieve the information-theoretical lower bound on the excess risk, while kernel gradient flow can do so. Specifically, let's assume that $f_\star$ belongs to the *interpolation space* $[\mathcal{H}]^s$ of the RKHS $\mathcal{H}$ with $s > 0$ (see, e.g., Steinwart et al. (2009); Dieuleveut et al. (2017); Dicker et al. (2017); Pillaud-Vivien et al. (2018); Lin et al. (2020); Fischer and Steinwart (2020); Celisse and Wahl (2021)). It is then shown that the information-theoretical lower bound on the excess risk is $n^{-s\beta/(s\beta+1)}$. When $0 < s \leq 2$, Caponnetto and De Vito (2007); Yao et al. (2007); Lin et al. (2020); Zhang et al. (2023) have already shown that the upper bound of the excess risks of both KRR and the kernel gradient flow is $n^{-s\beta/(s\beta+1)}$, and hence they are minimax optimal. On the contrary, when $s > 2$, Yao et al. (2007); Lin et al. (2020) showed that the upper bound of the excess risks of kernel gradient flow is $n^{-s\beta/(s\beta+1)}$ while the best upper bound of the excess risks of KRR is $n^{-2\beta/(2\beta+1)}$ (Caponnetto and De Vito (2007)). Bauer et al. (2007); Gerfo et al. (2008); Dicker et al. (2017) conjectured that the convergence rate of KRR is bounded below by $n^{-2\beta/(2\beta+1)}$ and Li et al. (2022) rigorously proved it. The above phenomenon is often referred to as the *saturation effect* of KRR:

*KRR is inferior to certain spectral algorithms, such as kernel gradient flow, when $s > 2$.*

In recent years, neural network methods have gained tremendous success in many large-dimensional problems, such as computer vision He et al. (2016); Krizhevsky et al. (2017) and natural language processing Devlin (2018). Several groups of researchers tried to explain the superior performance of neural networks on large-dimensional data from the aspects of "lazy regime" (Arora et al. (2019); Du et al. (2019, 2018); Li and Liang (2018)). They noticed that, when the width of a neural network is sufficiently large, its parameters/weights stay in a small neighborhood of their initial position during the training process. Later, Jacot et al. (2018); Arora et al. (2019); Hu et al. (2021); Suh et al. (2021); Lai et al. (2023); Li et al. (2024) proved that the time-varying neural network kernel (NNK) converges (uniformly) to a time-invariant neural tangent kernel (NTK) as the width of the neural network goes to infinity, and thus the excess risk of kernel gradient flow with NTK converges (uniformly) to the excess risk of neural networks in the 'lazy regime'.

Inspired by the concepts of the "lazy regime" and the uniform convergence of excess risk, the machine learning community has experienced a renewed surge of interest in large-dimensional spectral algorithms. The earliest works focused on the consistency of two specific types of spectral algorithms: KRR and kernel interpolation (Liang and Rakhlin (2020); Liang et al. (2020); Ghorbani et al. (2020, 2021); Mei et al. (2021, 2022); Misiakiewicz and Mei (2022); Aerni et al. (2023); Barzilai and Shamir (2023)). In comparison, results on large-dimensional kernel gradient flow were somewhat scarce, and these results largely mirrored those associated with KRR (e.g., Ghosh et al. (2021)). Recently, Lu et al. (2023) proved that large-dimensional kernel gradient flow is minimax optimal when $s = 1$. Then, Zhang et al. (2024) provided upper and lower bounds on the convergence rate on the excess risk of KRR for any $s > 0$. Surprisingly, they discovered that for $s > 1$, the convergence rate of KRR did not match the lower bound on the minimax rate. Unfortunately, they didn't prove that certain spectral algorithms can reach the lower bound on the minimax rate they provided, and hence they didn't rigorously prove that the saturation effect of KRR occurs in large dimensions. Instead, Zhang et al. (2024) only conjectured that certain spectral algorithms (e.g., kernel gradient flow) can provide minimax optimal estimators after their main results.

If Zhang et al. (2024)'s conjecture is true, then we can safely conclude that: when the regression function $f_\star$ is smooth enough, KRR is inferior to kernel gradient flow in large dimensions as well. Consequently, previous results on large-dimensional KRR may not be directly extendable to large-

dimensional neural networks, even if the neural networks are in the 'lazy regime'. The main focus of this paper is to prove this conjecture by showing that kernel gradient flow is minimax optimal in large dimensions.

## 1.1 Related work

**Saturation effects of fixed-dimensional spectral algorithms.** When the dimension $d$ of the data is fixed, the saturation effect of KRR has been conjectured for decades and is rigorously proved in the recent work Li et al. (2022). Suppose $f_\star \in [\mathcal{H}]^s$ with $s > 2$. It is shown that: (i) the minimax optimal rate is $n^{-s\beta/(s\beta+1)}$ (Rastogi and Sampath (2017); Yao et al. (2007); Lin et al. (2020)); and (ii) the convergence rate on the excess risk of KRR is $n^{-2\beta/(2\beta+1)}$ (Li et al. (2022)). More recently, Li et al. (2024) determined the exact generalization error curves of a class of analytic spectral algorithms, which allowed them to further show the saturation effect of spectral algorithms with finite qualification $\tau$ (see, e.g., Appendix C): suppose $f_\star \in [\mathcal{H}]^s$ with $s > 2\tau$, then the convergence rate on the excess risk of the above spectral algorithms is $n^{-2\tau\beta/(2\tau\beta+1)}$.

**New phenomena in large-dimensional spectral algorithms.** In the large-dimensional setting where $n \asymp d^\gamma$ with $\gamma > 0$, new phenomena exhibited in spectral algorithms are popular topics in recent machine-learning research. A line of work focused on the polynomial approximation barrier phenomenon (e.g., Ghorbani et al. (2021); Donhauser et al. (2021); Mei et al. (2022); Xiao et al. (2023); Misiakiewicz (2022); Hu and Lu (2022)). They found that, for the square-integrable regression function, KRR and kernel gradient flow are consistent if and only if the regression function is a polynomial with a low degree. Another line of work considered the benign overfitting of kernel interpolation (i.e., kernel interpolation can generalize) (e.g., Liang and Rakhlin (2020); Liang et al. (2020); Aerni et al. (2023); Barzilai and Shamir (2023); Zhang et al. (2024)). Moreover, two recent work (Lu et al. (2023); Zhang et al. (2024)) discussed two new phenomena exhibited in large-dimensional KRR and kernel gradient flow: the multiple descent behavior and the periodic plateau behavior. The multiple descent behavior refers to the phenomenon that the curve of the convergence rate ( with respect to $n$ ) of the optimal excess risk is non-monotone and has several isolated peaks and valleys; while the periodic plateau behavior refers to the phenomenon that the curve of the convergence rate ( with respect to $d$ ) of the optimal excess risk has constant values when $\gamma$ is within certain intervals. Finally, Zhang et al. (2024) conjectured that the saturation effect of KRR occurs in large dimensions. The above works imply that these phenomena occur in many spectral algorithms in large dimensions, hence encouraging us to provide a unified explanation of these new phenomena.

## 1.2 Our contributions

In this paper, we focus on the large-dimensional spectral algorithms with inner product kernels, and we assume that the regression function falls into an interpolation space $[\mathcal{H}]^s$ with $s > 0$. We state our main results as follows:

**Theorem 1.1** (Restate Theorem 4.1 and 4.2, non-rigorous)**.** *Let $s > 0$, $\tau \geq 1$, and $\gamma > 0$ be fixed real numbers. Denote $p$ as the integer satisfying $\gamma \in [p(s+1), (p+1)(s+1))$. Then under certain conditions, the excess risk of large-dimensional spectral algorithm with qualification $\tau$ satisfies*

$$\mathbb{E}\left( \left\| \hat{f}_{\lambda^\star} - f_\star \right\|_{L^2}^2 \Big| X \right) = \begin{cases} \Theta_{\mathbb{P}}\left( d^{-\min\{\gamma - p, s(p+1)\}} \right) \cdot poly\left( \ln(d) \right), & s \leq \tau \\ \Theta_{\mathbb{P}}\left( d^{-\min\left\{\gamma - p, \frac{\tau(\gamma - p + 1) + p\tilde{s}}{\tau + 1}, \tilde{s}(p+1)\right\}} \right) \cdot poly\left( \ln(d) \right), & s > \tau, \end{cases}$$

*where $\tilde{s} = \min\{s, 2\tau\}$.*

More specifically, we list the main contributions of this paper as follows:

(1) In Theorem 3.1, we show that the convergence rate on the excess risk of (optimally-tuned) kernel gradient flow in large dimensions is $\Theta_{\mathbb{P}}(d^{-\min\{\gamma - p, s(p+1)\}}) \cdot \mathrm{poly}(\ln(d))$, which matches the lower bound on the minimax rate given in Theorem 3.3 (up to a logarithmic factor). We find that kernel gradient flow is minimax optimal for any $s > 0$ and any $\gamma > 0$, and KRR is not minimax optimal for $s > 1$ and for certain ranges of $\gamma$ (We provide a visual illustration in Figure 2). Consequently, we rigorously prove that the saturation effect of KRR occurs in large dimensions.

(2) In Theorem 3.3, we enhanced the previous minimax lower bound results given in Lu et al. (2023) and Zhang et al. (2024). Specifically, we show that the minimax lower bound is $\Omega(d^{-\min\{\gamma-p,s(p+1)\}})/\mathrm{poly}(\ln(d))$. In comparison, the previous minimax lower bound is $\Omega(d^{-\min\{\gamma-p,s(p+1)\}})/d^{\varepsilon}$ for any $\varepsilon > 0$, and the additional term $d^{\varepsilon}$ changes the desired convergence rate.

(3) In Section 4, we determine the convergence rate on the excess risk of large-dimensional spectral algorithms. From our results, we find several new phenomena exhibited in spectral algorithms in large-dimensional settings. We provide a visual illustration of the above phenomena in Figure 1: i) The first phenomenon is the polynomial approximation barrier, and as shown in Figure 1(a), when $s$ is close to zero, the curve of the convergence rate of spectral algorithm drops when $\gamma \approx p$ for any integer $p$ and will stay invariant for most of the other $\gamma$; ii) The second one is the periodic plateau behavior, and as shown in Figure 1(b) and Figure 1(c), when $0 < s < 2\tau$ and $\gamma \in [p(s+1)+s+(\max\{s,\tau\}-\tau)/\tau, (p+1)(s+1))$ for an integer $p \geq 0$, the convergence rate does not change when $\gamma$ varies; iii) The final one is the saturation effect, and as shown in Figure 1(c) and Figure 1(d), when $s > \tau$, the convergence rate of spectral algorithm can not achieve the minimax lower bound for certain ranges of $\gamma$. A detailed discussion about the above three phenomena can be found in Section 4.

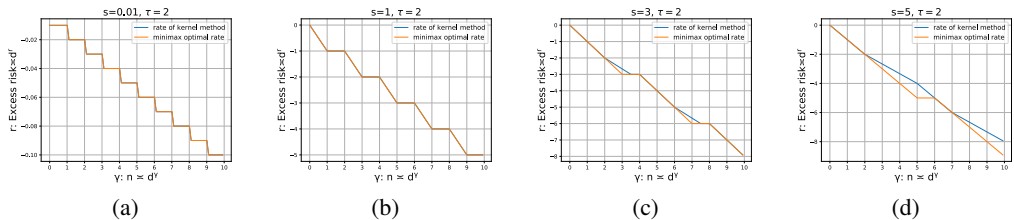

Figure 1: Convergence rates of spectral algorithm with qualification $\tau = 2$ in Theorem 4.1, Theorem 4.2, and corresponding minimax lower rates in Theorem 3.3 with respect to dimension $d$. We present four graphs corresponding to four kinds of source conditions: $s = 0.01, 1, 3, 5$. The x-axis represents asymptotic scaling, $\gamma : n \asymp d^{\gamma}$; the y-axis represents the convergence rate of excess risk, $r :$ Excess risk $\asymp d^{r}$.

## 2 Preliminaries

Suppose that we have observed $n$ i.i.d. samples $(x_i, y_i), i \in [n]$ from the model:

$$y = f_\star(x) + \epsilon, \tag{1}$$

where $x_i$'s are sampled from $\rho_{\mathcal{X}}$, $\rho_{\mathcal{X}}$ is the marginal distribution on $\mathcal{X} \subset \mathbb{R}^{d+1}$, $y \in \mathcal{Y} \subset \mathbb{R}$, $f_\star$ is some function defined on a compact set $\mathcal{X}$, and

$$\mathbb{E}_{(x,y)\sim\rho}\left[\epsilon^2 \mid x\right] \leq \sigma^2, \quad \rho_{\mathcal{X}}\text{-a.e. } x \in \mathcal{X},$$

for some fixed constant $\sigma > 0$, where $\rho$ is the joint distribution of $(x, y)$ on $\mathcal{X} \times \mathcal{Y}$. Denote the $n \times 1$ data vector of $y_i$'s and the $n \times d$ data matrix of $x_i$'s by $Y$ and $X$ respectively.

### 2.1 Kernel ridge regression and kernel gradient flow

In this subsection, we introduce two specific spectral algorithms, kernel ridge regression and kernel gradient flow, which produce estimators of the regression function $f_\star$. A further discussion on general spectral algorithms will be provided in Section 4.

Throughout the paper, we denote $\mathcal{H}$ as a separable RKHS on $\mathcal{X}$ with respect to a continuous and positive definite kernel function $K(\cdot, \cdot) : \mathcal{X} \times \mathcal{X} \to \mathbb{R}$ and there exists a constant $\kappa$ satisfying

$$\max_{x \in \mathcal{X}} K(x, x) \leq \kappa^2.$$

**Kernel ridge regression**    Kernel ridge regression (KRR) constructs an estimator $\hat{f}_\lambda^{\text{KRR}}$ by solving the penalized least square problem

$$\hat{f}_\lambda^{\text{KRR}} = \arg\min_{f \in \mathcal{H}} \left( \frac{1}{n} \sum_{i=1}^{n} (y_i - f(x_i))^2 + \lambda \|f\|_{\mathcal{H}}^2 \right),$$

where $\lambda > 0$ is referred to as the regularization parameter. The representer theorem (see, e.g., Steinwart and Christmann (2008)) gives an explicit expression of the KRR estimator, i.e.,

$$\hat{f}_\lambda^{\text{KRR}}(x) = K(x, X)(K(X, X) + n\lambda \mathbf{I})^{-1} Y. \tag{2}$$

**Kernel gradient flow**    The gradient flow of the loss function $\mathcal{L} = \frac{1}{2n} \sum_i (y_i - f(x_i))^2$ induced a gradient flow in $\mathcal{H}$ which is given by

$$\frac{\mathrm{d}}{\mathrm{d}t} \hat{f}_t^{\text{GF}}(x) = -\frac{1}{n} K(x, X)(\hat{f}_t^{\text{GF}}(X) - Y). \tag{3}$$

If we further assume that $\hat{f}_0^{\text{GF}}(x) = 0$, then we can also give an explicit expression of the kernel gradient flow estimator

$$\hat{f}_t^{\text{GF}}(x) = K(x, X)K(X, X)^{-1}(\mathbf{I} - e^{-\frac{1}{n}K(X,X)t})Y. \tag{4}$$

### 2.2    The interpolation space

Define the integral operator $T_K$ as $T_K(f)(x) = \int K(x, x')f(x')\,\mathrm{d}\rho_{\mathcal{X}}(x')$. It is well known that $T_K$ is a positive, self-adjoint, trace-class, and hence a compact operator (Steinwart and Scovel (2012)). The celebrated Mercer's theorem further assures that

$$K(x, x') = \sum_j \lambda_j \phi_j(x)\phi_j(x'), \tag{5}$$

where the eigenvalues $\{\lambda_j, j = 1, 2, ...\}$ is a non-increasing sequence, and the corresponding eigenfunctions $\{\phi_j(\cdot), j = 1, 2, ...\}$ are orthonormal in $L^2(\mathcal{X}, \rho_{\mathcal{X}})$ function space.

The interpolation space $[\mathcal{H}]^s$ with source condition $s$ is defined as

$$[\mathcal{H}]^s := \left\{ \sum_j a_j \lambda_j^{s/2} \phi_j : (a_j)_j \in \ell_2 \right\} \subseteq L^2(\mathcal{X}, \rho_{\mathcal{X}}), \tag{6}$$

with the inner product deduced from

$$\Big\| \sum_{j=1}^{\infty} a_j \lambda_j^{s/2} \phi_j \Big\|_{[\mathcal{H}]^s} = \Big( \sum_{j=1}^{\infty} a_j^2 \Big)^{1/2}. \tag{7}$$

It is easy to show that $[\mathcal{H}]^s$ is also a separable Hilbert space with orthonormal basis $\{\lambda_j^{s/2}\phi_j\}_j$. Generally speaking, functions in $[\mathcal{H}]^s$ become smoother as $s$ increases (see, e.g., the example of Sobolev RKHS in Edmunds and Triebel (1996); Zhang et al. (2023).

### 2.3    Assumptions

In this subsection, we list the assumptions that we need for our main results.

To avoid potential confusion, we specify the following large-dimensional scenario for kernel regression where we perform our analysis: suppose that there exist three positive constants $c_1$, $c_2$ and $\gamma$, such that

$$c_1 d^\gamma \leq n \leq c_2 d^\gamma, \tag{8}$$

and we often assume that $d$ is sufficiently large.

In this paper, we only consider the inner product kernels defined on the sphere. An inner product kernel is a kernel function $K$ defined on $\mathbb{S}^d$ such that there exists a function $\Phi : [-1, 1] \to \mathbb{R}$ independent of $d$ satisfying that for any $x, x' \in \mathbb{S}^d$, we have $K(x, x') = \Phi(\langle x, x' \rangle)$. If we further

assume that the marginal distribution $\rho_{\mathcal{X}}$ is the uniform distribution on $\mathcal{X} = \mathbb{S}^d$, then the Mercer's decomposition for $K$ can be rewritten as

$$K(x, x') = \sum_{k=0}^{\infty} \mu_k \sum_{j=1}^{N(d,k)} Y_{k,j}(x) Y_{k,j}(x'), \tag{9}$$

where $Y_{k,j}$ for $j = 1, \cdots, N(d, k)$ are spherical harmonic polynomials of degree $k$ and $\mu_k$'s are the eigenvalues of $K$ with multiplicity $N(d, 0) = 1$; $N(d, k) = \frac{2k+d-1}{k} \cdot \frac{(k+d-2)!}{(d-1)!(k-1)!}, k = 1, 2, \cdots$. For more details of the inner product kernels, readers can refer to Gallier (2009).

*Remark* 2.1. We consider the inner product kernels on the sphere mainly because the harmonic analysis is clear on the sphere ( e.g., properties of spherical harmonic polynomials are more concise than the orthogonal series on general domains). This makes Mercer's decomposition of the inner product more explicit rather than several abstract assumptions ( e.g., Mei and Montanari (2022)). We also notice that very few results are available for Mercer's decomposition of a kernel defined on the general domain, especially when the dimension of the domain is taking into consideration. e.g., even the eigen-decay rate of the neural tangent kernels is only determined for the spheres. Restricted by this technical reason, most works analyzing the spectral algorithm in large-dimensional settings focus on the inner product kernels on spheres (Liang et al., 2020; Ghorbani et al., 2021; Misiakiewicz, 2022; Xiao et al., 2023; Lu et al., 2023, etc.). Though there might be several works that tried to relax the spherical assumption (e.g., Liang et al. (2020); Aerni et al. (2023); Barzilai and Shamir (2023), we can find that most of them (i) adopted a near-spherical assumption; (ii) adopted strong assumptions on the regression function, e.g., $f_\star(x) = x[1]x[2] \cdots x[L]$ for an integer $L > 0$, where $x[i]$ denotes the $i$-th component of $x$; or (iii) can not determine the convergence rate on the excess risk of the spectral algorithm.

To avoid unnecessary notation, let us make the following assumption on the inner product kernel $K$.

*Assumption* 1. $\Phi(t) \in \mathcal{C}^{\infty}([-1, 1])$ is a fixed function independent of $d$ and there exists a non-negative sequence of absolute constants $\{a_j \geq 0\}_{j \geq 0}$, such that we have

$$\Phi(t) = \sum_{j=0}^{\infty} a_j t^j,$$

where $a_j > 0$ for any $j \leq \lfloor \gamma \rfloor + 3$.

The purpose of Assumption 1 is to keep the main results and proofs clean. Notice that, by Theorem 1.b in Gneiting (2013), the inner product kernel $K$ on the sphere is semi-positive definite for all dimensions if and only if all coefficients $\{a_j, j = 0, 1, 2, ...\}$ are non-negative. One can easily extend our results in this paper when certain coefficients $a_k$'s are zero (e.g., one can consider the two-layer NTK defined as in Section 5 of Lu et al. (2023), with $a_i = 0$ for any $i = 3, 5, 7, \cdots$).

In the next assumption, we formally introduce the source condition, which characterizes the relative smoothness of $f_\star$ with respect to $\mathcal{H}$.

*Assumption* 2 (Source condition). Suppose that $f_\star(x) = \sum_{i=1}^{\infty} f_i \phi_i(x)$.

(a) $f_\star \in [\mathcal{H}]^s$ for some $s > 0$, and there exists a constant $R_\gamma$ only depending on $\gamma$, such that

$$\|f_\star\|_{[\mathcal{H}]^s} \leq R_\gamma. \tag{10}$$

(b) Denote $q$ as the smallest integer such that $q > \gamma$ and $\mu_q \neq 0$. Define $\mathcal{I}_{d,k}$ as the index set satisfying $\lambda_i \equiv \mu_k, i \in \mathcal{I}_{d,k}$. Further suppose that there exists an absolute constant $c_0 > 0$ such that for any $d$ and $k \in \{0, 1, \cdots, q\}$ with $\mu_k \neq 0$, we have

$$\sum_{i \in \mathcal{I}_{d,k}} \mu_k^{-s} f_i^2 \geq c_0. \tag{11}$$

Assumption 2 is a common assumption when one is interested in the tight bounds on the excess risk of spectral algorithms (e.g., Caponnetto and De Vito (2007); Fischer and Steinwart (2020), Eq.(8) in Cui et al. (2021), Assumption 3 in Li et al. (2024), and Assumption 5 in Zhang et al. (2024)). Assumption 2 implies that the regression function exactly falls into the interpolation space $[\mathcal{H}]^s$, that is, $f_\star \in [\mathcal{H}]^s$ and $f_\star \notin [\mathcal{H}]^t$ for any $t > s$. For example, from the proof part I of Lemma D.14, one can check that $f_\star$ with $\sum_{i \in \mathcal{I}_{d,p}} \mu_p^{-s} f_i^2 = \sum_{i \in \mathcal{I}_{d,p+1}} \mu_{p+1}^{-s} f_i^2 = 0$ can have a faster convergence rate on the excess risk.

*Notations.* Let's denote the norm in $L_2(\mathcal{X}, \rho_\mathcal{X})$ as $\|\cdot\|_{L_2}$. For a vector $x$, we use $x[i]$ to denote its $i$-th component. We use asymptotic notations $O(\cdot)$, $o(\cdot)$, $\Omega(\cdot)$ and $\Theta(\cdot)$. For instance, we say two (deterministic) quantities $U(d), V(d)$ satisfy $U(d) = o(V(d))$ if and only if for any $\varepsilon > 0$, there exists a constant $D_\varepsilon$ that only depends on $\varepsilon$ and the absolute positive constants $\sigma, \kappa, s, \gamma, c_0, c_1, c_2, \mathfrak{C}_1, \cdots, \mathfrak{C}_8 > 0$, such that for any $d > D_\varepsilon$, we have $U(d) < \varepsilon V(d)$. We also write $a_n = \text{poly}(b_n)$ if there exist a constant $\theta \geq 0$, such that $a_n = \Theta(b_n^\theta)$. We use the probability versions of the asymptotic notations such as $O_\mathbb{P}(\cdot), o_\mathbb{P}(\cdot), \Omega_\mathbb{P}(\cdot), \Theta_\mathbb{P}(\cdot)$. For instance, we say the random variables $X_n, Y_n$ satisfying $X_n = O_\mathbb{P}(Y_n)$ if and only if for any $\varepsilon > 0$, there exist constants $C_\varepsilon$ and $N_\varepsilon$ such that $P(|X_n| \geq C_\varepsilon |Y_n|) \leq \varepsilon, \forall n > N_\varepsilon$.

## 2.4 Review of the previous results

The following two results are restatements of Theorem 2 and Theorem 5 in Zhang et al. (2024).

**Proposition 2.2.** *Let $s \geq 1$ and $\gamma > 0$ be fixed real numbers. Denote $p$ as the integer satisfying $\gamma \in [p(s+1), (p+1)(s+1))$. Suppose that Assumption 1 and Assumption 2 hold for $s$ and $\gamma$. Let $\hat{f}_\lambda^{\text{KRR}}$ be the function defined in (2). Define $\tilde{s} = \min\{s, 2\}$, then there exists $\lambda^\star > 0$, such that we have*

$$\mathbb{E}\left(\left\|\hat{f}_{\lambda^\star}^{\text{KRR}} - f_\star\right\|_{L^2}^2 \,\Big|\, X\right) = \Theta_\mathbb{P}\left(d^{-\min\left\{\gamma - p, \frac{\gamma - p + p\tilde{s} + 1}{2}, \tilde{s}(p+1)\right\}}\right) \cdot poly\left(\ln(d)\right),$$

*where $\Theta_\mathbb{P}$ only involves constants depending on $s, \sigma, \gamma, c_0, \kappa, c_1$ and $c_2$. In addition, the convergence rates of the generalization error can not be faster than above for any choice of regularization parameter $\lambda = \lambda(d, n) \to 0$.*

**Proposition 2.3** (Lower bound on the minimax rate). *Let $s > 0$ and $\gamma > 0$ be fixed real numbers. Denote $p$ as the integer satisfying $\gamma \in [p(s+1), (p+1)(s+1))$. Let $\mathcal{P}$ consist of all the distributions $\rho$ on $\mathcal{X} \times \mathcal{Y}$ such that Assumption 1 and Assumption 2 hold for $s$ and $\gamma$. Then for any $\varepsilon > 0$, we have:*

$$\min_{\hat{f}} \max_{\rho \in \mathcal{P}} \mathbb{E}_{(X,Y) \sim \rho^{\otimes n}} \left\|\hat{f} - f_\star\right\|_{L^2}^2 = \Omega\left(d^{-\min\{\gamma - p, s(p+1)\}} \cdot d^{-\varepsilon}\right),$$

*where $\Omega$ only involves constants depending on $s, \sigma, \gamma, c_0, \kappa, c_1, c_2$ and $\varepsilon$.*

From the above two propositions, we can find that when $s > 1$, the convergence rate on the excess risk of KRR does not always match the lower bound on the minimax optimal rate. Zhang et al. (2024) further conjectured that the lower bound on the minimax optimal rate provided in Proposition 2.3 is tight (ignoring the additional term $d^{-\varepsilon}$). Hence, they believed that the saturation effect exists for large-dimensional KRR.

# 3 Main results

In this section, we determine the convergence rate on the excess risk of kernel gradient flow as $d^{-\min\{\gamma - p, s(p+1)\}}\text{poly}\left(\ln(d)\right)$, which differs from the lower bound on the minimax rate provided in Proposition 2.3 by $d^\varepsilon$ for any $\varepsilon > 0$. We then tighten the lower bound on the minimax rate to $d^{-\min\{\gamma - p, s(p+1)\}}/\text{poly}\left(\ln(d)\right)$. Based on the above results, we find that KRR is not minimax optimal for $s > 1$ and for certain ranges of $\gamma$. Therefore, we show that the saturation effect of KRR occurs in large dimensions.

## 3.1 Exact convergence rate on the excess risk of kernel gradient flow

We first state our main results in this paper.

**Theorem 3.1** (Kernel gradient flow). *Let $s > 0$ and $\gamma > 0$ be fixed real numbers. Denote $p$ as the integer satisfying $\gamma \in [p(s+1), (p+1)(s+1))$. Suppose that Assumption 1 and Assumption 2 hold for $s$ and $\gamma$. Let $\hat{f}_t^{\text{GF}}$ be the function defined in (4). Then there exists $t^\star > 0$, such that we have*

$$\mathbb{E}\left(\left\|\hat{f}_{t^\star}^{\text{GF}} - f_\star\right\|_{L^2}^2 \,\Big|\, X\right) = \Theta_\mathbb{P}\left(d^{-\min\{\gamma - p, s(p+1)\}}\right) \cdot poly\left(\ln(d)\right), \tag{12}$$

*where $\Theta_\mathbb{P}$ only involves constants depending on $s, \sigma, \gamma, c_0, \kappa, c_1$ and $c_2$.*

Theorem 3.1 is a direct corollary of Theorem 4.1 and Example 2. Combining with the previous results in Proposition 2.3, or our modified minimax rate given in Theorem 3.3, we can conclude that large-dimensional kernel gradient flow is minimax optimal for any $s > 0$ and any $\gamma > 0$. More importantly, the convergence rate of kernel gradient flow is faster than that of KRR given in Proposition 2.2 when (i) $1 < s \leq 2$ and $\gamma \in (p(s+1) + 1, p(s+1) + 2s - 1)$ for some $p \in \mathbb{N}$, or (ii) $s > 2$ and $\gamma \in (p(s+1) + 1, (p+1)(s+1))$ for some $p \in \mathbb{N}$. Therefore, we have proved the saturation effect of KRR in large dimensions.

*Remark* 3.2. When $p \geq 1$, the logarithm term $\mathrm{poly}(\ln(d))$ in (12) can be removed. When $p = 0$, we have $\mathrm{poly}(\ln(d)) = (\ln(d))^2$ in (12). See Appendix D.4 for details.

## 3.2 Improved minimax lower bound

Recall that Proposition 2.3 gave a lower bound on the minimax rate as $d^{-\min\{\gamma - p, s(p+1)\}} \cdot d^{-\varepsilon}$. The following theorem replaces the additional term $d^{-\varepsilon}$ (which has changed the convergence rate) into a logarithm term $\mathrm{poly}^{-1}(\ln(d))$ (which does not change the desired convergence rate).

**Theorem 3.3** (Improved minimax lower bound). *Let $s > 0$ and $\gamma > 0$ be fixed real numbers. Denote $p$ as the integer satisfying $\gamma \in [p(s+1), (p+1)(s+1))$. Let $\mathcal{P}$ consist of all the distributions $\rho$ on $\mathcal{X} \times \mathcal{Y}$ such that Assumption 1 and Assumption 2 hold for $s$ and $\gamma$. Then we have:*

$$\min_{\hat{f}} \max_{\rho \in \mathcal{P}} \mathbb{E}_{(X,Y) \sim \rho^{\otimes n}} \left\| \hat{f} - f_\star \right\|_{L^2}^2 = \Omega \left( d^{-\min\{\gamma - p, s(p+1)\}} \right) \Big/ poly(\ln(d)), \qquad (13)$$

*where $\Omega$ only involves constants depending on $s, \sigma, \gamma, c_0, \kappa, c_1$, and $c_2$.*

# 4 Exact convergence rate on the excess risk of spectral algorithms

In this section, we will give tight bounds on the excess risks of certain types of spectral algorithms, such as kernel ridge regression, iterated ridge regression, kernel gradient flow, and kernel gradient descent.

Given an analytic filter function $\varphi_\lambda(\cdot)$ with qualification $\tau \geq 1$ (refer to Appendix C for the definitions of analytic filter function and its qualification), we can define a spectral algorithm in the following way (see, e.g., Bauer et al. (2007)). For any $y \in \mathbb{R}$, let $K_x : \mathbb{R} \to \mathcal{H}$ be given by $K_x(y) = y \cdot K(x, \cdot)$, whose adjoint $K_x^* : \mathcal{H} \to \mathbb{R}$ is given by $K_x^*(f) = \langle K(x, \cdot), f \rangle_{\mathcal{H}} = f(x)$. Moreover, we denote by $T_x = K_x K_x^*$ and $T_X = \frac{1}{n} \sum_{i=1}^n T_{x_i}$. We also define the sample basis function

$$\hat{g}_Z = \frac{1}{n} \sum_{i=1}^n K_{x_i}(y_i) = \frac{1}{n} \sum_{i=1}^n y_i \cdot K(x_i, \cdot). \qquad (14)$$

Now, the estimator of the spectral algorithm is defined by

$$\hat{f}_\lambda = \varphi_\lambda(T_X) \hat{g}_Z. \qquad (15)$$

Many commonly used spectral algorithms can be constructed by certain analytic filter functions. We provide two examples (kernel ridge regression and kernel gradient flow) as follows, and put two more examples (iterated ridge regression and kernel gradient descent) in Appendix C. We provide rigorous proof for these examples in Lemma C.3.

**Example 1** (Kernel ridge regression). *The filter function of kernel ridge regression (KRR) is well-known to be*

$$\varphi_\lambda^{\mathrm{KRR}}(z) = \frac{1}{z + \lambda}, \quad \psi_\lambda^{\mathrm{KRR}}(z) = \frac{\lambda}{z + \lambda}, \quad \tau = 1. \qquad (16)$$

**Example 2** (Kernel gradient flow). *The filter function is*

$$\varphi_\lambda^{\mathrm{GF}}(z) = \frac{1 - e^{-tz}}{z}, \quad \psi_\lambda^{\mathrm{GF}}(z) = e^{-tz}, \quad t = \lambda^{-1}, \quad \tau = \infty. \qquad (17)$$

For any analytic filter function $\varphi_\lambda$ with qualification $\tau \geq 1$ and the corresponding estimator of the spectral algorithm defined in (15), the following two theorems provide exact convergence rates on the excess risk when (i) the regression function is less-smooth, i.e., we have $s \leq \tau$, and (ii) $s > \tau$, where $s$ is the source condition coefficient of the regression function given in Assumption 2.

**Theorem 4.1.** *Let $0 < s \le \tau$ and $\gamma > 0$ be fixed real numbers. Denote $p$ as the integer satisfying $\gamma \in [p(s+1), (p+1)(s+1))$. Suppose that Assumption 1 and Assumption 2 hold for $s$ and $\gamma$. Let $\varphi_\lambda(z)$ be an analytic filter function and $\hat{f}_\lambda$ be the function defined in (15). Suppose one of the following conditions holds:*

$$(i)\ \tau = \infty, \quad (ii)\ s > 1/(2\tau), \quad (iii)\ \gamma > ((2\tau+1)s)/(2\tau(1+s));$$

*then there exists $\lambda^\star > 0$, such that we have*

$$\mathbb{E}\left( \left\| \hat{f}_{\lambda^\star} - f_\star \right\|_{L^2}^2 \ \Big|\ X \right) = \Theta_{\mathbb{P}}\left( d^{-\min\{\gamma-p, s(p+1)\}} \right) \cdot poly\left( \ln(d) \right),$$

*where $\Theta_{\mathbb{P}}$ only involves constants depending on $s, \sigma, \gamma, c_0, \kappa, c_1$ and $c_2$.*

**Theorem 4.2.** *Let $s > \tau$ and $\gamma > 0$ be fixed real numbers. Denote $p$ as the integer satisfying $\gamma \in [p(s+1), (p+1)(s+1))$. Suppose that Assumption 1 and Assumption 2 hold for $s$ and $\gamma$. Let $\varphi_\lambda(z)$ be an analytic filter function and $\hat{f}_\lambda$ be the function defined in (15). Define $\tilde{s} = \min\{s, 2\tau\}$, then there exists $\lambda^\star > 0$, such that we have*

$$\mathbb{E}\left( \left\| \hat{f}_{\lambda^\star} - f_\star \right\|_{L^2}^2 \ \Big|\ X \right) = \Theta_{\mathbb{P}}\left( d^{-\min\left\{\gamma-p, \frac{\tau(\gamma-p+1)+p\tilde{s}}{\tau+1}, \tilde{s}(p+1)\right\}} \right) \cdot poly\left( \ln(d) \right),$$

*where $\Theta_{\mathbb{P}}$ only involves constants depending on $s, \sigma, \gamma, c_0, \kappa, c_1$ and $c_2$. In addition, the convergence rates of the generalization error can not be faster than above for any choice of regularization parameter $\lambda = \lambda(d, n) \to 0$.*

*Remark* 4.3. These theorems substantially generalize the results on exact generalization error bounds of analytic spectral algorithms under the fixed-dimensional setting given in Li et al. (2024). Although the "analytic functional argument" introduced in their proof is still vital for us to deal with the general spectral algorithms, their proof has to rely on the polynomial eigendecay assumption that $\lambda_j \asymp j^{-\beta}$ (Assumption 1), which does not hold in large dimensions since the hidden constant factors in the assumption vary with $d11$ (Lu et al. (2023)). Hence, their proof is not easy to generalize to large-dimensional spectral algorithms.

We provide some graphical illustrations of Theorem 4.1 and Theorem 4.2 in Figure 1 (with $\tau = 2$) and in Appendix A (with $\tau = 1$, $\tau = 2$, $\tau = 4$, and $\tau = \infty$, corresponding to KRR, iterated ridge regression in Example 3 and kernel gradient flow).

As a direct consequence of Theorem 3.3, Theorem 4.1, and Theorem 4.2, we find that for the spectral algorithm with estimator defined in (15), it is minimax optimal if $s \le \tau$ and the conditions in Theorem 4.1 hold. Moreover, these results show several phenomena for large-dimensional spectral algorithms.

**Saturation effect of large-dimensional spectral algorithms with finite qualification.** In the large-dimensional setting and for the inner product kernel on the sphere, our results show that the saturation effect of spectral algorithms occurs when $s > \tau$. As shown in Figure 1(c) and Figure 1(d), when $s > \tau$, no matter how carefully one tunes the regularization parameter $\lambda$, the convergence rate can not be faster than $d^{-\min\{\gamma-p, \frac{\tau(\gamma-p+1)+p\tilde{s}}{\tau+1}, \tilde{s}(p+1)\}}$, thus can not achieve the minimax lower bound $d^{-\min\{\gamma-p, s(p+1)\}}$.

**Periodic plateau behavior of spectral algorithms when $s \le 2\tau$.** When $0 < s \le 2\tau$ and $\gamma \in [p(s+1) + s + \max\{s, \tau\}/\tau - 1, (p+1)(s+1))$ for an integer $p \ge 0$, from Theorem 4.1 and Theorem 4.2, the convergence rate on the excess risk of spectral algorithm $d^{-s(p+1)}$. The above rate does not change when $\gamma$ varies, which can also be found in Figure 1(b) and Figure 1(c). BIn other words, if we fix a large dimension $d$ and increase $\gamma$ (or equivalently, increase the sample size $n$), the optimal rate of excess risk of a spectral algorithm stays invariant in certain ranges. Therefore, in order to improve the rate of excess risk, one has to increase the sample size above a certain threshold.

**Polynomial approximation barrier of spectral algorithms when $s \to 0$.** From Theorem 4.1, when $s$ is close to zero, the convergence rate $d^{-\min\{\gamma-p, s(p+1)\}}$ is unchanged in the range $\gamma \in [p(s+1) + s, (p+1)(s+1))$, and increases in the short range $\gamma \in [p(s+1), p(s+1) + s)$. In other words, the excess risk of spectral algorithms will drop when $\gamma$ exceeds $p(s+1) \approx p$ for any integer $p$ and will stay invariant for most of the other $\gamma$. We term the above phenomenon as the polynomial approximation barrier of spectral algorithms (borrowed from Ghorbani et al. (2021)), and it can be illustrated by Figure 1(a) with $s = 0.01$.

*Remark* 4.4. Ghorbani et al. (2021) discovered the polynomial approximation barrier of KRR. As shown by Figure 5 and Theorem 4 in Ghorbani et al. (2021), if $s = 0$ and the true function falls into $L^2 = [H]^0$, then with high probability we have

$$\left| \mathbb{E}\left( \left\| \hat{f}_{\lambda_\star}^{\text{KRR}} - f_\star \right\|_{L^2}^2 \right) - \left\| \mathrm{P}_{>p} f_\star \right\|_{L^2}^2 \right| \le \varepsilon \left( \left\| f_\star \right\|_{L^2}^2 + \sigma^2 \right), \tag{18}$$

where $p$ is the integer satisfying $\gamma \in [p, p+1)$, $\lambda_\star$ is defined as in Theorem 4 in Ghorbani et al. (2021), $\mathrm{P}_{>\ell}$ means the projection onto polynomials with degree $> \ell$, and $\varepsilon$ is any positive real number. Notice that (18) implies that the excess risk of KRR will drop when $\gamma$ exceeds any integer and will stay invariant for other $\gamma$, and is consistent with our results for spectral algorithms.

## 5 Conclusion

In this paper, we rigorously prove the saturation effect of KRR in large dimensions. Let $s > 0$ and $\gamma > 0$ be fixed real numbers, denote $p$ as the integer satisfying $\gamma \in [p(s+1), (p+1)(s+1))$. Given that the kernel is an inner product kernel defined on the sphere and that $f_\star$ falls into the interpolation space $[\mathcal{H}]^s$, we first show that the convergence rate on the excess risk of large-dimensional kernel gradient flow is $\Theta_{\mathbb{P}}\left( d^{-\min\{\gamma-p, s(p+1)\}} \right) \cdot \mathrm{poly}\left(\ln(d)\right)$ (Theorem 3.1), which is faster than that of KRR given in Zhang et al. (2024). We then determine the improved minimax lower bound as $\Omega\left( d^{-\min\{\gamma-p, s(p+1)\}} \right) / \mathrm{poly}\left(\ln(d)\right)$ (Theorem 3.3). Combining these results, we know that kernel gradient flow is minimax optimal in large dimensions, and KRR is inferior to kernel gradient flow in large dimensions. Our results suggest that previous results on large-dimensional KRR may not be directly extendable to large-dimensional neural networks if the regression function is over-smooth.

In Section 4, we generalize our results to certain spectral algorithms. We determine the convergence rate on the excess risk of large-dimensional spectral algorithms (Theorem 4.1 and Theorem 4.2). From these results, we find several new phenomena exhibited in large-dimensional spectral algorithms, including the saturation effect, the periodic plateau behavior, and the polynomial approximation barrier.

In this paper, we only consider the convergence rate on the excess risk of optimal-tuned large-dimensional spectral algorithms with uniform input distribution on a hypersphere. We believe that several results in fixed-dimensional settings with input distribution on more general domains (e.g., Haas et al. (2024); Li et al. (2024)) can indeed be extended to large-dimensional settings, although we must carefully consider the constants that depend on $d$. Furthermore, we believe that by considering the learning curve of large-dimensional spectral algorithms (i.e., the convergence rate on the excess risk of spectral algorithms with any regularization parameter $\lambda > 0$) or the convergence rate on the excess risk of large-dimensional kernel interpolation (i.e., KRR with $\lambda = 0$), further research can find a wealth of new phenomena compared with the fixed-dimensional setting.

## Acknowledgments and Disclosure of Funding

Lin's research was supported in part by the National Natural Science Foundation of China (Grant 92370122, Grant 11971257). The authors are grateful to the reviewers for their constructive comments that greatly improved the quality and presentation of this paper.

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

# A Graphical illustration and numerical experiments of main results

## A.1 Graphical illustration of Theorem 3.1, Theorem 4.1, and Theorem 4.2

Recall that Theorem 3.1, Theorem 4.1, and Theorem 4.2 determined the convergence rate on the excess risk of: (i) large-dimensional kernel gradient flow with $s > 0$; (ii) large-dimensional spectral algorithm with $\tau \geq 1$ and $s \leq \tau$; and (iii) large-dimensional spectral algorithm with $\tau \geq 1$ and $s > \tau$.

In Figure 1, we have provided a visual illustration of Theorem 4.1 and Theorem 4.2 when $\tau = 2$. Now, in Figure 2, we provide more visual illustrations of the results of spectral algorithms with $\tau = 1$, $\tau = 2$, $\tau = 4$, and $\tau = \infty$, which correspond to kernel ridge regression (KRR), iterated ridge regression in Example 3, and kernel gradient flow.

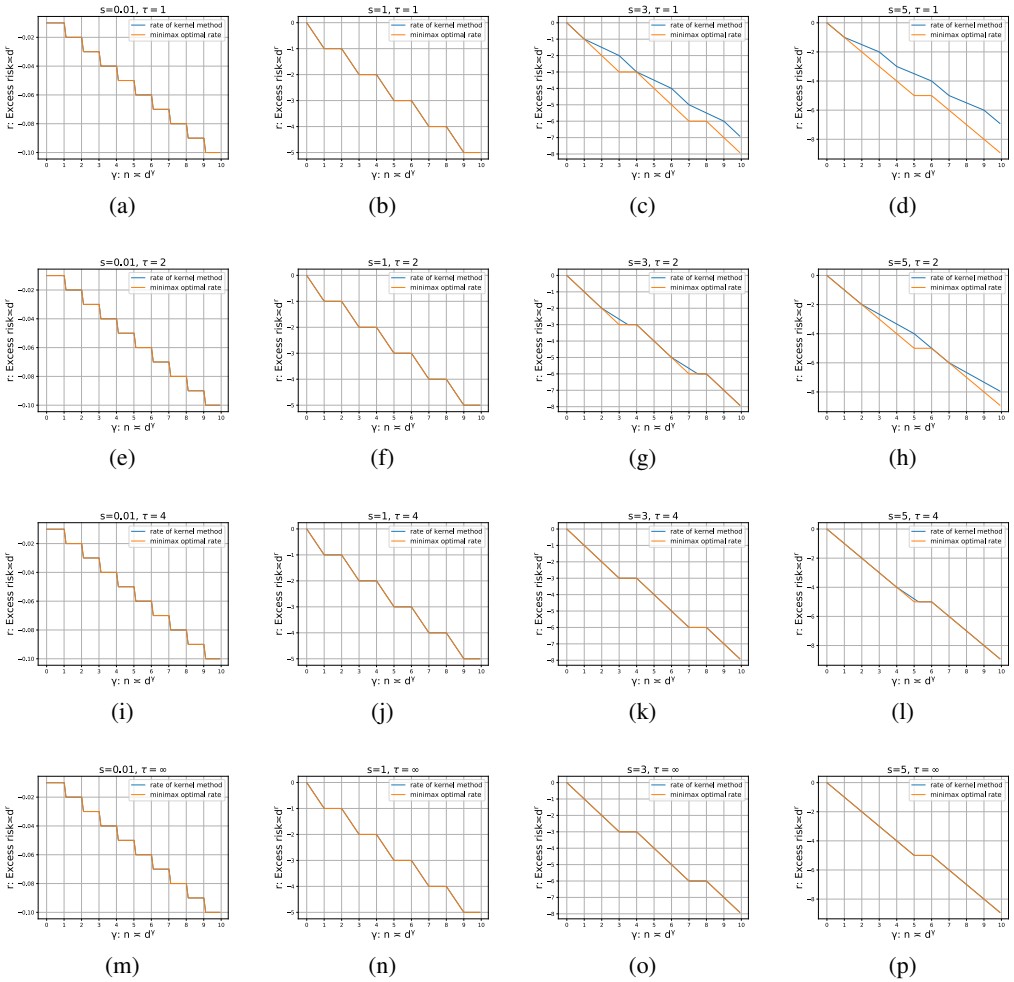

Figure 2: Convergence rates of spectral algorithms with qualification $\tau = 1$ (KRR), $\tau = 2$ (iterated ridge regression), $\tau = 4$ (iterated ridge regression), and $\tau = \infty$ (kernel gradient flow) in Theorem 4.1, Theorem 4.2, and corresponding minimax lower rates in Theorem 3.3 with respect to dimension $d$. We present four graphs corresponding to four kinds of source conditions: $s = 0.01, 1, 3, 5$. The x-axis represents asymptotic scaling, $\gamma : n \asymp d^{\gamma}$; the y-axis represents the convergence rate of excess risk, $r :$ Excess risk $\asymp d^r$.

## A.2 Numerical experiments

We conducted two experiments using two specific kernels: the RBF kernel and the NTK kernel. Experiment 1 was designed to confirm the optimal rate of kernel gradient flow and KRR when $s = 1$. Experiment 2 was designed to illustrate the saturation effect of KRR when $s > 1$.

**Experiment 1:** We consider the following two inner product kernels:

(i) RBF kernel with a fixed bandwidth:

$$K^{\mathrm{rbf}}(x, x') = \exp\left\{\left(-\frac{\|x - x'\|_2^2}{2}\right)\right\}, \quad x, x' \in \mathbb{S}^d.$$

(ii) Neural Tangent Kernel (NTK) of a two-layer ReLU neural network:

$$K^{\mathrm{ntk}}(x, x') := \Phi(\langle x, x' \rangle), \quad x, x' \in \mathbb{S}^d,$$

where $\Phi(t) = [\sin(\arccos t) + 2(\pi - \arccos t)t]/(2\pi)$.

The RBF kernel satisfies Assumption 1. For the NTK, the coefficients of $\Phi(\cdot)$, $\{a_j\}_{j=0}^\infty$, satisfy $a_j > 0, j \in \{0, 1\} \cup \{2, 4, 6, \ldots\}$ and $a_j = 0, j \in \{3, 5, 7, \ldots\}$ (see, e.g., Lu et al. (2023)). As noted after Assumption 1, our results can be extended to inner product kernels with certain zero coefficients $a_j$. Specifically, for any $\gamma > 0$, as long as $a_j > 0$ for $j = \lfloor\gamma\rfloor, \lfloor\gamma\rfloor + 1$, the proof and convergence rate remain the same. Therefore, for $\gamma < 2$ in our experiments, the convergence rates for NTK will be the same as for the RBF kernel.

We used the following data generation procedure:

$$y_i = f_*(x_i) + \epsilon_i, \quad i = 1, \ldots, n,$$

where each $x_i$ is i.i.d. sampled from the uniform distribution on $\mathbb{S}^d$, and $\epsilon_i \overset{\text{i.i.d.}}{\sim} \mathcal{N}(0, 1)$.

We selected the training sample sizes $n$ with corresponding dimensions $d$ such that $n = d^\gamma, \gamma = 0.5, 1.0, 1.5, 1.8$. For each kernel and dimension $d$, we consider the following regression function $f_*$:

$$f_*(x) = K(u_1, x) + K(u_2, x) + K(u_3, x), \quad \text{for some} \quad u_1, u_2, u_3 \in \mathbb{S}^d. \tag{19}$$

This function is in the RKHS $\mathcal{H}$, and it is easy to prove that, for any $u_0 \in \mathbb{S}^d$, Assumption 2 (b) holds for $K(u_0, \cdot)$ with $s = 1$. Therefore, Assumption 2 holds for $s = 1$. We used logarithmic least squares to fit the excess risk with respect to the sample size, resulting in the convergence rate $r$. As shown in Figure 3 and Figure 4, the experimental results align well with our theoretical findings.

**Experiment 2:** We use most of the settings from Experiment 1, except that the regression function is changed to $f_*(x) = \sqrt{\mu_2^s N(d, 2)} P_2(<\xi, x>)$ with $s = 1.9$, $P_2(t) := (dt^2 - 1)/(d - 1)$ the Gegenbauer polynomial, and $\xi \in \mathbb{S}^d$. Notice that the addition formula $P_2(<\xi, x>) = \frac{1}{N(d,2)} \sum_{j=1}^{N(d,2)} Y_{2,j}(\xi) Y_{2,j}(x)$ implies that

$$\|f_*\|_{[\mathcal{H}]^s}^2 = \frac{1}{N(d, 2)} \sum_{j=1}^{N(d,2)} Y_{2,j}^2(\xi) = P_2(1) = 1,$$

hence $f_* \in [\mathcal{H}]^s$ and satisfies Assumption 2.

Our experiment settings are similar to those on page 30 of Li et al. (2022). We choose the regularization parameter for KRR and kernel gradient flow as $\lambda = 0.05 \cdot d^{-\theta}$. For KRR, since Corollary D.16 suggests that the optimal regularization parameter is $\lambda \asymp d^{-0.7}$, we set $\theta = 0.7$. Similarly, based on Corollary D.16, we set $\theta = 0.5$ for kernel gradient flow. Additionally, we set $\gamma = 1.8$. The results indicate that the best convergence rate of KRR is slower than that of kernel gradient flow, implying that KRR is inferior to kernel gradient flow when the regression function is sufficiently smooth.

## B  Proof of Theorem 3.3

We first restate Theorem 3.3.

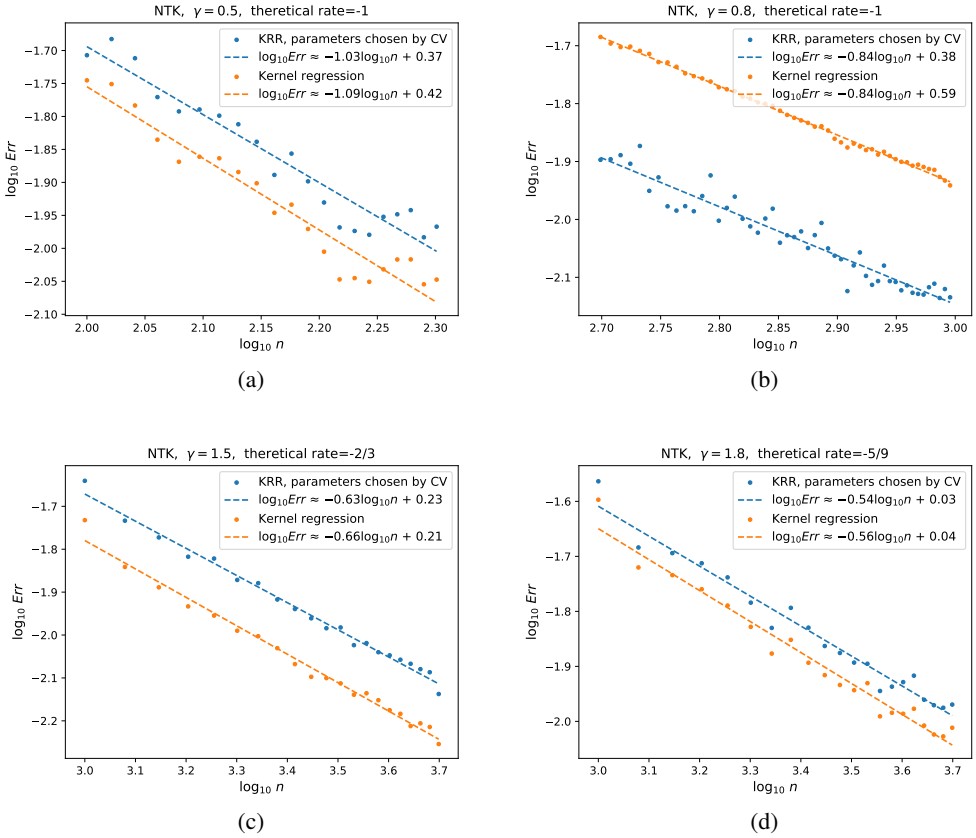

Figure 3: Results of Experiment 1. We repeated each experiment 50 times and reported the average excess risk for (a) kernel gradient flow (labeled as "kernel regression" in our reports) and (b) kernel ridge regression (KRR) on 1000 test samples. We randomly selected $u_1, u_2, u_3$ and kept them fixed for each repeat. We choose the stopping time $t$ in kernel gradient flow as $C_1 n^{0.5}$, where $C_1 \in \{0.001, 0.01, 0.1, 1, 10, 100, 1000\}$. We use 5-fold cross-validation to select the regularization parameter $\lambda$ in kernel ridge regression. The alternative values of $\lambda$ in cross-validation are $C_2 n^{-C_3}$, where $C_2 \in \{0.001, 0.005, 0.01, 0.1, 0.5, 1, 2, 5, 10, 40, 100, 300, 1000\}, C_3 \in \{0.1, 0.2, \dots, 1.5\}$.

**Theorem B.1** (Restate Theorem 3.3). *Let $s > 0$ and $\gamma > 0$ be fixed real numbers. Denote $p$ as the integer satisfying $\gamma \in [p(s+1), (p+1)(s+1))$. Let $\mathcal{P}$ consist of all the distributions $\rho$ on $\mathcal{X} \times \mathcal{Y}$ such that Assumption 1 and Assumption 2 hold for $s$ and $\gamma$. Then for any $d \geq \mathfrak{C}$, a sufficiently large constant only depending on $s$, $\gamma$, $c_1$, and $c_2$, we have the following claims:*

*(i) When $\gamma \in (p(s+1), p+ps+s]$, we have*

$$\min_{\hat{f}} \max_{\rho \in \mathcal{P}} \mathbb{E}_{(X,Y) \sim \rho^{\otimes n}} \left\| \hat{f} - f_\star \right\|_{L^2}^2 \geq \frac{\ln \ln(d)}{50(\gamma - p(s+1))(\ln(d))^2} d^{p-\gamma}.$$

*(ii) When $\gamma \in (p+ps+s, (p+1)(s+1)]$, we have*

$$\min_{\hat{f}} \max_{\rho \in \mathcal{P}} \mathbb{E}_{(X,Y) \sim \rho^{\otimes n}} \left\| \hat{f} - f_\star \right\|_{L^2}^2 = \Omega \left( d^{-s(p+1)} \right),$$

*where $\Omega$ only involves constants depending on $s, \sigma, \gamma, c_0, \kappa, c_1$, and $c_2$.*

*Proof of Theorem B.1.* The item (ii) is a direct corollary of Theorem 5 in Zhang et al. (2024). Now we begin to proof the item (i). We need the following lemma.

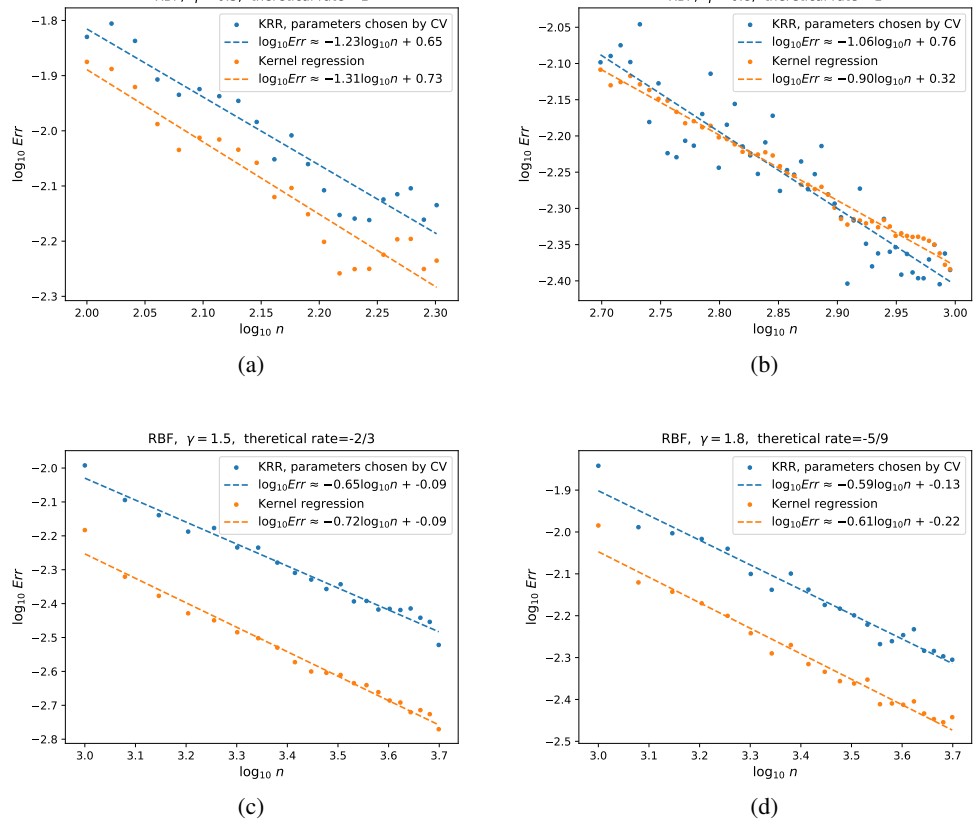

Figure 4: A similar plot as Figure 3, but with the RBF kernel.

**Lemma B.2** (Restate Lemma 4.1 in Lu et al. (2023)). *For any $\delta \in (0,1)$ and any $0 < \tilde{\varepsilon}_1, \tilde{\varepsilon}_2 < \infty$ only depending on $n$, $d$, $\{\lambda_j\}$, $c_1$, $c_2$, and $\gamma$ and satisfying*

$$\frac{V_K(\tilde{\varepsilon}_2, \mathcal{D}) + n\tilde{\varepsilon}_2^2 + \ln(2)}{V_2(\tilde{\varepsilon}_1, \mathcal{B})} \leq \delta, \tag{20}$$

*we have*

$$\min_{\hat{f}} \max_{\rho \in \mathcal{P}} \mathbb{E}_{(X,Y) \sim \rho^{\otimes n}} \left\| \hat{f} - f_\star \right\|_{L^2}^2 \geq \frac{1-\delta}{4} \tilde{\varepsilon}_1^2, \tag{21}$$

*where $\rho_{f_\star}$ is the joint-p.d.f. of $x, y$ given by (1) with $f = f_\star$, $\mathcal{B} := \left\{ f \in \mathcal{H}, \ \|f\|_{[\mathcal{H}]^s} \leq R_\gamma \right\}$*

$$\mathcal{D} := \left\{ \rho_f \ \middle| \ \text{joint distribution of } (y,x) \text{ where } x \sim \rho_\mathcal{X}, y = f(x) + \epsilon, \epsilon \sim N(0, \sigma^2), f \in \mathcal{B} \right\},$$

*and $V_2$, $V_K$ are the $\varepsilon$-covering entropies ( as defined in Yang and Barron (1999); Lu et al. (2023)) of $(\mathcal{B}, d^2 = \| \cdot \|_{L^2}^2)$ and $(\mathcal{D}, d^2 = $ KL divergence ).*

Suppose $\gamma \in (p(s+1), p + ps + s]$. Let $C(p) = \mathfrak{C}_{12}/10$ be a constant only depending on $\gamma$, where $\mathfrak{C}_{12}$ are given in Lemma D.13. Then we introduce

$$\tilde{\varepsilon}_1^2 \triangleq d^{p-\gamma}/\ln(d) \text{ and } \tilde{\varepsilon}_2^2 \triangleq C(p)\frac{d^p}{n} \ln\ln(d). \tag{22}$$

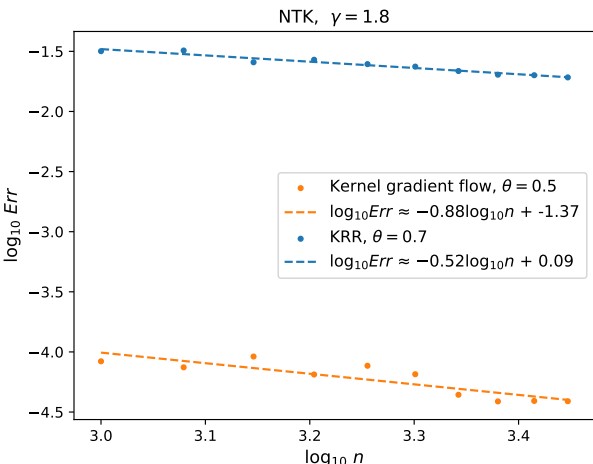

Figure 5: Results of Experiment 2. It can be seen that the best rate of excess risk for KRR is slower than that of kernel gradient flow.

Let us further assume that $d \geq \mathfrak{C}$, where $\mathfrak{C}$ is a sufficiently large constant only depending on $\gamma$, $s$, and $c_1$. By Lemma D.11 and Lemma D.13 we have

$$\tilde{\varepsilon}_1^2 = d^{p-\gamma}/\ln(d) < \frac{\mathfrak{C}_9}{d^{ps}} \leq \mu_p^s$$

$$\mu_{p+1}^s < \tilde{\varepsilon}_2^2 = C(p)\frac{d^p}{n}\ln\ln(d) \leq \frac{C(p)}{c_1}d^{p-\gamma}\ln\ln(d) < \mu_p^s \tag{23}$$

$$n\tilde{\varepsilon}_2^2 \overset{\text{Definition of } \mathfrak{C}_{12}}{\leq} \frac{1}{10}N(d,p)\ln\ln(d).$$

Therefore, for any $d \geq \mathfrak{C}$, where $\mathfrak{C}$ is a sufficiently large constant only depending on $s$, $\gamma$, and $c_1$, we have

$$V_2(\tilde{\varepsilon}_1, \mathcal{B}) \overset{\text{Lemma A.5 in Lu et al. (2023)}}{\geq} K(\tilde{\varepsilon}_1) \geq \frac{1}{2}N(d,p)\ln\left(\frac{\mu_p^s}{\tilde{\varepsilon}_1^2}\right)$$

$$\overset{\text{Definition of } \tilde{\varepsilon}_1^2}{\geq} \frac{1}{2}N(d,p)\ln\left(\mathfrak{C}_9 d^{\gamma-p(s+1)}\ln(d)\right) \tag{24}$$

$$\geq \frac{1}{2}N(d,p)\left[(\gamma-p(s+1))\ln(d) + \frac{1}{2}\ln\ln(d)\right].$$

On the other hand, from Lemma D.11, Lemma D.13, and Lemma D.12, one can check the following claim:

**Claim 1.** *Suppose $\gamma \in (p(s+1), p+ps+s]$. For any $d \geq \mathfrak{C}$, where $\mathfrak{C}$ is a sufficiently large constant only depending on $s$, $\gamma$, $c_1$, and $c_2$, we have*

$$K\left(\sqrt{2}\sigma\tilde{\varepsilon}_2/6\right) \leq \frac{1}{2}N(d,p)\ln\left(\frac{18\mu_p^s}{\sigma^2\tilde{\varepsilon}_2^2}\ln\ln(d)\right).$$

Therefore, for any $d \geq \mathfrak{C}$, where $\mathfrak{C}$ is a sufficiently large constant only depending on $s, \gamma, c_1$, and $c_2$, we have

$$
\begin{aligned}
V_K(\tilde{\varepsilon}_2, \mathcal{D}) = V_2(\sqrt{2}\sigma\tilde{\varepsilon}_2, \mathcal{B}) &\stackrel{\text{Lemma A.5 in Lu et al. (2023)}}{\leq} K\left(\sqrt{2}\sigma\tilde{\varepsilon}_2/6\right) \\
&\stackrel{\text{Claim 1}}{\leq} \frac{1}{2}N(d,p)\ln\left(\frac{18\mu_p^s}{\sigma^2\tilde{\varepsilon}_2^2}\ln\ln(d)\right) \\
&\stackrel{\text{Definition of } \tilde{\varepsilon}_2^2}{\leq} \frac{1}{2}N(d,p)\ln\left(18\mathfrak{C}_{10}\sigma^{-2}[C(p)]^{-1}c_2 d^{\gamma-p(s+1)}\right) \\
&\leq \frac{1}{2}N(d,p)\left[(\gamma-p(s+1))\ln(d) + \frac{1}{5}\ln\ln(d)\right].
\end{aligned}
\tag{25}
$$

Combining (23), (24), and (25), we finally have:

$$
\frac{V_K(\tilde{\varepsilon}_2, \mathcal{D}) + n\tilde{\varepsilon}_2^2 + \ln(2)}{V_2(\tilde{\varepsilon}_1, \mathcal{B})} \leq \frac{[10(\gamma-p(s+1))\ln(d) + 4\ln\ln(d)]}{[10(\gamma-p(s+1))\ln(d) + 5\ln\ln(d)]} < 1,
$$

and from Lemma B.2, we get

$$
\begin{aligned}
\min_{\hat{f}}\max_{f_\star\in\mathcal{B}}\mathbb{E}_{(\mathbf{X},\mathbf{y})\sim\rho_{f_\star}^{\otimes n}}\left\|\hat{f}-f_\star\right\|_{L^2}^2 &\geq \frac{\ln\ln(d)}{4\ln(d)\left[10(\gamma-p(s+1))\ln(d) + 5\ln\ln(d)\right]}d^{p-\gamma} \\
&\geq \frac{\ln\ln(d)}{50(\gamma-p(s+1))(\ln(d))^2}d^{p-\gamma},
\end{aligned}
$$

finishing the proof. ∎

## C  Definition of analytic filter functions

We first introduce the following definition of analytic filter functions (Bauer et al. (2007); Li et al. (2024)).

*Definition* C.1 (Analytic filter functions). Let $\left\{\varphi_\lambda : [0, \kappa^2] \to \mathbb{R}_{\geq 0} \mid \lambda \in (0,1)\right\}$ be a family of functions indexed with regularization parameter $\lambda$ and define the remainder function

$$
\psi_\lambda(z) := 1 - z\varphi_\lambda(z). \tag{26}
$$

We say that $\{\varphi_\lambda \mid \lambda \in (0,1)\}$ (or simply $\varphi_\lambda(z)$) is an analytic filter function if:

(1) $z\varphi_\lambda(z) \in [0,1]$ is non-decreasing with respect to $z$ and non-increasing with respect to $\lambda$.

(2) The *qualification* of this filter function is $\tau \in [1, \infty]$ such that $\forall\, 0 \leq \tau' \leq \tau$ (and also $\tau' < \infty$), there exist positive constants $\mathfrak{C}_i$ only depending on $\tau'$, $i = 1,2,3,4,5$, such that we have

$$
\varphi_\lambda(z) \geq \mathfrak{C}_1 z^{-1}, \quad \psi_\lambda(z) \leq \mathfrak{C}_2(z/\lambda)^{-\tau'}, \quad \forall\lambda \in (0,1), z > \lambda \tag{27}
$$

$$
\mathfrak{C}_3 \leq \lambda\varphi_\lambda(z) \leq \mathfrak{C}_4, \quad \psi_\lambda(z) \geq \mathfrak{C}_5, \quad \forall\lambda \in (0,1), z \leq \lambda. \tag{28}
$$

(3) If $\tau < \infty$, then there exists a positive constant $\mathfrak{C}_6$ only depending on $\tau$ and $\lambda_1$, such that we have

$$
\psi_\lambda(\lambda_1) \geq \mathfrak{C}_6\lambda^\tau, \tag{29}
$$

where $\lambda_1$ is the largest eigenvalue of $K$ defined in (5); and there exist positive constants $\mathfrak{C}_7$ and $\mathfrak{C}_8$ only depending on $\tau$, such that we have

$$
(z/\lambda)^{2\tau}\psi_\lambda^2(z) \geq \mathfrak{C}_7, \quad \forall\lambda \in (0,1), z > \lambda \tag{30}
$$

$$
(z/\lambda)^{2\tau}\psi_\lambda^2(z) \leq \mathfrak{C}_8 z\varphi_\lambda(z), \quad \forall\lambda \in (0,1), z \leq \lambda. \tag{31}
$$

(4) Let

$$
\begin{aligned}
D_\lambda = &\left\{z \in \mathbb{C} : \operatorname{Re} z \in [-\lambda/2, \kappa^2], \, |\operatorname{Im} z| \leq \operatorname{Re} z + \lambda/2\right\} \\
&\cup \left\{z \in \mathbb{C} : \left|z - \kappa^2\right| \leq \kappa^2 + \lambda/2, \, \operatorname{Re} z \geq \kappa^2\right\};
\end{aligned}
$$

Then $\varphi_\lambda(z)$ can be extended to be an analytic function on some domain containing $D_\lambda$ and the following conditions holds for all $\lambda \in (0,1)$:

(C1) $|(z + \lambda)\varphi_\lambda(z)| \leq \tilde{E}$ for all $z \in D_\lambda$;

(C2) $|(z + \lambda)\psi_\lambda(z)| \leq \tilde{F}\lambda$ for all $z \in D_\lambda$;

where $\tilde{E}, \tilde{F}$ are positive constants.

*Remark* C.2. We remark that some of the above properties are not essential for the definition of filter functions in the literature (Bauer et al., 2007; Gerfo et al., 2008), but we introduce them to avoid some unnecessary technicalities in the proof. The requirements of analytic filter functions are first considered in Li et al. (2024) and used for their "analytic functional argument", which will also be vital in our proof.

The following examples show many commonly used analytic filter functions and their proofs can be found in Lemma C.3, see also Li et al. (2024).

**Example 3** (Iterated ridge regression). *Let $q \geq 1$ be fixed. We define*

$$\varphi_\lambda^{\mathrm{IT},q}(z) = \frac{1}{z}\left[1 - \frac{\lambda^q}{(z + \lambda)^q}\right], \quad \psi_\lambda^{\mathrm{IT},q}(z) = \frac{\lambda^q}{(z + \lambda)^q}, \quad \tau = q. \tag{32}$$

**Example 4** (Kernel gradient descent). *The gradient descent method is the discrete version of gradient flow. Let $\eta > 0$ be a fixed step size. Then, iterating gradient descent with respect to the empirical loss $t$ steps yields the filter function*

$$\varphi_\lambda^{\mathrm{GD}}(z) = \eta \sum_{k=0}^{t-1}(1 - \eta z)^k = \frac{1 - (1 - \eta z)^t}{z}, \quad \lambda = (\eta t)^{-1}, \tag{33}$$

$$\psi_\lambda^{\mathrm{GD}}(z) = (1 - \eta z)^t, \quad \tau = \infty. \tag{34}$$

*Moreover, when $\eta$ is small enough, say $\eta < 1/(2\kappa^2)$, we have $\mathrm{Re}(1 - \eta z) > 0$ for $z \in D_\lambda$, so we can take the single-valued branch of $(1 - \eta z)^t$ even when $t$ is not an integer. Therefore, we can extend the definition of the filter function so that $\lambda$ can be arbitrary and $t = (\eta\lambda)^{-1}$.*

**Lemma C.3.** *$\varphi_\lambda^{\mathrm{KRR}}$, $\varphi_\lambda^{\mathrm{IT},q}$, $\varphi_\lambda^{\mathrm{GF}}$, and $\varphi_\lambda^{\mathrm{GD}}$ are analytic filter functions.*

*Proof.* Notice that (i) $z \leq z + \lambda \leq 2z$ when $z > \lambda$; and that (ii) $\lambda \leq z + \lambda \leq 2\lambda$ when $z \leq \lambda$. Hence, the constants $\mathfrak{C}_1$, $\mathfrak{C}_2$, $\mathfrak{C}_3$, $\mathfrak{C}_4$, and $\mathfrak{C}_6$ are given in Li et al. (2024).

For $\mathfrak{C}_5$, when $z \leq \lambda$, we can take $\mathfrak{C}_5 = \min\{1/2, 2^{-q}, e^{-1}, e^{-1}\} > 0$.

For $\mathfrak{C}_7$, when $z > \lambda$, we have

$$(z/\lambda)^{2\tau}(\psi_\lambda^{\mathrm{KRR}}(z))^2 = \left(\frac{z}{z + \lambda}\right)^2 \geq 1/4$$

$$(z/\lambda)^{2\tau}(\psi_\lambda^{\mathrm{IT},q}(z))^2 = \left(\frac{z}{z + \lambda}\right)^{2q} \geq 2^{-2q}.$$

For $\mathfrak{C}_8$, when $z \leq \lambda$, we have

$$\frac{z^{2\tau-1}(\psi_\lambda^{\mathrm{KRR}}(z))^2}{\lambda^{2\tau}\varphi_\lambda^{\mathrm{KRR}}(z)} = \frac{z}{z + \lambda} \leq \frac{1}{2}$$

$$\frac{z^{2\tau-1}(\psi_\lambda^{\mathrm{IT},q}(z))^2}{\lambda^{2\tau}\varphi_\lambda^{\mathrm{IT},q}(z)} = \frac{z^{2q}}{(z + \lambda)^{2q} - [\lambda(z + \lambda)]^q} \leq \frac{1}{2^{2q} - 2^q}.$$

$\blacksquare$

# D   Proof of Theorem 4.1 and Theorem 4.2

## D.1   Bias-variance decomposition

We first apply a standard bias-variance decomposition on the excess risk of spectral algorithms, and readers can also refer to Zhang et al. (2023, 2024) for more details.

Recall the definition of $\hat{g}_Z$ and $\hat{f}_\lambda$ in (14) and (15). Let's define their conditional expectations as

$$\tilde{g}_Z := \mathbb{E}\left(\hat{g}_Z|X\right) = \frac{1}{n}\sum_{i=1}^{n} K_{x_i} f_\star(x_i) \in \mathcal{H}; \tag{35}$$

and

$$\tilde{f}_\lambda := \mathbb{E}\left(\hat{f}_\lambda|X\right) = \varphi_\lambda\left(T_X\right)\tilde{g}_Z \in \mathcal{H}. \tag{36}$$

Let's also define their expectations as

$$g = \mathbb{E}\hat{g}_Z = \int_{\mathcal{X}} K(x,\cdot)f_\star(x)\,\mathrm{d}\rho_{\mathcal{X}}(x) \in \mathcal{H}, \tag{37}$$

and

$$f_\lambda = \varphi_\lambda\left(T\right)g. \tag{38}$$

Then we have the decomposition

$$\begin{aligned}
\hat{f}_\lambda - f_\star &= \frac{1}{n}\varphi_\lambda\left(T_X\right)\sum_{i=1}^{n} K_{x_i}y_i - f_\star \\
&= \frac{1}{n}\varphi_\lambda\left(T_X\right)\sum_{i=1}^{n} K_{x_i}(f_\rho^*(x_i) + \epsilon_i) - f_\star \\
&= \varphi_\lambda\left(T_X\right)\tilde{g}_Z + \frac{1}{n}\sum_{i=1}^{n}\varphi_\lambda\left(T_X\right)K_{x_i}\epsilon_i - f_\star \\
&= \left(\tilde{f}_\lambda - f_\star\right) + \frac{1}{n}\sum_{i=1}^{n}\varphi_\lambda\left(T_X\right)K_{x_i}\epsilon_i.
\end{aligned} \tag{39}$$

Taking expectation over the noise $\epsilon$ conditioned on $X$ and noticing that $\epsilon|X$ are independent noise with mean 0 and variance $\sigma^2$, we obtain the bias-variance decomposition:

$$\mathbb{E}\left(\left\|\hat{f}_\lambda - f_\star\right\|_{L^2}^2 \,\Big|\, X\right) = \mathbf{Bias}^2(\lambda) + \mathbf{Var}(\lambda), \tag{40}$$

where

$$\mathbf{Bias}^2(\lambda) := \left\|\tilde{f}_\lambda - f_\star\right\|_{L^2}^2, \quad \mathbf{Var}(\lambda) := \frac{\sigma^2}{n^2}\sum_{i=1}^{n}\left\|\varphi_\lambda\left(T_X\right)K(x_i,\cdot)\right\|_{L^2}^2. \tag{41}$$

Given the decomposition (40), we next derive the upper and lower bounds of $\mathbf{Bias}^2(\lambda)$ and $\mathbf{Var}(\lambda)$ in the following two subsections.

Before we close this subsection, let's introduce some quantities and an assumption that will be used frequently in our proof later. Denote the true function as $f_\star = \sum_{i=1}^{\infty} f_i\phi_i(x)$, let's define the following quantities:

$$\begin{gathered}
\mathcal{N}_{1,\varphi}(\lambda) = \sum_{j=1}^{\infty}\left[\lambda_j\varphi_\lambda(\lambda_j)\right]; \;\; \mathcal{N}_{2,\varphi}(\lambda) = \sum_{j=1}^{\infty}\left[\lambda_j\varphi_\lambda(\lambda_j)\right]^2; \\
\mathcal{M}_{1,\varphi}(\lambda) = \operatorname*{ess\,sup}_{x\in\mathcal{X}}\left|\sum_{j=1}^{\infty}\left(\psi_\lambda(\lambda_j)f_j\phi_j(x)\right)\right|; \;\; \mathcal{M}_{2,\varphi}(\lambda) = \sum_{j=1}^{\infty}\left(\psi_\lambda(\lambda_j)f_j\right)^2;
\end{gathered} \tag{42}$$

moreover, when $\varphi_\lambda = \varphi_\lambda^{\mathrm{KRR}}$, we denote $\mathcal{N}_k(\lambda) = \mathcal{N}_{k,\varphi^{\mathrm{KRR}}}(\lambda)$ and $\mathcal{M}_k(\lambda) = \mathcal{M}_{k,\varphi^{\mathrm{KRR}}}(\lambda)$ for simplicity, where $k = 1, 2$.

*Assumption* 3. Suppose that

$$\operatorname*{ess\,sup}_{x\in\mathcal{X}}\sum_{j=1}^{\infty}\left[\lambda_j\varphi_\lambda(\lambda_j)\right]^2\phi_j^2(x) \leq \mathcal{N}_{2,\varphi}(\lambda); \tag{43}$$

and

$$\operatorname*{ess\,sup}_{x \in \mathcal{X}} \sum_{j=1}^{\infty} \left[\lambda_j \varphi_\lambda(\lambda_j)\right] \phi_j^2(x) \le \mathcal{N}_{1,\varphi}(\lambda); \tag{44}$$

and

$$\operatorname*{ess\,sup}_{x \in \mathcal{X}} \sum_{j=1}^{\infty} \left[\lambda_j \varphi_\lambda^{\mathsf{KRR}}(\lambda_j)\right] \phi_j^2(x) \le \mathcal{N}_1(\lambda). \tag{45}$$

For simplicity of notations, we denote $h_x(\cdot) = K(x, \cdot)$, $x \in \mathcal{X}$ in the rest of the proof. Moreover, we denote $T_\lambda := (T + \lambda)^{-1}$ and $T_{X\lambda} := (T_X + \lambda)^{-1}$.

### D.2 Variance term

The following proposition rewrites the variance term using the empirical semi-norm.

**Proposition D.1** (Restate Lemma 9 in Zhang et al. (2024))**.** *The variance term in* (41) *satisfies that*

$$\mathbf{Var}(\lambda) = \frac{\sigma^2}{n} \int_{\mathcal{X}} \|\varphi_\lambda(T_X) h_x(\cdot)\|_{L^2,n}^2 \, \mathrm{d}\rho_{\mathcal{X}}(x). \tag{46}$$

The operator form (46) allows us to apply concentration inequalities and establish the following two-step approximation.

$$\int_{\mathcal{X}} \|\varphi_\lambda(T_X) h_x\|_{L^2,n}^2 \, \mathrm{d}\rho_{\mathcal{X}}(x) \overset{\mathbf{A}}{\approx} \int_{\mathcal{X}} \|\varphi_\lambda(T) h_x\|_{L^2,n}^2 \, \mathrm{d}\rho_{\mathcal{X}}(x) \overset{\mathbf{B}}{\approx} \int_{\mathcal{X}} \|\varphi_\lambda(T) h_x\|_{L^2}^2 \, \mathrm{d}\rho_{\mathcal{X}}(x). \tag{47}$$

**Approximation B** The following lemma characterizes the magnitude of Approximation B in high probability. Recall the definitions of $\mathcal{N}_{1,\varphi}(\lambda)$ and $\mathcal{N}_{2,\varphi}(\lambda)$ in (42).

**Lemma D.2** (Approximation B)**.** *Suppose that (43) in Assumption 3 holds. Then, for any fixed* $\delta \in (0, 1)$*, with probability at least* $1 - \delta$*, we have*

$$\frac{1}{2} \int_{\mathcal{X}} \|\varphi_\lambda(T) h_x\|_{L^2}^2 \, \mathrm{d}\rho_{\mathcal{X}}(x) - R_2 \tag{48}$$

$$\le \int_{\mathcal{X}} \|\varphi_\lambda(T) h_x\|_{L^2,n}^2 \, \mathrm{d}\rho_{\mathcal{X}}(x) \tag{49}$$

$$\le \frac{3}{2} \int_{\mathcal{X}} \|\varphi_\lambda(T) h_x\|_{L^2}^2 \, \mathrm{d}\rho_{\mathcal{X}}(x) + R_2, \tag{50}$$

*where*

$$R_2 = \frac{5\mathcal{N}_{2,\varphi}(\lambda)}{3n} \ln \frac{2}{\delta}. \tag{51}$$

*Proof.* Define a function

$$\begin{aligned} f(z) &= \int_{\mathcal{X}} \left(\varphi_\lambda(T) h_x(z)\right)^2 \mathrm{d}\rho_{\mathcal{X}}(x) \\ &= \int_{\mathcal{X}} \sum_{j=1}^{\infty} \left(\lambda_j \varphi_\lambda(\lambda_j)\right)^2 \phi_j^2(x) \phi_j^2(z) \mathrm{d}\rho_{\mathcal{X}}(x) \\ &= \sum_{j=1}^{\infty} \left(\lambda_j \varphi_\lambda(\lambda_j)\right)^2 \phi_j^2(z). \end{aligned} \tag{52}$$

Since (43) in Assumption 3 holds, we have

$$\|f\|_{L^\infty} \le \mathcal{N}_{2,\varphi}(\lambda); \quad \|f\|_{L^1} = \mathcal{N}_{2,\varphi}(\lambda).$$

Applying Proposition 34 in Zhang et al. (2024) for $\sqrt{f}$ and noticing that $\|\sqrt{f}\|_{L^\infty} = \sqrt{\|f\|_{L^\infty}} = \mathcal{N}_{2,\varphi}(\lambda)^{\frac{1}{2}}$, we have

$$\frac{1}{2}\left\|\sqrt{f}\right\|_{L^2}^2 - \frac{5\mathcal{N}_{2,\varphi}(\lambda)}{3n}\ln\frac{2}{\delta} \leq \left\|\sqrt{f}\right\|_{L^2,n}^2 \leq \frac{3}{2}\left\|\sqrt{f}\right\|_{L^2}^2 + \frac{5\mathcal{N}_{2,\varphi}(\lambda)}{3n}\ln\frac{2}{\delta}, \qquad (53)$$

with probability at least $1 - \delta$.

On the one hand, we have

$$\begin{aligned}
\left\|\sqrt{f}\right\|_{L^2,n}^2 &= \int_{\mathcal{X}} f(z)\mathrm{d}P_n(z) = \int_{\mathcal{X}}\left[\int_{\mathcal{X}}(\varphi_\lambda(T)h_x(z))^2\,\mathrm{d}\rho_{\mathcal{X}}(x)\right]\mathrm{d}P_n(z) \\
&= \int_{\mathcal{X}}\left[\int_{\mathcal{X}}(\varphi_\lambda(T)h_x(z))^2\,\mathrm{d}P_n(z)\right]\mathrm{d}\rho_{\mathcal{X}}(x) \\
&= \int_{\mathcal{X}}\|\varphi_\lambda(T)h_x\|_{L^2,n}^2\,\mathrm{d}\rho_{\mathcal{X}}(x).
\end{aligned}$$

On the other hand, we have

$$\begin{aligned}
\left\|\sqrt{f}\right\|_{L^2}^2 &= \int_{\mathcal{X}} f(z)\mathrm{d}\rho_{\mathcal{X}}(z) \\
&= \int_{\mathcal{X}}\left[\int_{\mathcal{X}}(\varphi_\lambda(T)h_x(z))^2\,\mathrm{d}\rho_{\mathcal{X}}(x)\right]\mathrm{d}\rho_{\mathcal{X}}(z) \\
&= \int_{\mathcal{X}}\|\varphi_\lambda(T)h_x\|_{L^2}^2\,\mathrm{d}\rho_{\mathcal{X}}(x).
\end{aligned}$$

Therefore, (53) implies the desired results. ∎

### Approximation A

**Lemma D.3.** *Suppose that (43) and (45) in Assumption 3 hold. Suppose that there exists a constant $\epsilon$ only depending on $s$ and $\gamma$, such that $\lambda = \lambda(n, d)$ satisfies $n^{\epsilon - 1}\mathcal{N}_1(\lambda) \to 0$. Then there exists an absolute constant $C_1$, such that for any fixed $\delta \in (0, 1)$, when $n$ is sufficiently large, with probability at least $1 - \delta$, we have*

$$\left|\int_{\mathcal{X}}\|\varphi_\lambda(T_X)h_x\|_{L^2,n}^2\,\mathrm{d}\rho_{\mathcal{X}}(x) - \int_{\mathcal{X}}\|\varphi_\lambda(T)h_x\|_{L^2,n}^2\,\mathrm{d}\rho_{\mathcal{X}}(x)\right| \qquad (54)$$

$$\leq C_1\left(\sqrt{\mathcal{N}_{2,\varphi}(\lambda)} + C_1\sqrt{v\mathcal{N}_1(\lambda)}\ln\lambda^{-1}\right)\cdot\sqrt{v\mathcal{N}_1(\lambda)}\ln\lambda^{-1}, \qquad (55)$$

*where $v = \frac{\mathcal{N}_1(\lambda)}{n}\ln n$.*

*Remark* D.4. The proof of Lemma D.3 is mainly based on Lemma 4.18 in Li et al. (2024). Notice that we replace the Assumption 2 in Li et al. (2024) by (45) in Assumption 3 (borrowed from Zhang et al. (2024)), since both of them can deduce same results given by Lemma 4.2 in Li et al. (2024) or Lemma 37 in Zhang et al. (2024).

*Proof.* We start with

$$\mathbf{D} = \left|\|\varphi_\lambda(T_X)h_x\|_{L^2} - \|\varphi_\lambda(T)h_x\|_{L^2}\right| \leq \left\|T^{\frac{1}{2}}\left[\varphi_\lambda(T) - \varphi_\lambda(T_X)\right]h_x\right\|_{\mathcal{H}}.$$

Using operator calculus, we get

$$\begin{aligned}
&T^{\frac{1}{2}}\left[\varphi_\lambda(T) - \varphi_\lambda(T_X)\right]h_x \\
&= T^{\frac{1}{2}}\left[\frac{1}{2\pi i}\oint_{\Gamma_\lambda} R_{T_X}(z)(T - T_X)R_T(z)\varphi_\lambda(z)\mathrm{d}z\right]h_x \\
&= \frac{1}{2\pi i}\oint_{\Gamma_\lambda} T^{\frac{1}{2}}(T_X - z)^{-1}(T - T_X)(T - z)^{-1}h_x\varphi_\lambda(z)\mathrm{d}z \\
&= \frac{1}{2\pi i}\oint_{\Gamma_\lambda} T^{\frac{1}{2}}T_\lambda^{-\frac{1}{2}}\cdot T_\lambda^{\frac{1}{2}}(T_X - z)^{-1}T_\lambda^{\frac{1}{2}}\cdot T_\lambda^{-\frac{1}{2}}(T - T_X)T_\lambda^{-\frac{1}{2}}\cdot T_\lambda^{\frac{1}{2}}(T - z)^{-1}T_\lambda^{\frac{1}{2}}\cdot T_\lambda^{-\frac{1}{2}}h_x\varphi_\lambda(z)\mathrm{d}z.
\end{aligned}$$

Therefore, taking the norms yields

$$\mathbf{D} \leq \frac{1}{2\pi} \left\| T^{\frac{1}{2}} T_\lambda^{-\frac{1}{2}} \right\| \cdot \left\| T_\lambda^{\frac{1}{2}} (T_X - z)^{-1} T_\lambda^{\frac{1}{2}} \right\| \cdot \left\| T_\lambda^{-\frac{1}{2}} (T - T_X) T_\lambda^{-\frac{1}{2}} \right\| \cdot \left\| T_\lambda^{\frac{1}{2}} (T - z)^{-1} T_\lambda^{\frac{1}{2}} \right\|$$

$$\cdot \left\| T_\lambda^{-\frac{1}{2}} h_x \right\|_{\mathcal{H}} \oint_{\Gamma_\lambda} |\varphi_\lambda(z) \mathrm{d}z|$$

$$= \frac{1}{2\pi} \cdot \mathbf{I} \cdot \mathbf{II} \cdot \mathbf{III} \cdot \mathbf{IV} \cdot \mathbf{V} \cdot \oint_{\Gamma_\lambda} |\varphi_\lambda(z) \mathrm{d}z|$$

$$\leq \frac{1}{2\pi} \cdot 1 \cdot \sqrt{6} C \cdot \sqrt{\frac{\mathcal{N}_1(\lambda)}{n} \ln n} \cdot C \cdot \sqrt{\mathcal{N}_1(\lambda)} \oint_{\Gamma_\lambda} |\varphi_\lambda(z) \mathrm{d}z|,$$

where in the second estimation, we use (**I**) operator calculus, (**II** and **IV**) Proposition E.8, (**III**) Lemma E.7, and (**V**) Lemma 37 in Zhang et al. (2024) for each term respectively. Finally, from (63) in Li et al. (2024), we get

$$\oint_{\Gamma_\lambda} |\varphi_\lambda(z) \mathrm{d}z| \leq C \ln \lambda^{-1}, \tag{56}$$

and thus there exists an absolute constant $C_1$, such that we have

$$\mathbf{D} = |\|\varphi_\lambda(T_X) h_x\|_{L^2} - \|\varphi_\lambda(T) h_x\|_{L^2}| \leq C_1 \sqrt{v \mathcal{N}_1(\lambda)} \ln \lambda^{-1}.$$

On the other hand, combining (52) and (43) in Assumption 3, we have $\|\varphi_\lambda(T) h_x\|_{L^2}^2 \leq \mathcal{N}_{2,\varphi}(\lambda)$, and hence

$$\|\varphi_\lambda(T_X) h_x\|_{L^2} + \|\varphi_\lambda(T) h_x\|_{L^2} \leq 2\|\varphi_\lambda(T) h_x\|_{L^2} + \mathbf{D}$$

$$\leq \sqrt{\mathcal{N}_{2,\varphi}(\lambda)} + C_1 \sqrt{v \mathcal{N}_1(\lambda)} \ln \lambda^{-1}.$$

Finally,

$$\left| \|\varphi_\lambda(T_X) h_x\|_{L^2}^2 - \|\varphi_\lambda(T) h_x\|_{L^2}^2 \right|$$

$$= |\|\varphi_\lambda(T_X) h_x\|_{L^2} - \|\varphi_\lambda(T) h_x\|_{L^2}| \left( \|\varphi_\lambda(T_X) h_x\|_{L^2} + \|\varphi_\lambda(T) h_x\|_{L^2} \right)$$

$$\leq C_1 \left( \sqrt{\mathcal{N}_{2,\varphi}(\lambda)} + C_1 \sqrt{v \mathcal{N}_1(\lambda)} \ln \lambda^{-1} \right) \cdot \sqrt{v \mathcal{N}_1(\lambda)} \ln \lambda^{-1},$$

and hence

$$\left| \int_{\mathcal{X}} \|\varphi_\lambda(T_X) h_x\|_{L^2,n}^2 \mathrm{d}\rho_{\mathcal{X}}(x) - \int_{\mathcal{X}} \|\varphi_\lambda(T) h_x\|_{L^2,n}^2 \mathrm{d}\rho_{\mathcal{X}}(x) \right|$$

$$\leq \frac{1}{n} \sum_{i=1}^{n} \left| \|\varphi_\lambda(T_X) h_{x_i}\|_{L^2}^2 - \|\varphi_\lambda(T) h_{x_i}\|_{L^2}^2 \right|$$

$$\leq \sup_{x \in \mathcal{X}} \left| \|\varphi_\lambda(T_X) h_x\|_{L^2}^2 - \|\varphi_\lambda(T) h_x\|_{L^2}^2 \right|$$

$$\leq C_1 \left( \sqrt{\mathcal{N}_{2,\varphi}(\lambda)} + C_1 \sqrt{v \mathcal{N}_1(\lambda)} \ln \lambda^{-1} \right) \cdot \sqrt{v \mathcal{N}_1(\lambda)} \ln \lambda^{-1},$$

∎

**Final proof of the variance term**  Now we are ready to state the theorem about the variance term.

**Theorem D.5.** *Suppose that (43) and (45) in Assumption 3 hold. Suppose there exists a constant $\epsilon > 0$ only depending on $s$ and $\gamma$, such that $\lambda = \lambda(n, d)$ satisfies*

$$\mathcal{N}_1(\lambda) \cdot n^{\epsilon-1} \to 0, \tag{57}$$

$$\frac{\mathcal{N}_1^2(\lambda)}{n \mathcal{N}_{2,\varphi}(\lambda)} \cdot \ln(n) (\ln \lambda^{-1})^2 \to 0; \tag{58}$$

*then we have*

$$\mathbf{Var}(\lambda) = [1 + o_{\mathbb{P}}(1)] \frac{\sigma^2}{n} \mathcal{N}_{2,\varphi}(\lambda). \tag{59}$$

*Proof.* Recall that $\mathbf{Var}(\lambda) = \frac{\sigma^2}{n} \int_{\mathcal{X}} \|\varphi_\lambda (T_X) h_x\|_{L^2,n}^2 \, \mathrm{d}\rho_{\mathcal{X}}(x)$. Hence, when $n$ is large enough, with probability at least $1 - \delta$ we have

$$
\left| \int_{\mathcal{X}} \|\varphi_\lambda (T_X) h_x\|_{L^2,n}^2 \, \mathrm{d}\rho_{\mathcal{X}}(x) - \int \|\varphi_\lambda(T)h_x\|_{L^2}^2 \mathrm{d}\rho_{\mathcal{X}}(x) \right|
$$

$$
\leq \left| \int_{\mathcal{X}} \|\varphi_\lambda (T_X) h_x\|_{L^2,n}^2 \, \mathrm{d}\rho_{\mathcal{X}}(x) - \int_{\mathcal{X}} \|\varphi_\lambda (T) h_x\|_{L^2,n}^2 \, \mathrm{d}\rho_{\mathcal{X}}(x) \right|
$$

$$
+ \left| \int_{\mathcal{X}} \|\varphi_\lambda (T) h_x\|_{L^2,n}^2 \, \mathrm{d}\rho_{\mathcal{X}}(x) - \int_{\mathcal{X}} \|\varphi_\lambda(T)h_x\|_{L^2}^2 \mathrm{d}\rho_{\mathcal{X}}(x) \right|
$$

$$
\overset{\text{Lemma D.2}}{\leq} \left| \int_{\mathcal{X}} \|\varphi_\lambda (T_X) h_x\|_{L^2,n}^2 \, \mathrm{d}\rho_{\mathcal{X}}(x) - \int_{\mathcal{X}} \|\varphi_\lambda (T) h_x\|_{L^2,n}^2 \, \mathrm{d}\rho_{\mathcal{X}}(x) \right| + \frac{5\mathcal{N}_{2,\varphi}(\lambda)}{3n} \ln \frac{2}{\delta}
$$

$$
\overset{\text{Lemma D.3}}{\leq} \left( \sqrt{\mathcal{N}_{2,\varphi}(\lambda)} \cdot C_1 \sqrt{v\mathcal{N}_1(\lambda)} \ln \lambda^{-1} + C_1^2 v\mathcal{N}_1(\lambda)(\ln \lambda^{-1})^2 \right) + \frac{5\mathcal{N}_{2,\varphi}(\lambda)}{3n} \ln \frac{2}{\delta}
$$

$$
\overset{\text{Definition of } v}{=} \sqrt{\frac{\mathcal{N}_{2,\varphi}(\lambda)}{n}} \mathcal{N}_1(\lambda) \cdot C_1 \sqrt{\ln(n)} \ln \lambda^{-1} + \frac{\mathcal{N}_1^2(\lambda)}{n} \cdot C_1^2 \ln(n)(\ln \lambda^{-1})^2 + \frac{\mathcal{N}_{2,\varphi}(\lambda)}{n} \cdot \frac{5}{3} \ln \frac{2}{\delta}
$$

$$
= \mathbf{I} \cdot C_1 \sqrt{\ln(n)} \ln \lambda^{-1} + \mathbf{II} \cdot C_1^2 \ln(n)(\ln \lambda^{-1})^2 + \mathbf{III} \cdot \frac{5}{3} \ln \frac{2}{\delta}.
$$

When $n \geq \mathfrak{C}$, a sufficiently large constant only depending on $\gamma$ and $C_1$, we have

$$
\mathbf{I} \cdot C_1 \sqrt{\ln(n)} \ln \lambda^{-1} \leq \frac{1}{6} \mathcal{N}_{2,\varphi}(\lambda).
$$

Furthermore, when $\frac{\mathcal{N}_1^2(\lambda)}{n\mathcal{N}_{2,\varphi}(\lambda)} \cdot n^\epsilon \to 0$, we have $\mathbf{I} \cdot C_1 \sqrt{\ln(n)} \ln \lambda^{-1}/\mathcal{N}_{2,\varphi}(\lambda) \to 0$ and $\mathbf{II} \cdot C_1^2 \ln(n)(\ln \lambda^{-1})^2/\mathcal{N}_{2,\varphi}(\lambda) \to 0$.

Finally, from (52) we have

$$
\|\varphi_\lambda(T)h_x\|_{L^2}^2 = \sum_{i=1}^\infty (\lambda_j \varphi_\lambda(\lambda_j))^2 \phi_i^2(z),
$$

and thus the deterministic term writes

$$
\int_{\mathcal{X}} \|\varphi_\lambda(T)h_x\|_{L^2}^2 \mathrm{d}\rho_{\mathcal{X}}(x) = \mathcal{N}_{2,\varphi}(\lambda).
$$

∎

## D.3 Bias term

In this subsection, our goal is to determine the upper and lower bounds of bias under some approximation conditions.

The triangle inequality implies that

$$
\begin{aligned}
\mathbf{Bias}(\lambda) &= \left\| \tilde{f}_\lambda - f_\star \right\|_{L^2} \geq \|f_\lambda - f_\star\|_{L^2} - \left\| \tilde{f}_\lambda - f_\lambda \right\|_{L^2} \\
\mathbf{Bias}(\lambda) &\leq \|f_\lambda - f_\star\|_{L^2} + \left\| \tilde{f}_\lambda - f_\lambda \right\|_{L^2}.
\end{aligned}
\tag{60}
$$

The following lemma characterizes the dominant term of $\mathbf{Bias}(\lambda)$.

**Lemma D.6.** *For any $\lambda > 0$, we have*

$$
\|f_\lambda - f_\star\|_{L^2} = \mathcal{M}_{2,\varphi}(\lambda)^{\frac{1}{2}}.
\tag{61}
$$

*Proof.* We have

$$\|f_\lambda - f_\star\|_{L^2}^2 = \left\|\sum_{i=1}^\infty \lambda_i \varphi_\lambda(\lambda_i) f_i \phi_i(x) - \sum_{i=1}^\infty f_i \phi_i(x)\right\|_{L^2}^2$$

$$= \left\|\sum_{i=1}^\infty \psi_\lambda(\lambda_i) f_i \phi_i(x)\right\|_{L^2}^2$$

$$= \sum_{i=1}^\infty (\psi_\lambda(\lambda_i) f_i)^2$$

$$= \mathcal{M}_{2,\varphi}(\lambda).$$

∎

The following lemma bounds the remainder term of **Bias**$(\lambda)$ when $s \geq 1$.

**Lemma D.7.** *Suppose that (45) in Assumption 3 holds. Suppose that there exist constants $\epsilon$ and $\mathfrak{C}$ only depending on $s$ and $\gamma$, such that $\lambda = \lambda(n, d)$ satisfies*

$$n^{\epsilon-1}\mathcal{N}_1(\lambda) \to 0, \tag{62}$$

$$\frac{\mathcal{N}_1(\lambda)\mathcal{M}_{1,\varphi}^2(\lambda)}{n^2} = o\left(\mathcal{M}_{2,\varphi}(\lambda) + \frac{\sigma^2}{n}\mathcal{N}_{2,\varphi}(\lambda)\right), \tag{63}$$

$$\frac{\mathcal{N}_1(\lambda)}{n}\ln(n)(\ln\lambda^{-1})^2 \cdot \sum_{j=1}^\infty \frac{\lambda^2 \lambda_i \varphi_\lambda^2(\lambda_i)}{\lambda + \lambda_i} f_i^2 = o\left(\mathcal{M}_{2,\varphi}(\lambda) + \frac{\sigma^2}{n}\mathcal{N}_{2,\varphi}(\lambda)\right); \tag{64}$$

*then we have*

$$\left\|\tilde{f}_\lambda - f_\lambda\right\|_{L^2}^2 = o_\mathbb{P}\left(\mathcal{M}_{2,\varphi}(\lambda) + \frac{\sigma^2}{n}\mathcal{N}_{2,\varphi}(\lambda)\right). \tag{65}$$

*Proof.* Do the decomposition,

$$\begin{aligned}
\tilde{f}_\lambda - f_\lambda &= \varphi_\lambda(T_X)\tilde{g}_X - (\psi_\lambda(T_X) + \varphi_\lambda(T_X)T_X)f_\lambda \\
&= \varphi_\lambda(T_X)(\tilde{g}_X - T_X f_\lambda) - \psi_\lambda(T_X)T\varphi_\lambda(T)f_\star \\
&= \varphi_\lambda(T_X)(\tilde{g}_X - T_X f_\lambda) - \varphi_\lambda(T_X)\psi_\lambda(T)g + \varphi_\lambda(T_X)\psi_\lambda(T)g - \psi_\lambda(T_X)T\varphi_\lambda(T)f_\star \\
&= \varphi_\lambda(T_X)\left[\tilde{g}_X - T_X f_\lambda - \psi_\lambda(T)g\right] + [\varphi_\lambda(T_X)\psi_\lambda(T)Tf_\star - \psi_\lambda(T_X)T\varphi_\lambda(T)f_\star] \\
&= \varphi_\lambda(T_X)(\tilde{g}_X - T_X f_\lambda - g + Tf_\lambda) + (\varphi_\lambda(T_X)T\psi_\lambda(T) - \psi_\lambda(T_X)T\varphi_\lambda(T))f_\star \\
&= \mathbf{I} + \mathbf{II}.
\end{aligned} \tag{66}$$

**Bound on I:** For the first term in (66), we have

$$\begin{aligned}
\|\mathbf{I}\|_{L^2} &= \|\varphi_\lambda(T_X)(\tilde{g}_X - T_X f_\lambda - g + Tf_\lambda)\|_{L^2} \\
&= \left\|T^{\frac{1}{2}}\varphi_\lambda(T_X)(\tilde{g}_X - T_X f_\lambda - g + Tf_\lambda)\right\|_{\mathcal{H}} \\
&\leq \left\|T^{\frac{1}{2}}T_\lambda^{-\frac{1}{2}}\right\| \cdot \left\|T_\lambda^{\frac{1}{2}}\varphi_\lambda(T_X)T_\lambda^{\frac{1}{2}}\right\| \cdot \left\|T_\lambda^{-\frac{1}{2}}\left[(\tilde{g}_X - T_X f_\lambda) - (g - Tf_\lambda)\right]\right\|_{\mathcal{H}} \\
&\overset{\text{(72) in Zhang et al. (2024)}}{\leq} \left\|T_\lambda^{\frac{1}{2}}\varphi_\lambda(T_X)T_\lambda^{\frac{1}{2}}\right\| \cdot \left\|T_\lambda^{-\frac{1}{2}}\left[(\tilde{g}_X - T_X f_\lambda) - (g - Tf_\lambda)\right]\right\|_{\mathcal{H}} \\
&\overset{\text{Proposition } E.1}{\leq} 4\left\|T_\lambda^{\frac{1}{2}}T_{X\lambda}^{-1}T_\lambda^{\frac{1}{2}}\right\| \cdot \left\|T_\lambda^{-\frac{1}{2}}\left[(\tilde{g}_X - T_X f_\lambda) - (g - Tf_\lambda)\right]\right\|_{\mathcal{H}} \\
&\overset{\text{(62) and (73) in Zhang et al. (2024)}}{\leq} 12\left\|T_\lambda^{-\frac{1}{2}}\left[(\tilde{g}_X - T_X f_\lambda) - (g - Tf_\lambda)\right]\right\|_{\mathcal{H}},
\end{aligned}$$

Denote $\xi_i = \xi(x_i) = T_\lambda^{-\frac{1}{2}}(K_{x_i} f_\star(x_i) - T_{x_i} f_\lambda)$. To use Bernstein inequality, we need to bound the $m$-th moment of $\xi(x)$:

$$
\begin{aligned}
\mathbb{E} \|\xi(x)\|_{\mathcal{H}}^m &= \mathbb{E} \left\| T_\lambda^{-\frac{1}{2}} K_x (f_\star - f_\lambda(x)) \right\|_{\mathcal{H}}^m \\
&\leq \mathbb{E}\left( \left\| T_\lambda^{-\frac{1}{2}} K(x, \cdot) \right\|_{\mathcal{H}}^m \mathbb{E}\big( |(f_\star - f_\lambda(x))|^m \mid x \big) \right).
\end{aligned}
\tag{67}
$$

Note that Lemma 37 in Zhang et al. (2024) shows that

$$
\left\| T_\lambda^{-\frac{1}{2}} K(x, \cdot) \right\|_{\mathcal{H}} \leq \mathcal{N}_1(\lambda)^{\frac{1}{2}}, \quad \mu\text{-a.e. } x \in \mathcal{X};
$$

By definition of $\mathcal{M}_{1,\varphi}(\lambda)$, we also have

$$
\|f_\lambda - f_\star\|_{L^\infty} = \left\| \sum_{i=1}^\infty \psi_\lambda(\lambda_i) f_i \phi_i(x) \right\|_{L^\infty} = \mathcal{M}_{1,\varphi}(\lambda).
\tag{68}
$$

In addition, we have proved in Lemma D.6 that

$$
\mathbb{E}|(f_\lambda(x) - f_\star(x))|^2 = \mathcal{M}_{2,\varphi}(\lambda).
$$

So we get the upper bound of (67), i.e.,

$$
\begin{aligned}
(67) &\leq \mathcal{N}_1(\lambda)^{\frac{m}{2}} \cdot \|f_\lambda - f_\star\|_{L^\infty}^{m-2} \cdot \mathbb{E}|(f_\lambda(x) - f_\star(x))|^2 \\
&= \mathcal{N}_1(\lambda)^{\frac{m}{2}} \mathcal{M}_{1,\varphi}(\lambda)^{m-2} \mathcal{M}_{2,\varphi}(\lambda) \\
&= \left( \mathcal{N}_1(\lambda)^{\frac{1}{2}} \mathcal{M}_{1,\varphi}(\lambda) \right)^{m-2} \left( \mathcal{N}_1(\lambda)^{\frac{1}{2}} \mathcal{M}_{2,\varphi}(\lambda)^{\frac{1}{2}} \right)^2.
\end{aligned}
$$

Using Lemma 36 in Zhang et al. (2024) with therein notations: $L = \mathcal{N}_1(\lambda)^{\frac{1}{2}} \mathcal{M}_{1,\varphi}(\lambda)$ and $\sigma = \mathcal{N}_1(\lambda)^{\frac{1}{2}} \mathcal{M}_{2,\varphi}(\lambda)^{\frac{1}{2}}$, for any fixed $\delta \in (0,1)$, with probability at least $1 - \delta$, we have

$$
\|\mathbf{I}\|_{L^2} \leq 12 \cdot 4\sqrt{2} \log \frac{2}{\delta} \left( \frac{\mathcal{N}_1(\lambda)^{\frac{1}{2}} \mathcal{M}_{1,\varphi}(\lambda)}{n} + \frac{\mathcal{N}_1(\lambda)^{\frac{1}{2}} \mathcal{M}_{2,\varphi}(\lambda)^{\frac{1}{2}}}{\sqrt{n}} \right).
\tag{69}
$$

**Bound on II:** For the second term in (66), we have

$$
\begin{aligned}
\|\mathbf{II}\|_{L^2} &= \|(\varphi_\lambda(T_X) T \psi_\lambda(T) - \psi_\lambda(T_X) T \varphi_\lambda(T)) f_\star\|_{L^2} \\
&\leq \left\| T^{\frac{1}{2}} (\varphi_\lambda(T_X) T \psi_\lambda(T) - \psi_\lambda(T) T \varphi_\lambda(T)) f_\star \right\|_{\mathcal{H}} \\
&\quad + \left\| T^{\frac{1}{2}} (\psi_\lambda(T_X) T \varphi_\lambda(T) - \psi_\lambda(T) T \varphi_\lambda(T)) f_\star \right\|_{\mathcal{H}}.
\end{aligned}
\tag{70}
$$

For the first term in (70), we still employ the analytic functional argument:

$$
\begin{aligned}
& T^{\frac{1}{2}} (\varphi_\lambda(T_X) T \psi_\lambda(T) - \psi_\lambda(T) T \varphi_\lambda(T)) f_\star \\
&= T^{\frac{1}{2}} (\varphi_\lambda(T_X) - \varphi_\lambda(T)) T \psi_\lambda(T) f_\star \\
&= \frac{1}{2\pi i} \oint_{\Gamma_\lambda} T^{\frac{1}{2}} (T_X - z)^{-1} (T_X - T)(T - z)^{-1} \varphi_\lambda(z) T \psi_\lambda(T) f_\star \mathrm{d}z \\
&= \frac{1}{2\pi i} \oint_{\Gamma_\lambda} T^{\frac{1}{2}} T_\lambda^{-\frac{1}{2}} \cdot T_\lambda^{\frac{1}{2}} (T_X - z)^{-1} T_\lambda^{\frac{1}{2}} \cdot T_\lambda^{-\frac{1}{2}} (T - T_X) T_\lambda^{-\frac{1}{2}} \\
&\quad \cdot T_\lambda^{\frac{1}{2}} (T - z)^{-1} T_\lambda^{\frac{1}{2}} \cdot T_\lambda^{-\frac{1}{2}} T^{\frac{1}{2}} \cdot T^{\frac{1}{2}} \psi_\lambda(T) f_\star \varphi_\lambda(z) \mathrm{d}z.
\end{aligned}
$$

Therefore,

$$
\begin{aligned}
&2\pi\|T^{\frac{1}{2}}(\varphi_\lambda(T_X)T\psi_\lambda(T) - \psi_\lambda(T)T\varphi_\lambda(T))f_\star\|_{\mathcal{H}} \\
&\leq \oint_{\Gamma_\lambda} \left\|T^{\frac{1}{2}}T_\lambda^{-\frac{1}{2}}\right\| \cdot \left\|T_\lambda^{\frac{1}{2}}(T_X - z)^{-1}T_\lambda^{\frac{1}{2}}\right\| \cdot \left\|T_\lambda^{-\frac{1}{2}}(T - T_X)T_\lambda^{-\frac{1}{2}}\right\| \\
&\qquad \cdot \left\|T_\lambda^{\frac{1}{2}}(T - z)^{-1}T_\lambda^{\frac{1}{2}}\right\| \cdot \left\|T_\lambda^{-\frac{1}{2}}T^{\frac{1}{2}}\right\| \cdot \left\|T^{\frac{1}{2}}\psi_\lambda(T)f_\star\right\|_{\mathcal{H}} |\varphi_\lambda(z)\mathrm{d}z| \\
&\overset{\text{(72) in Zhang et al. (2024)}}{\leq} \oint_{\Gamma_\lambda} \left\|T_\lambda^{\frac{1}{2}}(T_X - z)^{-1}T_\lambda^{\frac{1}{2}}\right\| \cdot \left\|T_\lambda^{-\frac{1}{2}}(T - T_X)T_\lambda^{-\frac{1}{2}}\right\| \\
&\qquad \cdot \left\|T_\lambda^{\frac{1}{2}}(T - z)^{-1}T_\lambda^{\frac{1}{2}}\right\| \cdot \left\|T^{\frac{1}{2}}\psi_\lambda(T)f_\star\right\|_{\mathcal{H}} |\varphi_\lambda(z)\mathrm{d}z| \\
&\overset{\text{(45) and Proposition } E.8}{\leq} \sqrt{6}C^2 \oint_{\Gamma_\lambda} \left\|T_\lambda^{-\frac{1}{2}}(T - T_X)T_\lambda^{-\frac{1}{2}}\right\| \\
&\qquad \cdot \left\|T^{\frac{1}{2}}\psi_\lambda(T)f_\star\right\|_{\mathcal{H}} |\varphi_\lambda(z)\mathrm{d}z| \\
&\overset{\text{Lemma } E.7}{\leq} \sqrt{6}C^2\sqrt{v} \oint_{\Gamma_\lambda} \left\|T^{\frac{1}{2}}\psi_\lambda(T)f_\star\right\|_{\mathcal{H}} |\varphi_\lambda(z)\mathrm{d}z| \\
&\overset{\text{Definition of } \mathcal{M}_{2,\varphi}(\lambda)}{=} \sqrt{6}C^2\sqrt{v}\mathcal{M}_{2,\varphi}^{1/2}(\lambda) \oint_{\Gamma_\lambda} |\varphi_\lambda(z)\mathrm{d}z| \\
&\overset{(56)}{\leq} \sqrt{6}C^3\sqrt{v}\mathcal{M}_{2,\varphi}^{1/2}(\lambda)\ln\lambda^{-1},
\end{aligned}
\tag{71}
$$

where $v = \frac{\mathcal{N}_1(\lambda)}{n}\ln n$.

For the second term in (70), we have

$$
\begin{aligned}
&T^{\frac{1}{2}}(\psi_\lambda(T_X)T\varphi_\lambda(T) - \psi_\lambda(T)T\varphi_\lambda(T))f_\star \\
&= T^{\frac{1}{2}}\left[\frac{1}{2\pi i}\oint_{\Gamma_\lambda} R_{T_X}(z)(T - T_X)R_T(z)\psi_\lambda(z)\mathrm{d}z\right]T\varphi_\lambda(T)f_\star \\
&= \frac{1}{2\pi i}\oint_{\Gamma_\lambda} T^{\frac{1}{2}}(T_X - z)^{-1}(T - T_X)(T - z)^{-1}\psi_\lambda(z)T\varphi_\lambda(T)f_\star \mathrm{d}z \\
&= \frac{1}{2\pi i}\int_{\Gamma_\lambda} T^{\frac{1}{2}}T_\lambda^{-\frac{1}{2}} \cdot T_\lambda^{\frac{1}{2}}(T_X - z)^{-1}T_\lambda^{\frac{1}{2}} \cdot T_\lambda^{-\frac{1}{2}}(T - T_X)T_\lambda^{-\frac{1}{2}} \\
&\qquad \cdot T_\lambda^{\frac{1}{2}}(T - z)^{-1}T_\lambda^{\frac{1}{2}} \cdot T_\lambda^{-\frac{1}{2}}T\varphi_\lambda(T)f_\star\psi_\lambda(z)\mathrm{d}z.
\end{aligned}
$$

Hence, similar to (71), we have

$$
\begin{aligned}
&2\pi\left\|T^{\frac{1}{2}}(\psi_\lambda(T_X)T\varphi_\lambda(T) - \psi_\lambda(T)T\varphi_\lambda(T))f_\star\right\|_{\mathcal{H}} \\
&\leq \int_{\Gamma_\lambda} \left\|T^{\frac{1}{2}}T_\lambda^{-\frac{1}{2}}\right\| \cdot \left\|T_\lambda^{\frac{1}{2}}(T_X - z)^{-1}T_\lambda^{\frac{1}{2}}\right\| \cdot \left\|T_\lambda^{-\frac{1}{2}}(T - T_X)T_\lambda^{-\frac{1}{2}}\right\| \\
&\qquad \cdot \left\|T_\lambda^{\frac{1}{2}}(T - z)^{-1}T_\lambda^{\frac{1}{2}}\right\| \cdot \left\|T_\lambda^{-\frac{1}{2}}T\varphi_\lambda(T)f_\star\right\|_{\mathcal{H}} |\psi_\lambda(z)\mathrm{d}z| \\
&\leq \sqrt{6}C^2\sqrt{v}\left\|T_\lambda^{-\frac{1}{2}}T\varphi_\lambda(T)f_\star\right\|_{\mathcal{H}} \int_{\Gamma_\lambda} |\psi_\lambda(z)\mathrm{d}z| \\
&\overset{\text{Definition of analytic filter functions}}{\leq} \sqrt{6}C^2\sqrt{v}\left\|T_\lambda^{-\frac{1}{2}}T\varphi_\lambda(T)f_\star\right\|_{\mathcal{H}} C\tilde{F}\lambda\ln\lambda^{-1}.
\end{aligned}
\tag{72}
$$

Combining (66), (69), (70), (71), and (72), there exists a constant $\mathfrak{C}_1$ only depending on $\delta$ and $\tilde{F}$, such that we have

$$
\begin{aligned}
&\left\|\tilde{f}_\lambda - f_\lambda\right\|_{L^2} \\
&\leq \mathfrak{C}_1 \left( \frac{\mathcal{N}_1(\lambda)^{\frac{1}{2}} \mathcal{M}_{1,\varphi}(\lambda)}{n} + \frac{\mathcal{N}_1(\lambda)^{\frac{1}{2}} \mathcal{M}_{2,\varphi}(\lambda)^{\frac{1}{2}}}{\sqrt{n}} \right) \\
&\quad + \mathfrak{C}_1 \sqrt{v} \mathcal{M}_{2,\varphi}^{1/2}(\lambda) \ln \lambda^{-1} + \mathfrak{C}_1 \sqrt{v} \left\| T_\lambda^{-\frac{1}{2}} T \varphi_\lambda(T) f_\star \right\|_{\mathcal{H}} \lambda \ln \lambda^{-1} \\
&\stackrel{(62)}{\leq} \left( n^{-1} \mathcal{N}_1(\lambda) \right)^{1/2} \cdot \mathfrak{C}_1 \mathfrak{C}^{1/2} \cdot (\mathcal{M}_{2,\varphi}(\lambda))^{1/2} \\
&\quad + \left( n^{-1} \mathcal{N}_1(\lambda) \right)^{1/2} \cdot \mathfrak{C}_1 \cdot (\mathcal{M}_{2,\varphi}(\lambda))^{1/2} \\
&\quad + \left( n^{\epsilon-1} \mathcal{N}_1(\lambda) \right)^{1/2} \cdot \mathfrak{C}_1 \cdot (\mathcal{M}_{2,\varphi}(\lambda))^{1/2} \\
&\quad + o \left( \mathcal{M}_{2,\varphi}(\lambda) + \frac{\sigma^2}{n} \mathcal{N}_{2,\varphi}(\lambda) \right)^{1/2}.
\end{aligned}
\tag{73}
$$

$\blacksquare$

When $s < 1$, we can use the following lemma to bound the remainder term of $\mathbf{Bias}(\lambda)$. This lemma is a modification of Lemma D.7, and its proof is partly based on Lemma 26 in Zhang.

**Lemma D.8.** *Suppose that (45) in Assumption 3 holds. Suppose that there exist constants $\epsilon$ and $\mathfrak{C}$ only depending on $s$ and $\gamma$, such that $\lambda = \lambda(n, d)$ satisfies*

$$
n^{\epsilon-1} \mathcal{N}_1(\lambda) \to 0,
\tag{74}
$$

$$
\frac{\mathcal{N}_1(\lambda)}{n} \ln(n)(\ln \lambda^{-1})^2 \cdot \sum_{j=1}^{\infty} \frac{\lambda^2 \lambda_i \varphi_\lambda^2(\lambda_i)}{\lambda + \lambda_i} f_i^2 \ll \left( \mathcal{M}_{2,\varphi}(\lambda) + \frac{\sigma^2}{n} \mathcal{N}_{2,\varphi}(\lambda) \right);
\tag{75}
$$

$$
n^{-1} \mathcal{N}_1(\lambda)^{\frac{1}{2}} \left( \|f_\lambda\|_{L^\infty} + n^{\frac{1-s}{2}+\epsilon} \right) = o \left( \mathcal{M}_{2,\varphi}(\lambda) + \frac{\sigma^2}{n} \mathcal{N}_{2,\varphi}(\lambda) \right)^{1/2};
\tag{76}
$$

*then we have*

$$
\left\| \tilde{f}_\lambda - f_\lambda \right\|_{L^2}^2 = o_{\mathbb{P}} \left( \mathcal{M}_{2,\varphi}(\lambda) + \frac{\sigma^2}{n} \mathcal{N}_{2,\varphi}(\lambda) \right).
\tag{77}
$$

*Proof.* Similar to the proof in Lemma D.7, we have the decomposition $\tilde{f}_\lambda - f_\lambda = \mathbf{I} + \mathbf{II}$, with

$$
\|\mathbf{I}\|_{L^2}^2 \leq 12^2 \left\| T_\lambda^{-\frac{1}{2}} \left[ (\tilde{g}_X - T_X f_\lambda) - (g - T f_\lambda) \right] \right\|_{\mathcal{H}}^2,
$$

$$
\|\mathbf{II}\|_{L^2}^2 = o \left( \mathcal{M}_{2,\varphi}(\lambda) + \frac{\sigma^2}{n} \mathcal{N}_{2,\varphi}(\lambda) \right).
$$

Denote $\xi_i = \xi(x_i) = T_\lambda^{-\frac{1}{2}} (K_{x_i} f_\star(x_i) - T_{x_i} f_\lambda)$. Further consider the subset $\Omega_1 = \{x \in \mathcal{X} : |f_\star(x)| \leq t\}$ and $\Omega_2 = \mathcal{X} \backslash \Omega_1$, where $t$ will be chosen appropriately later. Decompose $\xi_i$ as $\xi_i I_{x_i \in \Omega_1} + \xi_i I_{x_i \in \Omega_2}$ and we have the following decomposition:

$$
\left\| T_\lambda^{-\frac{1}{2}} \left[ (\tilde{g}_X - T_X f_\lambda) - (g - T f_\lambda) \right] \right\|_{\mathcal{H}} = \left\| \frac{1}{n} \sum_{i=1}^{n} \xi_i - \mathbb{E} \xi_x \right\|_{\mathcal{H}}
\tag{78}
$$

$$
\leq \left\| \frac{1}{n} \sum_{i=1}^{n} \xi_i I_{x_i \in \Omega_1} - \mathbb{E} \xi_x I_{x \in \Omega_1} \right\|_{\mathcal{H}} + \left\| \frac{1}{n} \sum_{i=1}^{n} \xi_i I_{x_i \in \Omega_2} \right\|_{\mathcal{H}} + \left\| \mathbb{E} \xi_x I_{x \in \Omega_2} \right\|_{\mathcal{H}}
$$

$$
:= \mathrm{I} + \mathrm{II} + \mathrm{III}.
\tag{79}
$$

Next we choose $t = n^{\frac{1-s}{2}+\epsilon_t}, q = \frac{2}{1-s} - \epsilon_q$ such that

$$
\epsilon_t < \epsilon; \quad \text{and} \quad \frac{1-s}{2} + \epsilon_t > 1 / \left( \frac{2}{1-s} - \epsilon_q \right).
\tag{80}
$$

Then we can bound the three terms in (78) as follows:

(i) For the first term in (78), denoted as I, notice that

$$\|(f_\lambda - f_\star)\, I_{x_i \in \Omega_1}\|_{L^\infty} \le \|f_\lambda\|_{L^\infty} + n^{\frac{1-s}{2} + \epsilon_t}. \tag{81}$$

Imitating (67) in the proof of Lemma D.7, we have

$$\mathrm{I} = o_{\mathbb{P}}\left(\mathcal{M}_{2,\varphi}(\lambda) + \frac{\sigma^2}{n}\mathcal{N}_{2,\varphi}(\lambda)\right)^{1/2}. \tag{82}$$

(ii) For the second term in (78), denoted as II. Since $q = \frac{2}{1-s} - \epsilon_q < \frac{2}{1-s}$, Lemma 42 in Zhang et al. (2024) shows that,

$$[\mathcal{H}]^s \hookrightarrow L^q(\mathcal{X}, \mu), \tag{83}$$

with embedding norm less than a constant $C_{s,\kappa}$. Then Assumption 2 (a) implies that there exists $0 < C_q < \infty$ only depending on $\gamma, s$ and $\kappa$ such that $\|f_\star\|_{L^q(\mathcal{X},\mu)} \le C_q$. Using the Markov inequality, we have

$$P(x \in \Omega_2) = P\Big(|f_\star(x)| > t\Big) \le \frac{\mathbb{E}|f_\star(x)|^q}{t^q} \le \frac{(C_q)^q}{t^q}.$$

Further, since (80) guarantees $t^q \gg n$, we have

$$\tau_n := P\left(\mathrm{II} > 0\right) \tag{84}$$

$$\le P\Big(\exists x_i \text{ s.t. } x_i \in \Omega_2,\Big) = 1 - P\Big(x_i \notin \Omega_2, \forall x_i, i = 1, 2, \cdots, n\Big)$$

$$= 1 - P\Big(x \notin \Omega_2\Big)^n$$

$$= 1 - P\Big(|f_\star(x)| \le t\Big)^n$$

$$\le 1 - \left(1 - \frac{(C_q)^q}{t^q}\right)^n \to 0. \tag{85}$$

(iii) For the third term in (78), denoted as III. Since Lemma 37 in Zhang et al. (2024) implies that $\|T_\lambda^{-\frac{1}{2}} k(x, \cdot)\|_{\mathcal{H}} \le \mathcal{N}_1(\lambda)^{\frac{1}{2}}, \mu$-a.e. $x \in \mathcal{X}$, so

$$\mathrm{III} \le \mathbb{E}\|\xi_x I_{x \in \Omega_2}\|_{\mathcal{H}} \le \mathbb{E}\left[\|T_\lambda^{-\frac{1}{2}} k(x, \cdot)\|_{\mathcal{H}} \cdot \left|\big(f_\star - f_\lambda(x)\big) I_{x \in \Omega_2}\right|\right]$$

$$\le \mathcal{N}_1(\lambda)^{\frac{1}{2}} \mathbb{E}\left|\big(f_\star - f_\lambda(x)\big) I_{x \in \Omega_2}\right|$$

$$\le \mathcal{N}_1(\lambda)^{\frac{1}{2}} \|f_\star - f_\lambda\|_{L^2}^{\frac{1}{2}} \cdot P\left(x \in \Omega_2\right)^{\frac{1}{2}}$$

$$\le \mathcal{N}_1(\lambda)^{\frac{1}{2}} \mathcal{M}_{2,\varphi}(\lambda)^{\frac{1}{2}} t^{-\frac{q}{2}}, \tag{86}$$

where we use Cauchy-Schwarz inequality for the third inequality and Lemma D.6 for the fourth inequality. Recalling that the choices of $t, q$ satisfy $t^{-q} \ll n^{-1}$ and we have assumed $n^{\epsilon-1}\mathcal{N}_1(\lambda) \to 0$, we have

$$\mathrm{III} = o\left(\mathcal{M}_{2,\varphi}(\lambda)^{\frac{1}{2}}\right). \tag{87}$$

Plugging (82), (84) and (87) into (78), we finish the proof. ∎

**Final proof of the bias term** Now we are ready to state the theorem about the bias term.

**Theorem D.9** ($s \ge 1$). *Suppose that (45) in Assumption 3 holds. Suppose that there exist constants $\epsilon$ and $\mathfrak{C}$ only depending on $s$ and $\gamma$, such that $\lambda = \lambda(n, d)$ satisfies*

$$n^{\epsilon-1}\mathcal{N}_1(\lambda) \to 0,$$

$$\frac{\mathcal{N}_1(\lambda)\mathcal{M}_{1,\varphi}^2(\lambda)}{n^2} \ll \left(\mathcal{M}_{2,\varphi}(\lambda) + \frac{\sigma^2}{n}\mathcal{N}_{2,\varphi}(\lambda)\right),$$

$$\frac{\mathcal{N}_1(\lambda)}{n} \ln(n)(\ln \lambda^{-1})^2 \cdot \sum_{j=1}^{\infty} \frac{\lambda^2 \lambda_i \varphi_\lambda^2(\lambda_i)}{\lambda + \lambda_i} f_i^2 \ll \left(\mathcal{M}_{2,\varphi}(\lambda) + \frac{\sigma^2}{n}\mathcal{N}_{2,\varphi}(\lambda)\right);$$

*then we have*

$$\left|\mathbf{Bias}^2(\lambda) - \mathcal{M}_{2,\varphi}(\lambda)\right| = o_{\mathbb{P}}\left(\mathcal{M}_{2,\varphi}(\lambda) + \frac{\sigma^2}{n}\mathcal{N}_{2,\varphi}(\lambda)\right). \tag{88}$$

**Theorem D.10** ($s < 1$). *Suppose that (45) in Assumption 3 holds. Suppose that there exist constants $\epsilon$ and $\mathfrak{C}$ only depending on $s$ and $\gamma$, such that $\lambda = \lambda(n, d)$ satisfies*

$$n^{\epsilon-1}\mathcal{N}_1(\lambda) \to 0,$$

$$\frac{\mathcal{N}_1(\lambda)}{n}\ln(n)(\ln\lambda^{-1})^2 \cdot \sum_{j=1}^{\infty}\frac{\lambda^2\lambda_i\varphi_\lambda^2(\lambda_i)}{\lambda+\lambda_i}f_i^2 \ll \left(\mathcal{M}_{2,\varphi}(\lambda) + \frac{\sigma^2}{n}\mathcal{N}_{2,\varphi}(\lambda)\right);$$

$$n^{-1}\mathcal{N}_1(\lambda)^{\frac{1}{2}}\left(\|f_\lambda\|_{L^\infty} + n^{\frac{1-s}{2}+\epsilon}\right) = o\left(\mathcal{M}_{2,\varphi}(\lambda) + \frac{\sigma^2}{n}\mathcal{N}_{2,\varphi}(\lambda)\right)^{1/2};$$

*then we have*

$$\left|\mathbf{Bias}^2(\lambda) - \mathcal{M}_{2,\varphi}(\lambda)\right| = o_\mathbb{P}\left(\mathcal{M}_{2,\varphi}(\lambda) + \frac{\sigma^2}{n}\mathcal{N}_{2,\varphi}(\lambda)\right). \tag{89}$$

### D.4 Quantity calculations and conditions verification for the inner product kernels

In the previous two sections, we have successfully bounded the bias and the variance terms by the quantities $\mathcal{M}_{2,\varphi}(\lambda)$ and $\mathcal{N}_{2,\varphi}(\lambda)$. In this subsection, we will focus on the inner product kernels on the sphere. We will (i) determine the rates for the above quantities, and (ii) verify all the conditions in Theorem D.5, Theorem D.9 and Theorem D.10.

Recall that $\mu_k$ and $N(d, k)$, defined in (9), are the eigenvalues of the inner product kernel $K$ defined on the sphere and the corresponding multiplicity. The following three lemmas (mainly cited from Lu et al. (2023)) give concise characterizations of $\mu_k$ and $N(d, k)$, which is sufficient for the analysis in this paper.

**Lemma D.11.** *For any fixed integer $p \geq 0$, there exist constants $\mathfrak{C}$, $\mathfrak{C}_9$ and $\mathfrak{C}_{10}$ only depending on $p$ and $\{a_j\}_{j\leq p+1}$, such that for any $d \geq \mathfrak{C}$, we have*

$$\mathfrak{C}_9 d^{-k} \leq \mu_k \leq \mathfrak{C}_{10}d^{-k}, \quad k = 0, 1, \cdots, p+1. \tag{90}$$

**Lemma D.12.** *For any fixed integer $p \geq 0$, there exist constants $\mathfrak{C}$ only depending on $p$ and $\{a_j\}_{j\leq p+1}$, such that for any $d \geq \mathfrak{C}$, we have*

$$\mu_k \leq \frac{\mathfrak{C}_{10}}{\mathfrak{C}_9}d^{-1}\mu_p, \quad k = p+1, p+2, \cdots$$

*where $\mathfrak{C}_9$ and $\mathfrak{C}_{10}$ are constants given in Lemma D.11.*

**Lemma D.13.** *For any fixed integer $p \geq 0$, there exist constants $\mathfrak{C}_{11}$, $\mathfrak{C}_{12}$ and $\mathfrak{C}$ only depending on $p$, such that for any $d \geq \mathfrak{C}$, we have*

$$\mathfrak{C}_{11}d^k \leq N(d, k) \leq \mathfrak{C}_{12}d^k, \quad k = 0, 1, \cdots, p+1. \tag{91}$$

With these lemmas, we can begin to bound the quantities $\mathcal{M}_{2,\varphi}(\lambda)$ and $\mathcal{N}_{2,\varphi}(\lambda)$.

**Lemma D.14.** *Suppose that Assumption 1 and Assumption 2 hold for $s$ and an integer $p$. Suppose $\ell \leq p$, $t = \lambda^{-1} \in (d^\ell, d^{\ell+1}]$. Then we have the following bound.*

$$\mathcal{M}_{2,\varphi}(\lambda) = \begin{cases} \Theta\left(d^{-s(\ell+1)}\right) & \tau = \infty \\ \Theta\left(t^{-2\tau}d^{\ell(2\tau-s)} + d^{-s(\ell+1)}\right) & s \leq 2\tau < \infty \\ \Theta\left(\lambda^{2\tau}\right) & s > 2\tau \end{cases}$$

$$\frac{\mathcal{N}_{2,\varphi}(\lambda)}{n} = \Theta\left(\frac{d^\ell}{n} + \frac{t^2}{nd^{\ell+1}}\right) \tag{92}$$

$$\sum_{k=0}^{\infty}\frac{\lambda^2\mu_k\varphi_\lambda^2(\mu_k)}{\lambda+\mu_k}\sum_{j=1}^{N(d,k)}f_{k,j}^2 = O\left(\lambda^2 d^{\max\{p(2-s),0\}} + d^{-s(\ell+1)}\right);$$

*and thus Assumption 3 holds. Moreover, when $s \geq 1$, We have*

$$\mathcal{M}_{1,\varphi}^2(\lambda) = \begin{cases} O\left(d^{-(\ell+1)(s-1)}\right) & \tau = \infty \\ O\left(\lambda^{2\tau-1}d^{\ell(2\tau-s)} + d^{-(\ell+1)(s-1)}\right) & s \leq 2\tau < \infty \\ O\left(\lambda^{2\tau-1}\right) & s > 2\tau \end{cases} \tag{93}$$

*Proof.* **I.** We begin with $\mathcal{M}_{2,\varphi}(\lambda)$. If $s \leq 2\tau$ and $\tau < \infty$, then we have

$$\mathcal{M}_{2,\varphi}(\lambda) = \sum_{k=0}^{\infty} \psi_\lambda^2(\mu_k) \sum_{j=1}^{N(d,k)} f_{k,j}^2$$

$$\leq \sum_{k=0}^{\ell} \mathfrak{C}_2^2 (t\mu_k)^{-2\tau} (\mu_k)^s \sum_{j=1}^{N(d,k)} (\mu_k)^{-s} f_{k,j}^2 + \sum_{k=\ell+1}^{\infty} \psi_\lambda^2(\mu_k) \sum_{j=1}^{N(d,k)} f_{k,j}^2$$

$$\leq \sum_{k=0}^{\ell} \mathfrak{C}_2^2 (t\mu_k)^{-2\tau} (\mu_k)^s \sum_{j=1}^{N(d,k)} (\mu_k)^{-s} f_{k,j}^2 + \sum_{k=\ell+1}^{\infty} (\mu_k)^s \sum_{j=1}^{N(d,k)} (\mu_k)^{-s} f_{k,j}^2$$

$$\leq \mathfrak{C}_2^2 t^{-2\tau} (\mathfrak{C}_9 d^{-\ell})^{s-2\tau} \sum_{k=0}^{\ell} \sum_{j=1}^{N(d,k)} (\mu_k)^{-s} f_{k,j}^2 + (\mathfrak{C}_{10} d^{-\ell-1})^s \sum_{k=\ell+1}^{\infty} \sum_{j=1}^{N(d,k)} (\mu_k)^{-s} f_{k,j}^2$$

$$= O\left( t^{-2\tau} d^{\ell(2\tau-s)} + d^{-s(\ell+1)} \right);$$

and when $\tau = \infty$, a similar argument ( taking $\tau' < \tau$ and let $\tau' \to \infty$, then we have $(td^{-\ell})^{-2\tau'} \to 0$) shows that $\mathcal{M}_{2,\varphi}(\lambda) = O(d^{-s(\ell+1)})$.

Similarly, if $s \leq 2\tau$, then we have

$$\mathcal{M}_{2,\varphi}(\lambda) \geq \mathbf{1}\left\{\tau < \infty\right\} \sum_{k=0}^{\ell} \mathfrak{C}_7^2 (t\mu_k)^{-2\tau} (\mu_k)^s \sum_{j=1}^{N(d,k)} (\mu_k)^{-s} f_{k,j}^2$$

$$+ \sum_{k=\ell+1}^{\infty} \psi_\lambda^2(\mu_k) \sum_{j=1}^{N(d,k)} f_{k,j}^2$$

$$\geq \mathbf{1}\left\{\tau < \infty\right\} \Omega\left( t^{-2\tau} d^{\ell(2\tau-s)} \right)$$

$$+ \sum_{k=\ell+1}^{\infty} \mathfrak{C}_5^2 (\mu_k)^s \sum_{j=1}^{N(d,k)} (\mu_k)^{-s} f_{k,j}^2$$

$$\geq \mathbf{1}\left\{\tau < \infty\right\} \Omega\left( t^{-2\tau} d^{\ell(2\tau-s)} \right)$$

$$+ \mathfrak{C}_5^2 (\mathfrak{C}_{10} d^{-\ell-1})^s \sum_{j=1}^{N(d,\ell)} (\mu_\ell)^{-s} f_{\ell,j}^2$$

$$= \mathbf{1}\left\{\tau < \infty\right\} \Omega\left( t^{-2\tau} d^{\ell(2\tau-s)} \right) + \Omega\left( d^{-s(\ell+1)} \right).$$

If $2\tau < s$, then

$$\mathcal{M}_{2,\varphi}(\lambda) = \sum_{k=0}^{\infty} \psi_\lambda^2(\mu_k) \sum_{j=1}^{N(d,k)} f_{k,j}^2$$

$$\overset{\text{Lemma } E.3}{\leq} \kappa^{2(s-2\tau)} \lambda^{2\tau} \sum_{k=0}^{\infty} \sum_{j=1}^{N(d,k)} \mu_k^{-s} f_{k,j}^2$$

$$= O\left( \lambda^{2\tau} \right).$$

Similarly, if $2\tau < s$, then we have

$$\mathcal{M}_{2,\varphi}(\lambda) \geq \psi_\lambda^2(\mu_0) f_{0,1}^2$$

$$\geq \mathfrak{C}_6^2 f_{0,1}^2 \cdot \lambda^{2\tau}$$

$$= \Omega\left( \lambda^{2\tau} \right).$$

**II.** Now let's bound the second term $\mathcal{N}_{2,\varphi}(\lambda)/n$. We have

$$
\begin{aligned}
\frac{\mathcal{N}_{2,\varphi}(\lambda)}{n} &= \frac{1}{n} \sum_{k=0}^{\infty} N(d,k) \left[\mu_k \varphi_\lambda(\mu_k)\right]^2 \\
&\leq \frac{1}{n} \sum_{k=0}^{\ell} N(d,k) + \frac{1}{n} \sum_{k=\ell+1}^{\infty} N(d,k) \left[\mu_k \varphi_\lambda(\mu_k)\right]^2 \\
&\leq \frac{1}{n} \sum_{k=0}^{\ell} N(d,k) + \frac{\mathfrak{C}_4^2 t^2}{n} \sum_{k=\ell+1}^{\infty} N(d,k)(\mu_k)^2 \\
&\leq \ell \frac{N(d,\ell)}{n} + \frac{\mathfrak{C}_4^2 t^2}{n} \mu_{\ell+1} \\
&= O\left(\frac{d^\ell}{n} + \frac{t^2}{nd^{\ell+1}}\right).
\end{aligned}
\tag{94}
$$

Similarly, we have

$$
\begin{aligned}
\frac{\mathcal{N}_{2,\varphi}(\lambda)}{n} &\geq \frac{\mathfrak{C}_1^2}{n} \sum_{k=0}^{\ell} N(d,k) + \frac{\mathfrak{C}_3^2 t^2}{n} \sum_{k=\ell+1}^{\infty} N(d,k)(\mu_k)^2 \\
&\geq \mathfrak{C}_1^2 \frac{N(d,\ell)}{n} + \frac{\mathfrak{C}_3^2 t^2}{n} \mu_{\ell+1} \\
&= \Omega\left(\frac{d^\ell}{n} + \frac{t^2}{nd^{\ell+1}}\right).
\end{aligned}
\tag{95}
$$

**III.** For the third term, we have

$$
\begin{aligned}
\sum_{k=0}^{\infty} \frac{\lambda^2 \mu_k \varphi_\lambda^2(\mu_k)}{\lambda + \mu_k} \sum_{j=1}^{N(d,k)} f_{k,j}^2 &\leq \lambda^2 R_\gamma^2 \left(\sum_{k=0}^{p} \mu_k^s \varphi_\lambda^2(\mu_k) + \lambda^{-1} \sum_{k=p+1}^{\infty} \mu_k^{s+1} \mathfrak{C}_4^2 \lambda^{-2}\right) \\
&= O\left(\lambda^2 d^{\max\{p(2-s),0\}} + \lambda^{-1} d^{-(s+1)(\ell+1)}\right) \\
&= O\left(\lambda^2 d^{\max\{p(2-s),0\}} + d^{-s(\ell+1)}\right)
\end{aligned}
$$

**IV.** Now we show that Assumption 3 holds. Notice that (45) has been verified in Lemma 20 of Zhang et al. (2024). Similarly, one can prove (43) and (44) hold using a similar proof as that for Lemma 20 of Zhang et al. (2024).

**V.** For the final term, when $s \geq 1$, we have

$$
\begin{aligned}
\mathcal{M}_{1,\varphi}^2(\lambda) &= \operatorname*{ess\,sup}_{\boldsymbol{x} \in \mathcal{X}} \left|\sum_{i=1}^{\infty} (\psi_\lambda(\lambda_i) f_i e_i(\boldsymbol{x}))\right|^2 \\
&\leq \left(\sum_{i=1}^{\infty} \frac{\psi_\lambda(\lambda_i)}{\lambda_i \varphi_\lambda(\lambda_i)} f_i^2\right) \cdot \operatorname*{ess\,sup}_{\boldsymbol{x} \in \mathcal{X}} \sum_{i=1}^{\infty} \left(\lambda_i \varphi_\lambda(\lambda_i) e_i(\boldsymbol{x})^2\right) \\
&\overset{\text{Assumption 3}}{\leq} \left(\sum_{i=1}^{\infty} \frac{\psi_\lambda(\lambda_i)}{\lambda_i \varphi_\lambda(\lambda_i)} f_i^2\right) \cdot \sum_{i=1}^{\infty} \lambda_i \varphi_\lambda(\lambda_i) \\
&:= \mathcal{Q}_{1,\varphi}(\lambda) \cdot \mathcal{N}_{1,\varphi}(\lambda).
\end{aligned}
\tag{96}
$$

For $\mathcal{Q}_{1,\varphi}(\lambda)$, when $\tau \geq s/2$ and $\tau < \infty$, we have

$$
\begin{aligned}
\mathcal{Q}_{1,\varphi}(\lambda) &= \sum_{k=0}^{\infty} \frac{\psi_\lambda^2(\mu_k)\mu_k^{s-1}}{\varphi_\lambda(\mu_k)} \sum_{j=1}^{N(d,k)} \mu_k^{-s} f_{k,j}^2 \\
&\leq \frac{\mathfrak{C}_2^2}{\mathfrak{C}_1} \sum_{k=0}^{\ell} \lambda^{2\tau} \mu_k^{-2\tau+s} \sum_{j=1}^{N(d,k)} \mu_k^{-s} f_{k,j}^2 \\
&\quad + (\mathfrak{C}_3)^{-1}\lambda \sum_{k=\ell+1}^{\infty} \mu_k^{s-1} \sum_{j=1}^{N(d,k)} \mu_k^{-s} f_{k,j}^2 \\
&= O\left(\lambda^{2\tau} d^{\ell(2\tau-s)} + \lambda d^{-(\ell+1)(s-1)}\right).
\end{aligned}
$$

(97)

Similarly, when $\tau = \infty$, we can show that $\mathcal{Q}_{1,\varphi}(\lambda) = O(\lambda d^{-(\ell+1)(s-1)})$.

And when $\tau < s/2$, we have

$$
\begin{aligned}
\mathcal{Q}_{1,\varphi}(\lambda) &= \sum_{k=0}^{\infty} \frac{\psi_\lambda^2(\mu_k)\mu_k^{s-1}}{\varphi_\lambda(\mu_k)} \sum_{j=1}^{N(d,k)} \mu_k^{-s} f_{k,j}^2 \\
&\overset{\text{Lemma } E.3}{\leq} \frac{\mathfrak{C}_2^2 \kappa^{2(s-2\tau)}}{\mathfrak{C}_1} \lambda^{2\tau} \sum_{k=0}^{p} \sum_{j=1}^{N(d,k)} \mu_k^{-s} f_{k,j}^2 \\
&\quad + \sum_{k=p+1}^{\infty} \frac{\psi_\lambda^2(\mu_k)\mu_k^{s-1}}{\varphi_\lambda(\mu_k)} \sum_{j=1}^{N(d,k)} \mu_k^{-s} f_{k,j}^2 \\
&\overset{(30)}{\leq} \frac{\mathfrak{C}_2^2 \kappa^{2(s-2\tau)}}{\mathfrak{C}_1} \lambda^{2\tau} \sum_{k=0}^{p} \sum_{j=1}^{N(d,k)} \mu_k^{-s} f_{k,j}^2 \\
&\quad + \sum_{k=p+1}^{\infty} \mathfrak{C}_8 \lambda^{2\tau} \sum_{j=1}^{N(d,k)} \mu_k^{-s} f_{k,j}^2 \\
&= O\left(\lambda^{2\tau}\right).
\end{aligned}
$$

For $\mathcal{N}_{1,\varphi}(\lambda)$, we have

$$
\begin{aligned}
\mathcal{N}_{1,\varphi}(\lambda) &= \sum_{k=0}^{\infty} N(d,k) \left[\mu_k \varphi_\lambda(\mu_k)\right] \\
&\leq \sum_{k=0}^{\ell} N(d,k) + \sum_{k=\ell+1}^{\infty} N(d,k) \left[\mu_k \varphi_\lambda(\mu_k)\right] \\
&\leq \sum_{k=0}^{\ell} N(d,k) + \mathfrak{C}_4 t \sum_{k=\ell+1}^{\infty} N(d,k)\mu_k \\
&\leq \ell N(d,\ell) + \mathfrak{C}_4 t \\
&= O\left(d^\ell + \lambda^{-1}\right) = O\left(\lambda^{-1}\right).
\end{aligned}
$$

(98)

Therefore, when $s \geq 1$, we have

$$
\mathcal{M}_{1,\varphi}^2(\lambda) = \begin{cases} O\left(d^{-(\ell+1)(s-1)}\right) & \tau = \infty \\ O\left(\lambda^{2\tau-1} d^{\ell(2\tau-s)} + d^{-(\ell+1)(s-1)}\right) & s \leq 2\tau < \infty \\ O\left(\lambda^{2\tau-1}\right) & s > 2\tau \end{cases}
$$

(99)

$\blacksquare$

From Lemma D.14, we have the following three corollaries.

**Corollary D.15.** *Let $1 \leq s \leq \tau$ and $\gamma > 0$ be fixed real numbers. Denote $p$ as the integer satisfying $\gamma \in [p(s+1), (p+1)(s+1))$. Suppose one of the following cases holds for $\lambda^\star = d^{-\ell}$ or $\lambda^\star = d^{-\ell} \cdot poly\,(\ln(d))$:*

*(1) $p \geq 1$, $p(s+1) \leq \gamma < ps + p + s$, $\ell = p + 1/2$*

*(2) $p \geq 1$, $ps + p + s \leq \gamma < ps + p + s + 1$, $\ell = (\gamma - (p+1)(s-1))/2$*

*(3) $\gamma < s$, $\ell = \min\{\gamma, 1\}/2$*

*(4) $s \leq \gamma < s + 1$, $\ell = (\gamma - (s-1))/2$*

*Then we have*

$$\mathcal{M}_{2,\varphi}(\lambda^\star) \lesssim \frac{\mathcal{N}_{2,\varphi}(\lambda^\star)}{n} = \Theta\left(d^{-s(p+1)} + \frac{d^p}{n}\right), \tag{100}$$

*or*

$$\mathcal{M}_{2,\varphi}(\lambda^\star) \lesssim \frac{\mathcal{N}_{2,\varphi}(\lambda^\star)}{n} = \Theta\left(d^{-s(p+1)} + \frac{d^p}{n}\right) \cdot poly\,(\ln(d)). \tag{101}$$

**Corollary D.16.** *Let $\tau < s \leq 2\tau$ and $\gamma > 0$ be fixed real numbers. Denote $p$ as the integer satisfying $\gamma \in [p(s+1), (p+1)(s+1))$. Denote $\Delta = \gamma - p(s+1)$. Suppose one of the following cases holds for $\lambda^\star = d^{-\ell}$ or $\lambda^\star = d^{-\ell} \cdot poly\,(\ln(d))$:*

*(1) $\gamma \geq 1$, $0 \leq \Delta \leq \tau$, $\ell = \ell_1 := p + \Delta/(2\tau)$*

*(2) $\gamma \geq 1$, $\tau \leq \Delta \leq s + s/\tau - 1$, $\ell = \ell_2 := p + (\Delta + 1)/(2\tau + 2)$*

*(3) $\gamma \geq 1$, $\Delta \geq s + s/\tau - 1$, $\ell = \ell_3 := p + (\Delta + 1 - s)/2$*

*(4) $\gamma < 1$, $\ell = \gamma/2$*

*Then we have*

$$\mathcal{M}_{2,\varphi}(\lambda^\star) \asymp \frac{\mathcal{N}_{2,\varphi}(\lambda^\star)}{n} = \Theta\left(d^{-\min\left\{\gamma - p, \frac{\tau(\gamma - p + 1) + ps}{\tau + 1}, s(p+1)\right\}}\right), \tag{102}$$

*or*

$$\mathcal{M}_{2,\varphi}(\lambda^\star) \asymp \frac{\mathcal{N}_{2,\varphi}(\lambda^\star)}{n} = \Theta\left(d^{-\min\left\{\gamma - p, \frac{\tau(\gamma - p + 1) + ps}{\tau + 1}, s(p+1)\right\}}\right) \cdot poly\,(\ln(d)). \tag{103}$$

*Proof.* Denote $\mathbf{I} = -2\ell\tau + 2p\tau - ps$, $\mathbf{II} = -sp - s$, $\mathbf{III} = p - \gamma$, and $\mathbf{IV} = 2\ell - \gamma - p - 1$. From Lemma D.14 we have

$$\mathcal{M}_{2,\varphi}(\lambda^\star) \asymp d^{\mathbf{I}} + d^{\mathbf{II}}, \quad \frac{\mathcal{N}_{2,\varphi}(\lambda^\star)}{n} \asymp d^{\mathbf{III}} + d^{\mathbf{IV}}.$$

We can verify that:

(1) When $0 \leq \Delta \leq \tau$ and $\ell = p + \Delta/(2\tau)$, we have

$$\mathbf{II} \leq \mathbf{I} = \mathbf{III} \geq \mathbf{IV} \text{ and } \min\left\{\gamma - p, \frac{\tau(\gamma - p + 1) + ps}{\tau + 1}, s(p+1)\right\} = \gamma - p;$$

(2) When $\tau \leq \Delta \leq s + s/\tau - 1$ and $\ell = p + (\Delta + 1)/(2\tau + 2)$, we have

$$\mathbf{II} \leq \mathbf{I} = \mathbf{IV} \geq \mathbf{III} \text{ and } \min\left\{\gamma - p, \frac{\tau(\gamma - p + 1) + ps}{\tau + 1}, s(p+1)\right\} = \frac{\tau(\gamma - p + 1) + ps}{\tau + 1};$$

(3) When $\Delta \geq s + s/\tau - 1$ and $\ell = p + (\Delta + 1 - s)/2$, we have

$$\mathbf{I} \leq \mathbf{II} = \mathbf{IV} \geq \mathbf{III} \text{ and } \min\left\{\gamma - p, \frac{\tau(\gamma - p + 1) + ps}{\tau + 1}, s(p+1)\right\} = s(p+1);$$

(4) When $\gamma < 1$ and $\ell = \gamma/2$, we have

$$\mathbf{III} \geq \max\{\mathbf{I}, \mathbf{II}, \mathbf{IV}\}.$$

∎

**Corollary D.17.** *Let $s < 1$ and $\gamma > 0$ be fixed real numbers. Denote $p$ as the integer satisfying $\gamma \in [p(s+1), (p+1)(s+1))$. Suppose one of the following cases holds for $\lambda^\star = d^{-\ell}$ or $\lambda^\star = d^{-\ell} \cdot poly\,(\ln(d))$:*

*(1) $\tau = \infty$, $p \geq 1$, $p(s+1) \leq \gamma < ps + p + s$, $\ell = p + s/2$*

*(2) $\tau = \infty$, $p \geq 1$, $ps + p + s \leq \gamma < ps + p + s + 1$, $\ell = (\gamma + p(1-s))/2$*

*(3) $\tau = \infty$, $\gamma < s$, $\ell = \min\{\gamma, 1, 2\gamma s\}/2$*

*(4) $\tau = \infty$, $s \leq \gamma < s + 1$, $\ell = \min\{(\gamma + (1-s))/2, \gamma(1+s) - s, \gamma/2\}$*

*(5) $\tau < \infty$, $p(s+1) \leq \gamma < ps + p + s$, $\ell = (\gamma + 2\tau p - sp - p)/(2\tau)$*

*(6) $\tau < \infty$, $ps + p + s \leq \gamma < ps + p + s + 1$, $\ell = p + s/(2\tau)$*

*Then we have*

$$\mathcal{M}_{2,\varphi}(\lambda^\star) + \frac{\mathcal{N}_{2,\varphi}(\lambda^\star)}{n} = \Theta\left(d^{-s(p+1)} + \frac{d^p}{n}\right), \tag{104}$$

*or*

$$\mathcal{M}_{2,\varphi}(\lambda^\star) + \frac{\mathcal{N}_{2,\varphi}(\lambda^\star)}{n} = \Theta\left(d^{-s(p+1)} + \frac{d^p}{n}\right) \cdot poly\,(\ln(d)). \tag{105}$$

### D.4.1 Verification of variance conditions

**Lemma D.18** (Verification of variance conditions for inner-product kernels)**.** *Suppose $n \asymp d^\gamma$ and $s \geq 1$, for $\gamma \in [p(s+1), (p+1)(s+1))$. For any given $\ell \geq 0$, if*

$$\lambda \geq \begin{cases} d^{-\ell}\left(1 + \ln^2(d)\mathbf{1}\{\gamma = 2, s = 1\}\right) & p \geq 1, \ 2\ell \leq \max\{2p+1, \gamma - (p+1)(s-1)\} \\ d^{-\ell}\ln^2(d) & p = 0, \gamma \geq 1, \ 2\ell \leq \max\{1, \gamma - (s-1)\} \\ d^{-\ell} & p = 0, \gamma < 1, \ 2\ell \leq \gamma; \end{cases}$$

*then there exists a constant $\epsilon > 0$ only depending on $s$ and $\gamma$, such that $\lambda = \lambda(n, d)$ satisfies*

$$\mathcal{N}_1(\lambda) \cdot n^{\epsilon - 1} \to 0,$$

$$\frac{\mathcal{N}_1^2(\lambda)}{n\mathcal{N}_{2,\varphi}(\lambda)} \cdot \ln(n)(\ln \lambda^{-1})^2 \to 0.$$

*Proof.* From Lemma 21 in Zhang et al. (2024), we have $\mathcal{N}_1(\lambda) \asymp \lambda^{-1}$. When $p = 0$, we have $\gamma - \ell > 0$. When $p \geq 1$, we have $\gamma - p - 1/2 \geq ps - 1/2 > 0$. Therefore, there exists a constant $\epsilon > 0$ only depending on $s$ and $\gamma$, such that we have

$$\mathcal{N}_1(\lambda) \cdot n^{\epsilon - 1} \to 0.$$

Denote $q := \lfloor \ell \rfloor$. From Lemma D.14, we further have $\mathcal{N}_{2,\varphi}(\lambda) = \Omega\left(d^q + \lambda^{-2}d^{-q-1}\right)$. Hence, we have

$$\frac{\mathcal{N}_1^2(\lambda)}{n\mathcal{N}_{2,\varphi}(\lambda)} \cdot \ln(n)(\ln \lambda^{-1})^2 = O\left(\frac{(\ln(d))^3}{n(\lambda^2 d^q + d^{-q-1})}\right).$$

Denote $\Delta := \frac{(\ln(d))^3}{n\lambda^2 d^q}$, $\Delta' := \frac{(\ln(d))^3}{d^{\gamma - q - 1}}$, then when $\Delta = o(1)$ or $\Delta' = o(1)$, we have:

$$\frac{\mathcal{N}_1^2(\lambda)}{n\mathcal{N}_{2,\varphi}(\lambda)} \cdot \ln(n)(\ln \lambda^{-1})^2 \to 0.$$

Now we show that $\Delta = o(1)$:

- When $p \geq 3$ and $p = 2, s > 1$, since $\gamma - 2\ell + q \geq (\gamma - \ell - 1) + (q + 1 - \ell) > 0$, we have $\Delta = o(1)$.

- When $p = 2, s = 1$, since $2\ell - q < \ell + 1 < 4 \leq \gamma$, we have $\Delta = o(1)$.

- When $p = 2, s = 1$, since $2\ell - q < \ell + 1 < 4 \leq \gamma$, we have $\Delta = o(1)$.

- When $p = 1, \gamma > 2s + 1$, since $\ell < 2$ and hence $2\ell - q < 3 \leq \gamma$, we have $\Delta = o(1)$.

- When $p = 1, s > 1, \gamma \leq 2s + 1$, or $p = 1, s = 1, \gamma > 2$, since $2\ell - q \leq 2 < \gamma$, we have $\Delta = o(1)$.

- When $p = 1, s = 1, \gamma = 2$, since $2\ell - q \leq 2 \leq \gamma$, we have $\Delta = O((\ln(d))^{-1})$.

- When $p = 0$, since $\gamma - 2\ell \geq 0$, we have $\Delta = O((\ln(d))^{-1})$.

■

**Lemma D.19** (Verification of variance conditions for inner-product kernels: saturation case). *Suppose* $\tau < s \leq 2\tau$. *Suppose* $n \asymp d^\gamma$, *for* $\gamma \in [p(s+1) + \tau, p(s+1) + s + s/\tau - 1]$. *For any given* $\ell \geq 0$, *if*

$$\lambda \geq d^{-\ell}, \quad \ell \leq p + (\gamma - p(s+1) + 1)/(2\tau + 2);$$

*then there exists a constant $\epsilon > 0$ only depending on $s$ and $\gamma$, such that $\lambda = \lambda(n,d)$ satisfies*

$$\mathcal{N}_1(\lambda) \cdot n^{\epsilon - 1} \to 0,$$

$$\frac{\mathcal{N}_1^2(\lambda)}{n\mathcal{N}_{2,\varphi}(\lambda)} \cdot \ln(n)(\ln \lambda^{-1})^2 \to 0.$$

*Proof.* From Lemma 21 in Zhang et al. (2024), we have $\mathcal{N}_1(\lambda) \asymp \lambda^{-1}$. Notice that we have

$$2(\tau + 1)(\gamma - p) \geq \begin{cases} ps - 1 & p \geq 1 \\ 2\tau^2 + (\tau - 1) & p = 0 \end{cases} > 0;$$

Therefore, there exists a constant $\epsilon > 0$ only depending on $\tau$, $s$, and $\gamma$, such that we have

$$\mathcal{N}_1(\lambda) \cdot n^{\epsilon - 1} \to 0.$$

Denote $q := \lfloor \ell \rfloor$. From Lemma D.14, we further have $\mathcal{N}_{2,\varphi}(\lambda) = \Omega\left(d^q + \lambda^{-2}d^{-q-1}\right)$. Hence, we have

$$\frac{\mathcal{N}_1^2(\lambda)}{n\mathcal{N}_{2,\varphi}(\lambda)} \cdot \ln(n)(\ln \lambda^{-1})^2 = O\left(\frac{(\ln(d))^3}{n(\lambda^2 d^q + d^{-q-1})}\right)$$

$$= O\left(\frac{(\ln(d))^3}{n\lambda^2 d^q}\right) + O\left(\frac{(\ln(d))^3}{d^{\gamma - q - 1}}\right).$$

Denote $\Delta := \frac{(\ln(d))^3}{n\lambda^2 d^q}$, $\Delta' := \frac{(\ln(d))^3}{d^{\gamma - q - 1}}$. We have:

- When $p \geq 1$, since

$$2(\tau + 1)[\gamma - 2\ell + q] \geq 2(\tau + 1)[(\gamma - \ell - 1) + (q + 1 - \ell)]$$

$$\geq \begin{cases} ps - 2 & p \geq 2 \\ 2(\tau + 1)(\tau - 1) + 2[\tau s + s - 1] & p = 1 \end{cases}$$

$$> 0,$$

  we have $\Delta = o(1)$.

- When $p = 0$, since $\gamma > 1$, we have $\Delta' = o(1)$.

■

**Lemma D.20** (Verification of variance conditions for inner-product kernels: misspecified case).
*Suppose $n \asymp d^\gamma$ and $0 < s < 1$, for $\gamma \in [p(s+1), (p+1)(s+1))$. For any given $\ell \geq 0$, if*

$$\lambda \geq \begin{cases} d^{-\ell} & p \geq 1, \ 2\ell \leq \max\{2p+s, \gamma+p(1-s)\} \\ d^{-\ell} & p = 0, \gamma > s, \ 2\ell \leq \gamma \\ d^{-\ell}\ln(d) & p = 0, \gamma \leq s, \ 2\ell \leq \gamma; \end{cases}$$

*then there exists a constant $\epsilon > 0$ only depending on $s$ and $\gamma$, such that $\lambda = \lambda(n, d)$ satisfies*

$$\mathcal{N}_1(\lambda) \cdot n^{\epsilon-1} \to 0,$$

$$\frac{\mathcal{N}_1^2(\lambda)}{n\mathcal{N}_{2,\varphi}(\lambda)} \cdot \ln(n)(\ln \lambda^{-1})^2 \to 0.$$

*Proof.* When $p \geq 1$, it is a direct result of step 2 (the verification of the second condition in (146) of Zhang et al. (2024)) in the proof of Theorem 3 in Zhang et al. (2024) and the fact that $\mathcal{N}_{2,\varphi}(\lambda) \asymp \mathcal{N}_2(\lambda)$.

When $p = 0$, a similar argument as the proof for Lemma D.18 give the desired results. ∎

### D.4.2 Verification of bias conditions

**Lemma D.21** (Verification of bias conditions). *Suppose $1 \leq s \leq \tau$. Suppose $n \asymp d^\gamma$, for $\gamma \in [p(s+1), (p+1)(s+1))$. For any given $\ell \geq 0$, if*

$$\lambda \geq \begin{cases} d^{-\ell}\left(1+\ln^2(d)\mathbf{1}\{\gamma=2, s=1\}\right) & p \geq 1, \ 2\ell \leq \max\{2p+1, \gamma-(p+1)(s-1)\} \\ d^{-\ell}\ln^2(d) & \gamma \in [1, s+1), \ 2\ell \leq \max\{1, \gamma-(s-1)\} \\ d^{-\ell} & \gamma \in (0,1), \ 2\ell \leq \gamma; \end{cases}$$

*then there exists a constant $\epsilon > 0$ only depending on $s$ and $\gamma$, such that $\lambda = \lambda(n, d)$ satisfies*

$$\frac{\mathcal{N}_1(\lambda)\mathcal{M}_{1,\varphi}^2(\lambda)}{n^2} \ll \left(\mathcal{M}_{2,\varphi}(\lambda) + \frac{\sigma^2}{n}\mathcal{N}_{2,\varphi}(\lambda)\right),$$

$$\frac{\mathcal{N}_1(\lambda)}{n}\ln(n)(\ln \lambda^{-1})^2 \cdot \sum_{j=1}^\infty \frac{\lambda^2 \lambda_i \varphi_\lambda^2(\lambda_i)}{\lambda+\lambda_i}f_i^2 \ll \left(\mathcal{M}_{2,\varphi}(\lambda) + \frac{\sigma^2}{n}\mathcal{N}_{2,\varphi}(\lambda)\right).$$

(106)

*Proof.* When $1 \leq s \leq \tau$, from Lemma D.14, we have

$$n\left(\mathcal{M}_{2,\varphi}(\lambda) + \frac{\sigma^2}{n}\mathcal{N}_{2,\varphi}(\lambda)\right) = \Omega\left(d^{\gamma-s(q+1)} + d^q\right)$$

$$\frac{\mathcal{N}_1(\lambda)\mathcal{M}_{1,\varphi}^2(\lambda)}{n} = O\left(\lambda^{2(s-1)}d^{-\gamma+qs} + \lambda^{-1}d^{-\gamma-(q+1)(s-1)}\right)$$

$$\mathcal{N}_1(\lambda)\ln(n)(\ln \lambda^{-1})^2 \cdot \sum_{j=1}^\infty \frac{(\lambda)^2\lambda_i\varphi_\lambda^2(\lambda_i)}{\lambda+\lambda_i}f_i^2 = O\left((\ln(d))^3\right) \cdot O\left(\lambda d^{\max\{q(2-s),0\}} + \lambda^{-1}d^{-s(q+1)}\right),$$

Denote $\mathbf{I} = \lambda^{2(s-1)}d^{-\gamma+qs}$, $\mathbf{II} = \lambda^{-1}d^{-\gamma-(q+1)(s-1)}$, $\mathbf{III} = \lambda d^{\max\{q(2-s),0\}}(\ln(d))^3$, and $\mathbf{IV} = \lambda^{-1}d^{-s(q+1)}(\ln(d))^3$.

For any $p \geq 0$ and any $s \geq 1$:

- From Lemma D.18, we have $\mathbf{IV} \ll d^{\gamma-s(q+1)}$.

- When $\gamma \geq 1$, we have $\gamma \geq p+1$, and hence $\mathbf{II} \ll \mathbf{IV} \ll d^{\gamma-s(q+1)}$; when $\gamma < 1$, we have $\mathbf{II} \ll d^q$ with $q = 0$.

- When $p \geq 1$ or $\gamma \in (s, s+1)$, since $-\ell s + qs \leq 0$, we have $\mathbf{I}/d^{\gamma-s(q+1)} = O(d^{-2(\gamma-\ell-s/2)}) \ll 1$; when $\gamma \in (0, s]$, we have $\mathbf{I} = O(d^{-2s\ell+2\ell-\gamma}) = O(d^{-2s\ell}) \ll d^q$ with $q = 0$.

- When $s \geq 2$, we have $\mathbf{III} \ll d^q$; when $s < 2$ and $p = 0$, we have $\mathbf{III} \ll d^q$; when $s < 2$ and $p \geq 1$ and $q \geq 1$, since $\gamma - \ell - s > \min\{(s+1)q - \ell, ps - 1/2\} > 0$, we have $\mathbf{III}/d^{\gamma - s(q+1)} = d^{-(\gamma - \ell - s) - 2(\ell - q)} \ll 1$ or $\mathbf{III}/d^q \ll 1$; when $s < 2$ and $p \geq 1$ and $q = 0$, we have $\mathbf{III} \ll d^q$.

Combining all these, we get the desired results. $\blacksquare$

**Lemma D.22.** *[Verification of bias conditions: saturation case] Suppose $\tau < s \leq 2\tau$. Suppose $n \asymp d^\gamma$, for $\gamma \in [p(s+1), (p+1)(s+1))$. For any given $\ell \geq 0$, if*

$$\lambda \geq \begin{cases} d^{-\ell} & p \geq 1, \ \ell \leq \max\{\ell_1, \ell_2, \ell_3\} \\ d^{-\ell} \ln^2(d) & \gamma \in [1, s+1), \ \ell \leq \max\{\ell_1, \ell_2, \ell_3\} \\ d^{-\ell} & \gamma \in (0, 1), \ 2\ell \leq \gamma, \end{cases}$$

*where $\tau$, $\Delta$, $\ell_1$, $\ell_2$, and $\ell_3$ are given in Lemma D.16; then there exists a constant $\epsilon > 0$ only depending on $s$ and $\gamma$, such that $\lambda = \lambda(n, d)$ satisfies*

$$\frac{\mathcal{N}_1(\lambda)\mathcal{M}_{1,\varphi}^2(\lambda)}{n^2} \ll \left( \mathcal{M}_{2,\varphi}(\lambda) + \frac{\sigma^2}{n}\mathcal{N}_{2,\varphi}(\lambda) \right),$$

$$\frac{\mathcal{N}_1(\lambda)}{n} \ln(n)(\ln \lambda^{-1})^2 \cdot \sum_{j=1}^{\infty} \frac{\lambda^2 \lambda_i \varphi_\lambda^2(\lambda_i)}{\lambda + \lambda_i} f_i^2 \ll \left( \mathcal{M}_{2,\varphi}(\lambda) + \frac{\sigma^2}{n}\mathcal{N}_{2,\varphi}(\lambda) \right). \tag{107}$$

*Proof.* When $\tau < s \leq 2\tau$, from Lemma D.14, we have

$$n\left( \mathcal{M}_{2,\varphi}(\lambda) + \frac{\sigma^2}{n}\mathcal{N}_{2,\varphi}(\lambda) \right) = \Omega\left( \lambda^{2\tau} d^{q(2\tau - s)} + d^{\gamma - s(q+1)} + d^q \right)$$

$$\frac{\mathcal{N}_1(\lambda)\mathcal{M}_{1,\varphi}^2(\lambda)}{n} = O\left( \lambda^{2(\tau - 1)} d^{-\gamma + q(2\tau - s)} + \lambda^{-1} d^{-\gamma - (q+1)(s-1)} \right)$$

$$\mathcal{N}_1(\lambda)\ln(n)(\ln \lambda^{-1})^2 \cdot \sum_{j=1}^{\infty} \frac{(\lambda)^2 \lambda_i \varphi_\lambda^2(\lambda_i)}{\lambda + \lambda_i} f_i^2 = O\left( (\ln(d))^3 \right) \cdot O\left( \lambda d^{\max\{q(2-s),0\}} + \lambda^{-1} d^{-s(q+1)} \right).$$

Denote $\mathbf{I}' = \lambda^{2(\tau - 1)} d^{-\gamma + q(2\tau - s)}$, $\mathbf{II} = \lambda^{-1} d^{-\gamma - (q+1)(s-1)}$, $\mathbf{III} = \lambda d^{\max\{q(2-s),0\}}(\ln(d))^3$, and $\mathbf{IV} = \lambda^{-1} d^{-s(q+1)}(\ln(d))^3$.

For any $p \geq 0$ and any $1 \leq \tau < s \leq 2\tau$:

- From Lemma D.18 and Lemma D.19, since $\mathcal{N}_1(\lambda) \cdot n^{\epsilon - 1} \to 0$, we have $\mathbf{IV} \ll d^{\gamma - s(q+1)}$.

- When $\gamma \geq 1$, we have $\gamma \geq p + 1$, and hence $\mathbf{II} \ll \mathbf{IV} \ll d^{\gamma - s(q+1)}$; when $\gamma < 1$, we have $\mathbf{II} \ll d^q$ with $q = 0$.

- When $p \geq 1$, since $-\ell\tau + q\tau \leq 0$ and

$$\gamma - \ell - s/2$$
$$\geq \max\left\{ \frac{s(2p-1)}{2}, \frac{(2\tau+1)(\tau+ps) - (\tau+1)s + ps - 1}{2(\tau + 1}, ps + \frac{s(\tau+1)}{2\tau} - 1 \right\}$$
$$> 0,$$

  we have $\mathbf{I}'/d^{\gamma - s(q+1)} \ll 1$; when $p = 0$, we have $\mathbf{I}' = O(d^{-2\tau\ell + 2\ell - \gamma}) \ll d^q$ with $q = 0$.

- When $\gamma - p - ps \in [0, \tau] \cup [s + s/\tau - 1, s + 1]$, we have $\ell \leq \max\{\ell_1, \ell_3\}$. Similar to the proof in Lemma D.21, we can show that $\mathbf{III} \ll d^{\gamma - s(q+1)} + d^q$.

- Finally, consider the case $\gamma - p - ps \in [\tau, s + s/\tau - 1]$. When $s \geq 2$, we have $\mathbf{III} \ll d^q$; when $s < 2$, since $s > 1$, we have $\mathbf{III}/d^q = \lambda d^{-q(s-1)} \ll 0$.

Combining all these, we get the desired results. $\blacksquare$

**Lemma D.23** (Verification of bias conditions: misspecified case). *Suppose $0 < s < 1$. Suppose $n \asymp d^\gamma$, for $\gamma \in [p(s+1), (p+1)(s+1))$. Suppose one of the following holds:*

*(1) $\tau = \infty$.*

*(2) $s > 1/(2\tau)$,*

*(3) $\gamma > ((2\tau+1)s)/(2\tau(1+s))$.*

*Suppose one of the following cases holds for $\lambda = d^{-\ell}$ or $\lambda = d^{-\ell}(\ln(d))^2$:*

*(1) $\tau = \infty$, $p(s+1) \leq \gamma \leq ps + p + s$,*
*$\ell \in [p, p + \min\{1/2, \gamma s\}]$*

*(2) $\tau = \infty$, $ps + p + s < \gamma < ps + p + s + 1$,*
*$\ell \in [p, \min\{(\gamma - (p+1)(s-1))/2, \gamma(1+s) - s(p+1)\}]$*

*(3) $\tau < \infty$, $p(s+1) \leq \gamma \leq ps + p + s$,*
*$\ell = (\gamma + 2\tau p - sp - p)/(2\tau)$*

*(4) $\tau < \infty$, $ps + p + s < \gamma < ps + p + s + 1$,*
*$\ell = p + s/(2\tau)$.*

*then there exists a constant $\epsilon > 0$ only depending on $s$ and $\gamma$, such that $\lambda = \lambda(n, d)$ satisfies*

$$\frac{\mathcal{N}_1(\lambda)}{n} \ln(n)(\ln \lambda^{-1})^2 \cdot \sum_{j=1}^{\infty} \frac{\lambda^2 \lambda_i \varphi_\lambda^2(\lambda_i)}{\lambda + \lambda_i} f_i^2 \ll \left( \mathcal{M}_{2,\varphi}(\lambda) + \frac{\sigma^2}{n} \mathcal{N}_{2,\varphi}(\lambda) \right);$$

$$n^{-2} \mathcal{N}_1(\lambda) \left( \|f_\lambda\|_{L^\infty} + n^{\frac{1-s}{2}+\epsilon} \right)^2 = o \left( \mathcal{M}_{2,\varphi}(\lambda) + \frac{\sigma^2}{n} \mathcal{N}_{2,\varphi}(\lambda) \right).$$

*Proof.* When $0 < s < 1$, from Lemma D.14, we have

$$n \left( \mathcal{M}_{2,\varphi}(\lambda) + \frac{\sigma^2}{n} \mathcal{N}_{2,\varphi}(\lambda) \right) = \Omega \left( d^{\gamma - s(p+1)} + d^p \right)$$

$$n^{-1} \mathcal{N}_1(\lambda) n^{1-s} = O \left( \lambda^{-1} d^{-\gamma s} \right)$$

$$\mathcal{N}_1(\lambda) \ln(n)(\ln \lambda^{-1})^2 \cdot \sum_{j=1}^{\infty} \frac{(\lambda)^2 \lambda_i \varphi_\lambda^2(\lambda_i)}{\lambda + \lambda_i} f_i^2 = O \left( (\ln(d))^3 \right) \cdot O \left( \lambda d^{\max\{p(2-s), 0\}} + \lambda^{-1} d^{-s(p+1)} \right),$$

and the convergence rate of $\|f_\lambda\|_{L^\infty}$ can be attained similar to Lemma 25 in Zhang et al. (2024). Since $\tau \geq 1$, similar to the proof of Theorem 3 of Zhang et al. (2024), when $1/2 < s < 1$, we have

$$n^{-2} \mathcal{N}_1(\lambda) \left( \|f_\lambda\|_{L^\infty} + n^{\frac{1-s}{2}+\epsilon} \right)^2 = o \left( \mathcal{M}_{2,\varphi}(\lambda) + \frac{\sigma^2}{n} \mathcal{N}_{2,\varphi}(\lambda) \right),$$

and when $s \leq 1/2$, we have

$$n^{-2} \mathcal{N}_1(\lambda) \|f_\lambda\|_{L^\infty}^2 = o \left( \mathcal{M}_{2,\varphi}(\lambda) + \frac{\sigma^2}{n} \mathcal{N}_{2,\varphi}(\lambda) \right).$$

Denote $\mathbf{I} = \lambda^{-1} d^{-\gamma s}$, $\mathbf{II} = \lambda d^{p(2-s)}(\ln(d))^3$, and $\mathbf{III} = \lambda^{-1} d^{-s(p+1)}(\ln(d))^3$.

For any $p \geq 0$ and any $0 < s < 1$:

- From Lemma D.20, we have $\mathbf{III} \ll d^{\gamma - s(p+1)}$,

- When $\gamma \leq ps + p + s$, we can show $\mathbf{I} \ll d^p$ when: (1) $p \geq 1$, or (2) $p = 0$ and $s > 1/(2\tau) > 0$, or (3) $\tau = \infty$,

- When $\gamma > ps + p + s$, we can show $\mathbf{I} \ll d^{\gamma - s(p+1)}$ holds if and only if $\tau = \infty$ or

$$\gamma > \frac{(2\tau+1)s + 2\tau(1+s)p}{2\tau(1+s)}, \quad \tau = \tau < \infty;$$

and the above inequality holds when (1) $p > 0$ or (2) $p = 0, s > 1/(2\tau) > 0$, or (3) $p = 0, \gamma > ((2\tau + 1)s)/(2\tau(1 + s))$;

- When $\gamma \leq ps + p + s$, since $\ell \geq p > p - ps$, we have $\mathbf{II} \ll d^p$;

- When $\gamma > ps + p + s$, since $\ell \geq p > p - ps$, we have $\mathbf{II} \ll d^{\gamma - s(p+1)}$.

Combining all these, we get the desired results. ∎

### D.5 Final proof of Theorem 4.1 and Theorem 4.2

For each case, the proof can be done in the following steps:

(i) When $\lambda \geq \lambda^\star$ and $s \leq 2\tau$, where the definition of the balanced parameter $\lambda^\star$ can be found in Corollary D.15 and Corollary D.16, we have

$$\mathcal{M}_{2,\varphi}(\lambda^\star) + \frac{\sigma^2}{n}\mathcal{N}_{2,\varphi}(\lambda^\star) = \Theta_\mathbb{P}\left(d^{-\beta^\star}\right) \cdot \mathrm{poly}\left(\ln(d)\right)$$

$$\mathcal{M}_{2,\varphi}(\lambda) + \frac{\sigma^2}{n}\mathcal{N}_{2,\varphi}(\lambda) = \Theta_\mathbb{P}\left(d^{-\beta}\right) \cdot \mathrm{poly}\left(\ln(d)\right),$$

where $d^{-\beta^\star}$ is the desired convergence rate given in Theorem 4.1 or Theorem 4.2 and $\beta \leq \beta^*$. Similarly, when $s > 2\tau$, by taking $s = 2\tau$ in Corollary D.16, we also have

$$\mathcal{M}_{2,\varphi}(\lambda^\star) + \frac{\sigma^2}{n}\mathcal{N}_{2,\varphi}(\lambda^\star) = \Theta_\mathbb{P}\left(d^{-\beta^\star}\right) \cdot \mathrm{poly}\left(\ln(d)\right)$$

$$\mathcal{M}_{2,\varphi}(\lambda) + \frac{\sigma^2}{n}\mathcal{N}_{2,\varphi}(\lambda) = \Theta_\mathbb{P}\left(d^{-\beta}\right) \cdot \mathrm{poly}\left(\ln(d)\right).$$

(ii) When $\lambda \geq \lambda^\star$, from Lemma D.14, Lemma D.18, Lemma D.19, Lemma D.20, Lemma D.21, Lemma D.22, and Lemma D.23, we know that conditions in Theorem D.5, Theorem D.9, and Theorem D.10 are satisfied. Therefore, we have

$$\mathbb{E}\left(\left\|\hat{f}_{\lambda^\star} - f_\star\right\|_{L^2}^2 \ \Big| \ \boldsymbol{X}\right) = \Theta_\mathbb{P}\left(d^{-\beta^\star}\right) \cdot \mathrm{poly}\left(\ln(d)\right)$$

$$\mathbb{E}\left(\left\|\hat{f}_{\lambda} - f_\star\right\|_{L^2}^2 \ \Big| \ \boldsymbol{X}\right) = \Theta_\mathbb{P}\left(d^{-\beta}\right) \cdot \mathrm{poly}\left(\ln(d)\right).$$

(iii) Finally, when $s > \tau$, we can further show that: the convergence rates of the generalization error can not be faster than above for any choice of regularization parameter $\lambda = \lambda(d, n) \to 0$. Notice that, when $s \geq 1$, for any $\lambda < \lambda^\star$, from the monotonicity of $\mathbf{Var}(\lambda)$ (see, e.g., Li et al. (2024); Zhang et al. (2024)), we have

$$\mathbb{E}\left[\left\|\hat{f}_{\lambda} - f_\star\right\|_{L^2}^2 \ \Big| \ \boldsymbol{X}\right] \geq \mathbf{Var}(\lambda) \geq \mathbf{Var}(\lambda^\star) \asymp \mathbb{E}\left[\left\|\hat{f}_{\lambda^\star} - f_\star\right\|_{L^2}^2 \ \Big| \ \boldsymbol{X}\right],$$

and hence

$$\mathbb{E}\left(\left\|\hat{f}_{\lambda} - f_\star\right\|_{L^2}^2 \ \Big| \ \boldsymbol{X}\right) = \Omega_\mathbb{P}\left(d^{-\beta^\star}\right) \cdot \mathrm{poly}\left(\ln(d)\right).$$

## E Auxiliary lemmas

**Proposition E.1.** *For any analytic filter function $\varphi_\lambda$, we have $(z + \lambda)\varphi_\lambda(z) \leq 4$ and $(z + \lambda)\psi_\lambda(z) \leq 4\lambda$.*

*Proof.* From (28), we have $(z + \lambda)\varphi_\lambda(z) \leq 2\max\{z, \lambda\}\varphi_\lambda(z) \leq 2\max\{1, \mathfrak{C}_4\} \leq 4$. From (27), we have $(z + \lambda)\psi_\lambda(z) \leq 2\max\{z, \lambda\}\psi_\lambda(z) \leq 2\max\{\mathfrak{C}_2, 1\}\lambda \leq 4\lambda$. ∎

**Lemma E.2.** *Let $\varphi_\lambda$ be an analytic filter function defined in Definition C.1. Then, for any $s \in [0, 1]$, we have*

$$\sup_{z \in [0, \kappa^2]} \varphi_\lambda(z)z^s \leq 4\lambda^{s-1}.$$

*Proof.* For any $z \in [0, \kappa^2]$, from Proposition E.1, we have $(z + \lambda)\varphi_\lambda(z) \leq 4$. Therefore, from Proposition B.3 in Li et al. (2024), we have

$$\varphi_\lambda(z)z^s \leq \frac{4z^s}{z + \lambda} \leq 4\lambda^{s-1}.$$

∎

**Lemma E.3.** *Let $\psi_\lambda$ be defined in Definition C.1. Then, for any $s > 2\tau$, we have*

$$\sup_{z \in [0, \kappa^2]} z^s \psi_\lambda^2(z) \leq \mathfrak{C}_2^2 \kappa^{2(s-2\tau)} \lambda^{2\tau}.$$

*Proof.* For any $z$, we have

$$\psi_\lambda(z) \leq \mathfrak{C}_2(z/\lambda)^{-\tau} \mathbf{1}\{z > \lambda\} + \mathbf{1}\{z \leq \lambda\} \leq \mathfrak{C}_2(z/\lambda)^{-\tau},$$

hence

$$z^s \psi_\lambda^2(z) \leq \mathfrak{C}_2^2 z^s z^{-2\tau} \lambda^{2\tau} \leq \mathfrak{C}_2^2 \kappa^{2(s-2\tau)} \lambda^{2\tau}.$$

∎

## E.1 Analytic functional calculus

The "analytic functional argument" introduced in Li et al. (2024) is vital in our proof for Theorem 4.1. For readers' convenience, we collect some of the main ingredients here, see Li et al. (2024) for details.

*Definition* E.4. Let $A$ be a linear operator on a Banach space $X$. The *resolvent set* $\rho(A)$ is given by

$$\rho(A) := \{\lambda \in \mathbb{C} \mid A - \lambda \text{ is invertible}\},$$

and we denote $R_A(\lambda) := (A - \lambda)^{-1}$. The spectrum of $A$ is defined by

$$\sigma(A) := \mathbb{C} \backslash \rho(A).$$

A simple but key ingredient in the analytic functional calculus is the following *resolvent identity*:

$$R_A(\lambda) - R_B(\lambda) = R_A(\lambda)(B - A)R_B(\lambda) = R_B(\lambda)(B - A)R_A(\lambda). \tag{108}$$

The resolvent allows us to define the value of $f(A)$ in analog to the form of Cauchy integral formula, where $A$ is an operator and $f$ is an analytic function. The following two propositions are well-known results on operator calculus.

**Proposition E.5** (analytic functional calculus)**.** *Let $A$ be an operator on a Hilbert space $H$ and $f$ be an analytic function defined on $D_f \subset \mathbb{C}$. Let $\Gamma$ be a contour contained in $D_f$ surrounding $\sigma(A)$. Then,*

$$f(A) = \frac{1}{2\pi i} \oint_\Gamma f(z)(z - A)^{-1} \mathrm{d}z = -\frac{1}{2\pi i} \oint_\Gamma f(z)R_A(z)\mathrm{d}z, \tag{109}$$

*and it is independent of the choice of $\Gamma$.*

Now, let $\Gamma$ be a contour contained in $D_f$ surrounding both $\sigma(A)$ and $\sigma(B)$. Using (108), we get

$$f(A) - f(B) = -\frac{1}{2\pi i} \oint_\Gamma f(z)\left[R_A(z) - R_B(z)\right] \mathrm{d}z = \frac{1}{2\pi i} \oint_\Gamma R_B(z)(A - B)R_A(z)f(z)\mathrm{d}z. \tag{110}$$

**Proposition E.6** (Spectral mapping theorem)**.** *Let $A$ be a bounded self-adjoint operator and $f$ be a continuous function on $\sigma(A)$. Then*

$$\sigma(f(A)) = \{f(\lambda) \mid \lambda \in \sigma(A)\}. \tag{111}$$

*Consequently, $\|f(A)\| = \sup_{\lambda \in \sigma(A)} |f(\lambda)| \leq \|f\|_\infty$.*

Let us define the contour $\Gamma_\lambda$ considered in Li et al. (2024) by

$$
\begin{aligned}
\Gamma_\lambda &= \Gamma_{\lambda,1} \cup \Gamma_{\lambda,2} \cup \Gamma_{\lambda,3} \\
\Gamma_{\lambda,1} &= \left\{ x \pm (x+\eta)i \in \mathbb{C} \mid x \in [-\eta, 0] \right\} \\
\Gamma_{\lambda,2} &= \left\{ x \pm (x+\eta)i \in \mathbb{C} \mid x \in (0, \kappa^2) \right\} \\
\Gamma_{\lambda,3} &= \left\{ z \in \mathbb{C} \mid \left| z - \kappa^2 \right| = \kappa^2 + \eta, \ \mathrm{Re}(z) \geq \kappa^2 \right\},
\end{aligned}
\tag{112}
$$

where $\eta = \lambda/2$. Then, since $T$ and $T_X$ are positive self-adjoint operators with $\|T\|, \|T_X\| \leq \kappa^2$, we have $\sigma(T), \sigma(T_X) \subset [0, \kappa^2]$. Therefore, $\Gamma_\lambda$ is indeed a contour satisfying the requirement in Proposition E.5.

**Proposition E.7.** *Suppose that (45) in Assumption 3 holds. Suppose that $\lambda = \lambda(n, d)$ satisfies $v := \frac{\mathcal{N}_1(\lambda)}{n} \ln n = o(1)$. Then for any fixed $\delta \in (0, 1)$, when $n$ is sufficiently large, with probability at least $1 - \delta$, we have*

$$
\| T_\lambda^{-\frac{1}{2}} (T - T_X) T_\lambda^{-\frac{1}{2}} \| \leq \sqrt{v}.
$$

$$
\left\| T_\lambda^{-\frac{1}{2}} T_{X\lambda}^{\frac{1}{2}} \right\|^2 \leq 2
\tag{113}
$$

$$
\left\| T_\lambda^{\frac{1}{2}} T_{X\lambda}^{-\frac{1}{2}} \right\|^2 \leq 3.
\tag{114}
$$

*Proof.* These inequalities are direct results of (56), (58), and (59) in Zhang et al. (2024). ∎

**Proposition E.8** (Restate Proposition 4.13 in Li et al. (2024) with only the constant modified)**.** *When (113) holds, there is an absolute constant that for any $z \in \Gamma_\lambda$,*

$$
\begin{aligned}
\| T_\lambda^{\frac{1}{2}} (T - z)^{-1} T_\lambda^{\frac{1}{2}} \| &\leq C \\
\| T_\lambda^{\frac{1}{2}} (T_X - z)^{-1} T_\lambda^{\frac{1}{2}} \| &\leq \sqrt{6} C.
\end{aligned}
\tag{115}
$$

