# OpenReview forum: "On the Saturation Effects of Spectral Algorithms in Large Dimensions"
_NeurIPS.cc/2024/Conference — NeurIPS 2024 poster_

### Official Review · Reviewer_XV4b · 2024-07-04

**Soundness:** 4
**Presentation:** 4
**Contribution:** 3
**Rating:** 7
**Confidence:** 4

**Summary:**

This paper concerns the convergence rate of spectral methods, particularly kernel ridge regression (KRR) and kernel gradient flow (KGF), in large-dimensional settings where the sample size $n$ is of the same magnitude as a power $\gamma$ of the input dimension $ d $, i.e., $ n \asymp d^\gamma $. It reveals a new phenomenon of the saturation effect in large dimensions, which is different from its fixed-dimensional counterpart. Specifically, it shows that in large dimensions, KRR still suffers from the saturation effect while KGF does not. The key techniques involve the use of analytic filter functions to characterize the regressors from different spectral algorithms, followed by standard concentration results on the bias-variance decomposition of the excess risk.

**Strengths:**

This paper offers solid theoretical results which improve over previous literature or reveal novel phenomenon. It offers important insights on spectral algorithms in large input dimensional setting, for example the interpolation (with different qualification $\tau$) of the learning rate between KRR and KGF, the exact description of the phenomena *periodic plateau behaviour* and *polynomial approximation barrier*.
This paper also offers numerical validations on their claim.

**Weaknesses:**

There is no major weakness spotted in this paper.

**Questions:**

I appreciate the result of this paper and hence look for any possible extension from the current result.

This paper focuses on dot-product kernels $K=\Phi(\langle \cdot, \cdot \rangle)$ with inputs distributed uniformly on a hypersphere and with polynomial spectral decay.

1. By [Belkin2018] and [Haas2024], it simply seems the function $\Phi$ cannot be smooth but at least 1-differentiable, say $\Phi$ is induced by the ReLU-NTK. If $\Phi$ is smooth, for example $K$ is the Gaussian kernel, then the spectral decay is exponential. Can the result in this paper extend to this case? If yes, where is the adaptation? If no or not obvious, what would be the main technical difficulties?

2. In realistic setting, uniform input distribution on a hypersphere is too restrictive. Could one relax the condition to the distributions which have support on the whole sphere instead? I recall Lemma Lemma F.9 in [Haas2024] stating the spectral decay is still polynomial in this case. Could one extend the analysis in this case?

Also, I have some technical questions concerning the appendix.

3. In Eq (62) in Lemma D.7, the left hand side (LHS) should be independent to the noise, but why is there a term with $\sigma^2$ on the right hand side (RHS)?

4. Less like a question but more like a comment: I think that there is a typo in Eq (42): it should be $... \phi\_j^2(x)\leq ...$ instead of $... \phi\_i^2(x)\leq ...$. Also, is Eq (44) actually redundant as a special case of Eq (43) given the notation mentioned in line 602 - 603?

Reference:
- Haas, Moritz, et al. "Mind the spikes: Benign overfitting of kernels and neural networks in fixed dimension." Advances in Neural Information Processing Systems 36 (2024).
- Belkin, Mikhail. "Approximation beats concentration? An approximation view on inference with smooth radial kernels." Conference On Learning Theory. PMLR, 2018.

**Limitations:**

All assumptions and conditions are stated clearly in the paper.

---

> ### Author Rebuttal · Authors · 2024-08-04
>
> We sincerely thank you for taking the time to read our paper and for providing valuable feedback. We are pleased to see that you not only accurately described the contributions of our work but also gave it high praise. We would like to address the questions and comments you raised regarding possible extensions of our research.
>
> **Author's response to Question 1:**
>
> Thank you for suggesting checking our assumptions on two specific kernels. After a careful check on the paper [Belkin2018] and [Haas2024], we find that $\Phi$ for both ReLU-NTK and the Gaussian kernel are analytic, with all coefficients $a_j>0$, thus satisfying Assumption 1. Therefore, the results of this paper apply to both ReLU-NTK and the Gaussian kernel. Please let us clarify in detail:
>
> - From the definition of ReLU-NTK in [Bietti2021], $\Phi$ for ReLU-NTK is analytic (see, e.g., page 6 in [Bietti2021]). Moreover, Corollary 3 of [Bietti2021] as well as Lemma B.2 of [Haas2024] both imply that we have all coefficients $a_j>0$. Notice that the proof for the positive-definiteness of the kernel in the above studies relies on [Gneiting2013], hence they also require that $\Phi$ is analytic.
>
> - From the definition of the Gaussian kernel defined on the sphere, we can show that $\Phi$ for the Gaussian kernel is analytic, with all coefficients $a_j>0$.
>
> Since both kernels satisfy Assumption 1, we have the following claims:
>
> - You are right. When $d$ is fixed dimensions, [Bietti2021] showed that the eigenvalues of ReLU-NTK satisfy $ \mu_k \asymp k^{-d} $ and $ N(d, k) \asymp k^d $ for large $k$, leading to $\lambda_j \asymp j^{-\beta}$ with $\beta = (d+1)/d$; whereas it is known that the eigenvalues of the Gaussian kernel defined on sphere decay exponentially (see, e.g, [Amnon2020]). Therefore, we may need different tools to deal with the rate of the excess risk of fixed-dimensional spectral algorithms with ReLU-NTK and the one with the Gaussian kernel.
>
> - In large dimensions, the spectra of both ReLU-NTK and the Gaussian kernel exhibit a strong block structure as follows: $ \mu_k = \Theta(d^{-k}) $ and $ N(d, k) = \Theta(d^{k}) $ for $ k \leq p+1 $ (see, e.g., Lemmas D.11 and D.13). Therefore, our results hold for both ReLU-NTK and the Gaussian kernel in large dimensions without any adaptation.
>
> ---
>
>
> **Author's response to Question 2:**
>
> We agree that a uniform input distribution on spheres is too restrictive. As noted in Remark 2.1, most studies analyzing spectral algorithms in large-dimensional settings concentrate on inner product kernels on spheres for two main reasons:
>
> - Firstly, harmonic analysis on the sphere is clearer and more concise. For example, the properties of spherical harmonic polynomials are simpler than those of orthogonal series on general domains. This clarity makes Mercer’s decomposition of the inner product more explicit, avoiding several abstract assumptions.
>
> - Secondly, there are few results available for Mercer’s decomposition of kernels on general domains, especially when considering the domain’s dimension. For example, no results determine the eigenvalues of Sobolev space in large dimensions.
>
> Thus, if we believe that large-dimensional KRR and KGF exhibit some new phenomena, it would be prudent to start with a more tractable setting, and then extend the results to general domains.
>
> Regarding your second question, thank you for bringing up the interesting work [Haas2024] and suggesting an extension of our analysis. We believe that the results in [Haas2024] can indeed be extended to large-dimensional settings, although we must carefully consider the constants that depend on $d$. For example, the density in Lemma D.7 of [Haas2024] is lower and upper bounded by two constants depending on $d$.
>
> We appreciate your insightful suggestion. We will add the paper [Haas2024] to the discussion section of our manuscript, and we will consider extending our results to general domains in future work.
>
> ---
>
>
> **Author's response to Question 3:**
> Thank you for pointing out the potential issue in Eq (62). We have revised our manuscript and updated Eq (62) to the following version :
> $$
> \frac{ N_1 (\lambda) M_{1, \varphi}^2 (\lambda)}{n^2} = o( M_{2, \varphi} (\lambda) + \frac{\sigma^2}{n} N_{2,\varphi} (\lambda) );
> $$
> Moreover, we followed your suggestion and added a rigorous definition of $o(1)$ in our manuscript: We say two (deterministic) quantities $U(d), V(d)$ satisfy $U(d) =  o(V(d))$ if and only if for any $\varepsilon > 0$, there exists a constant $D_{\varepsilon}$ that only depends on $\varepsilon$ and the absolute positive constants $\sigma, \kappa, s, \gamma, c_0, c_1, c_2, C_1, \cdots, C_8 > 0$, such that for any $d > D_{\varepsilon}$, we have $U(d)< \varepsilon V(d)$.
>
> We hope that the updated definitions clarify that Eq (62) is correct when $\sigma>0$ is an absolute constant defined in (1).
>
> ---
>
>
> **Author's response to Question 4:**
>
> Thank you for your careful reading of our proof and for pointing out the typo in Eq (42). We will correct it in the updated manuscript. Regarding your comment about Eq (44), we believe it is not redundant. In Eq (44), $\varphi_{\lambda}$ refers to the filter function of a specific spectral algorithm (not necessarily KRR), while $\varphi_{\lambda}^{\text{KRR}}$ in (43) specifically denotes the filter function for KRR.
>
> ### Reference:
> - Alberto Bietti and Francis Bach. "Deep equals shallow for ReLU networks in kernel regimes." In International Conference on Learning Representations (ICLR), 2021.
> - Tilmann Gneiting. "Strictly and non-strictly positive definite functions on spheres." Bernoulli, 19(4): 1327–1349, 2013.
> - Geifman, Amnon, et al. "On the similarity between the Laplace and neural tangent kernels." Advances in Neural Information Processing Systems 33 (2020): 1451-1461.

---

> > ### Comment · Reviewer_XV4b · 2024-08-09
> >
> > Thank you very much for your detailed response. I am content to see that the results are valid for various kernels in high-dimensional setting. I would lean to accept this paper.

---

> > > ### Author Response · Authors · 2024-08-10
> > >
> > > Thank you very much for your positive feedback. We are pleased that you find the results valid in high dimensions. Your support and recommendation to accept the paper are greatly appreciated.

---

### Official Review · Reviewer_JAP1 · 2024-07-09

**Soundness:** 3
**Presentation:** 3
**Contribution:** 2
**Rating:** 6
**Confidence:** 4

**Summary:**

In a large-dimension setting, i.e., the dimension $d$ of the input grows polynomially with respect to the sample size $n$, this manuscript rigorously proves upper and lower bounds for spectral algorithms and shows the dependence on the qualification and the interpolation index. Consequently, the manuscript proves the saturation effect in spectral algorithms for large-dimensional data.

**Strengths:**

1. Identify several phenomena in large-dimensional spectral algorithms based on the derived rates. These phenomena are also illustrated by figures, making the explanations easy to follow.

2. Discovered the thresholding for igniting saturation effect is different for large-dimensional and fixed-dimensional settings. Specifically, in a large-dimensional setting, the saturation effect occurs when the interpolation index exceeds the qualification, whereas in a fixed-dimensional setting, it must be more than twice the qualification.

**Weaknesses:**

1. By checking previous work [1,2] and the proofs in these works, it looks like Theorem 3.1 has been established in Section 4 of [1], while Theorem 4.1 and Theorem 4.2 are the direct extensions of partial results in Theorem 2 and Theorem 3 from [2]. For instance, the proofs of Theorems 4.1 and 4.2 are obtained by replacing the Tikhonov regularized filter function in the variance and bias decomposition of [2] with a general filter function satisfying specific conditions such as C1 and C2. Such a proof trick has been used in previous extend from KRR (Tikhonov regularization) to general spectral algorithms, i.e., [3] to [4].

   However, unlike the extension from [3] to [4], the current manuscript seems to be a partial extension of [2] with a similar proof trick as I mentioned before. Therefore, I have concerns about the technical contribution and novelty of this manuscript as a submission to a conference. This work seems more like an extension to a journal like JMLR, etc.

   I am just not sure whether such a partial extension of a previous article with almost the same proof technique is suitable for conference publication or whether it would be better evaluated in a journal. I defer this justification to the AC. Please disregard this comment if the AC deems the current context appropriate for conference publication.

2. I noticed there are some simulation experiments to confirm the saturation effect in fixed dimension KRR; see [5]. Is it possible to confirm the results in this manuscript? I understand given the rates are asymptotic, it might be hard to have thorough investigations due to the extremely large $d$. But I'm still curious if any preliminary experiments can be done.


[1] Lu, Weihao, et al. "Optimal rate of kernel regression in large dimensions." _arXiv preprint arXiv:2309.04268_ (2023).

[2] Zhang, Haobo, et al. "Optimal Rates of Kernel Ridge Regression under Source Condition in Large Dimensions." _arXiv preprint arXiv:2401.01270_ (2024).

[3] Zhang, Haobo, et al. "On the optimality of misspecified kernel ridge regression." _International Conference on Machine Learning_. PMLR, 2023.

[4] Zhang, Haobo, Yicheng Li, and Qian Lin. "On the optimality of misspecified spectral algorithms." _Journal of Machine Learning Research_ 25.188 (2024): 1-50.

[5] Li, Yicheng, Haobo Zhang, and Qian Lin. "On the Saturation Effect of Kernel Ridge Regression." International Conference on Learning Representations. (2024)

**Questions:**

1. I'm curious to know if it is possible to conduct a similar analysis under ultra-high-dimension settings like the dimension grows exponentially fast as the sample size $d = \exp\{n^{\gamma}\}$. Do we need additional techniques to conduct these analyses?
2. Based on the figure, it looks like even when $s> 2\tau$, as long as $d$ grows with $n$, the saturation effect will not happen, which is different from the fixed dimension setting. While this may be the consequence of the derived rate, can authors provide some intuition behind this?
3. Is there a particular reason that the authors concern $\gamma \in p(s+1),(p+1)(s+1))$ with $p$ as integer to derive the rates? Why is this ratio an integer?

**Limitations:**

Yes.

---

> ### Author Rebuttal · Authors · 2024-08-06
>
> Thank you for your thorough review and valuable comments. Below, we address your concerns and questions in detail.
>
> Concern 1: Please let us clarify our novelties and contributions below.
>
> - Although the saturation effect has been observed for kernel ridge regression in a large-dimensional setting [2], its persistence for other spectral algorithms was unclear. We determine the exact rate of excess risk for analytic spectral algorithms with qualification $\tau$. Key outcomes include:
>   - Kernel gradient flow is minimax optimal in large dimensions, with an excess risk rate of $d^{-\min(\gamma-p, s(p+1))}$.
>   - We proved the saturation effect for large-dimensional KRR when $s>1$ and for spectral algorithms with $\tau$ when $s>\tau$.
>
> Regarding your concerns on the technical contributions:
> - [3][4] addressed the convergence rate for fixed-dimensional spectral algorithms when $0<s\leq 2\tau$, focusing on non-saturated cases where KRR and kernel gradient flow are rate-optimal. In contrast, they did not consider the saturation effect of spectral algorithms when $s > 2\tau$.
> - [1] determined the minimax rate of kernel regression for $f \in [\mathcal{H}] ^ s$ where $s=1$. The proof in [1] relies heavily on empirical process theory, requiring bounds on the empirical loss and its difference from the expected loss (excess risk). This approach does not generalize to cases where $s \neq 1$.
> - [2] provided a minimax lower bound for $s>0$ using integral operator techniques for KRR, hinting at a saturation effect but not generalizing to other spectral algorithms.
>
> We extended results to general spectral algorithms in large dimensions using complex variable analysis (Appendix E.1 and [6]) to match upper and lower bounds. Large dimensions introduce unique challenges:
>   - The polynomial eigendecay assumption that $\lambda_j \asymp j^{-\beta}$ does not hold in large dimensions, since the hidden constant factors in the assumption vary with $d$.
>   - The embedding index assumption in [6] does not hold either. We develop a new condition in (63) and (74) to replace the embedding index assumption, and we verify this new condition in Appendix D.4.
>
> We believe our contributions are novel and valuable for the machine learning community, meriting publication at NeurIPS.
>
> **Reference: [6]** Li, Yicheng, et al. "Generalization Error Curves for Analytic Spectral Algorithms under Power-law Decay."
>
> ---
>
> Concern 2: Thank you for your suggestion. We agree that conducting thorough numerical investigations can be challenging due to the extremely large dimensionality $d$. Following your recommendation, we conducted two preliminary experiments. We will follow your advice to conduct more comprehensive experiments later and report the results in the updated manuscript.
>
> ---
>
> Question 1: In this paper, we consider the asymptotic framework $n \asymp d^\gamma$ with $\gamma>0$. As discussed in lines 97-115, many studies focus on the performance of spectral algorithms within this asymptotic framework. By comparing our work with these studies, we highlight the novelty and contributions of our results to the field. In contrast, we were not aware of similar studies under ultra-high-dimensional settings, hence we did not explore this scenario in our manuscript.
>
> Your insightful suggestion implies that this could be an interesting avenue for future research. When $n \asymp d^{\gamma}$ and $\gamma$ is sufficiently small, the excess risk in our results is of rate $d^{-\gamma}$. If we consider $\log(d)$ as a limiting case of $d^{\gamma}$, we might conjecture that the correct rate of the excess risk when $n \asymp \log(d)$ is $\log^{-1}(d)$.
>
> We believe your suggestion is promising, and we will consider conducting a similar analysis under ultra-high-dimension settings in future work.
>
> ---
>
> Question 2: Thank you for your question. We guess that there might be some typos in your expression, and we guess that you wanted to state the distinction between our results and existing results in fixed-dimension settings as follows:
> - Our results, Theorems 4.1 and 4.2, demonstrate that the saturation effect occurs in large-dimensional settings if and only if $s > \tau$. In Figure 2 (on page 13), we illustrate this with the spectral algorithm rate (blue line) and the minimax rate (orange line). The blue and orange lines exhibit non-overlapping regions if and only if $s > \tau$. Therefore, Figure 2 aligns with our claims.
> - In contrast, in fixed dimension settings, the saturation effect occurs if and only if $s > 2\tau$.
>
> Then let us provide you with some intuition behind the consequence of the derived rate. We highlight that the periodic behavior of the rates with respect to $ \gamma $ in Theorem 4.1 and 4.2 is closely related to the spectral properties of inner product kernels for uniformly distributed data on a large-dimensional sphere. In Lemmas D.11 and D.13, we show that $\mu_k= \Theta(d^{-k})$ and $N(d, k) = \Theta(d^{k})$ for $k \leq p+1$. Moreover, recall that the leading terms for the bias and variance are given by $M_{2, \varphi}(\lambda) + \frac{\sigma^2}{n} N_{2,\varphi}(\lambda)$ (see Appendices D.1-D.3). Therefore, by comparing between $M_{2, \varphi}(\lambda)$ and $\frac{\sigma^2}{n} N_{2,\varphi}(\lambda)$ (with calculations detailed in Lemma D.14) under the strong block structure of the spectrum, we found that the rate of excess risk behaves periodically for each period $\gamma \in [p(s+1),(p+1)(s+1))$, $p = 0, 1, \cdots$, and that the saturation effect occurs when $s > \tau$.
>
> ---
>
> Question 3: The intervals $[p(s+1),(p+1)(s+1))$, $p=0, 1, \cdots$ naturally from our derivation process. Thus, we may focus on $\gamma\in [p(s+1),(p+1)(s+1))$. Regarding your second question about why this ratio is an integer, we are not entirely certain whether you are referring to the constant $\gamma$ or the integer $p=\lfloor\frac{\gamma}{s+1} \rfloor$. For further clarification, please let us know.

---

> > ### Comment · Reviewer_JAP1 · 2024-08-13
> >
> > I appreciate the detailed response and explanation that confirms the technical contribution of this paper. Also, thanks for the additional experiments that enhanced the content of the manuscript.
> >
> > I have adjusted my rating, good luck!

---

> > > ### Author Response · Authors · 2024-08-13
> > >
> > > We are pleased that you recognize the technical contributions of our paper and the additional experiments we conducted. We will follow your suggestion to include these experiments in the updated manuscript. Thank you for raising your score.

---

### Official Review · Reviewer_QpMc · 2024-07-24

**Soundness:** 4
**Presentation:** 3
**Contribution:** 4
**Rating:** 7
**Confidence:** 4

**Summary:**

### Summary:



The authors study  the saturation of spectral algorithms (KRR & GF) in high-dimensions where $n,d$ are both large,  meaning that when KRR can't achieve information theoretic lower bounds with over smooth the regression functions while kernel Gradient Flow (GF) can.
Theorem 3.1 states the optimal convergence rate of kernel GF which matches the provided minimax lower bound in Theorem 3.3. Moreover, they find that KRR is unableto achieve this lower bound (being suboptimal) for interpolation spaces with $s >1$.

**Strengths:**

### Pros:

- very well-written
- tightening previous results on the minimax rate of kernel GF in high dimensions
- proving the saturation of KRR in high dimensions

**Weaknesses:**

### Cons:

 - the main results of this paper are stated on page 7, the presentation of the results is  slow

**Questions:**

This is an interesting paper about the saturation of KRR in high dimensions. The authors provide several new results and the paper is well written and organized.

- line 200 -- what does $f_\star$ mean? It is only defined in the next page.

---

> ### Author Rebuttal · Authors · 2024-08-04
>
> We sincerely thank you for your detailed review and thoughtful comments. We are grateful that you found our paper well-written and technically solid. Below, we address your concerns and questions in detail.
>
>
> **Author's response to Concern 1:**
> We appreciate your suggestion to present our results earlier in the paper. We agree that, in a 9-page conference paper, it is crucial to present the main results as early as possible. Therefore, we followed your recommendation and added a non-rigorous version of our main results (Theorem 4.1 and Theorem 4.2) in the contribution part (page 2 in our manuscript). For your convenience, we restate the non-rigorous version as follows:
>
> ### Theorem (Restate Theorem 4.1 and 4.2, non-rigorous)
> Let $s>0$, $\tau \geq 1$, and $\gamma>0$ be fixed real numbers. Denote $p$ as the integer satisfying $\gamma \in [p(s+1), (p+1)(s+1))$. Then under certain conditions, the excess risk of large-dimensional spectral algorithm with qualification $\tau$ is of order:
> $$
> \Theta_{\mathbb{P}} ( d^{-\min ( \gamma-p, s(p+1) )} ) , \quad s \leq \tau
> $$
> $$
> \Theta_{\mathbb{P}} ( d^{-\min ( \gamma-p, \frac{\tau(\gamma-p+1)+p\tilde{s}}{\tau+1}, \tilde{s}(p+1) )} ), \quad s > \tau,
> $$
> where $\tilde{s} = \min ${$s, 2\tau $}.
>
> We would greatly appreciate your further advice on how to better organize our paper.
>
>
> **Author's response to Question 1:**
> Thank you for highlighting this point. We will clarify the definition of $f_{\star}(x) = x[1]x[2]\cdots x[L]$ in line 200. Here, $f_{\star}$ represents the regression function, where $x[i]$ denotes the $i$-th component of $x$. We will ensure this definition is included in the revised manuscript.
>
> ---
>
> We thank you once again for your expertise and valuable feedback. Please let us know if you have any further questions.

---

> > ### Comment · Reviewer_QpMc · 2024-08-10
> >
> > Thank you! I appreciate the authors' response. It is really helpful to have such non-rigorous versions of the results earlier in the paper, and I'm happy that the authors included this to improve the presentation of their draft. I continue supporting this paper so I keep my score positive.

---

> > > ### Author Response · Authors · 2024-08-10
> > >
> > > Thank you as well! We are pleased that the changes you suggested have enhanced the clarity and presentation of our work. We sincerely appreciate your positive assessment of our paper.

---

### Author Rebuttal · Authors · 2024-08-06

Following the recommendation of the Reviewer JAP1, we conducted two preliminary experiments using two specific kernels: the RBF kernel and the NTK kernel. Experiment 1 was designed to confirm the optimal rate of kernel gradient flow and KRR when $s=1$. Experiment 2 was designed to illustrate the saturation effect of KRR when $s>1$.

**Experiment 1:** We consider the following two inner product kernels:

1. **RBF kernel with a fixed bandwidth:**

   $$
   K^{\mathrm{rbf}}(x,x^{\prime}) = \exp{\left(-\frac{\|x-x^{\prime}\|_{2}^{2}}{2}\right)}, ~~x, x^{\prime} \in \mathbb{S}^{d}.
   $$

2. **Neural Tangent Kernel (NTK) of a two-layer ReLU neural network:**

   $$
   K^{\mathrm{ntk}}(x, x^\prime) := \Phi(\langle x, x^{\prime} \rangle), ~~x, x^{\prime} \in \mathbb{S}^{d},
   $$

   where $\Phi(t)=\left[\sin{(\arccos t)}+2(\pi-\arccos t)t\right]/ (2 \pi)$.

The RBF kernel satisfies Assumption 1. For the NTK, the coefficients of $\Phi(\cdot)$, $(a_{j}) _ {j=0} ^ {\infty}$, satisfy $a_{j} > 0, j \in \{0, 1\} \cup \{2,4,6,\ldots\}$ and $a_{j} = 0, j \in \{3,5,7,\ldots\}$ (see, e.g., [Lu2023]). As noted after Assumption 1, our results can be extended to inner product kernels with certain zero coefficients $a_j$. Specifically, for any $\gamma>0$, as long as $a_{j} > 0$ for $j = \lfloor \gamma \rfloor, \lfloor \gamma \rfloor+1$, the proof and convergence rate remain the same. Therefore, for $\gamma<2$ in our experiments, the convergence rates for NTK will be the same as for the RBF kernel.

We used the following data generation procedure:

$$
y_{i} = f_{*}(x_{i}) + \epsilon_{i}, ~~ i = 1, \ldots, n,
$$

where each $x_{i}$ is i.i.d. sampled from the uniform distribution on $\mathbb{S} ^ {d}$, and $\epsilon_{i} \overset{\text{i.i.d.}}{\sim} \mathcal{N}(0,1)$.

We selected the training sample sizes $n$ with corresponding dimensions $d$ such that $n = d^{\gamma}, \gamma = 0.5, 1.0, 1.5, 1.8$. For each kernel and dimension $d$, we consider the following regression function $f_{*}$:

$$
f_{*}(x) = K(u_{1},x) + K(u_{2},x) + K(u_{3},x), \quad \text{for some}\quad u_{1}, u_{2}, u_{3} \in \mathbb{S}^{d}.
$$

This function is in the RKHS $\mathcal{H}$, and it is easy to prove that, for any $u \in \mathbb{S} ^ {d}$, Assumption 2 (b) in our revision holds for $K(u, \cdot)$ with $s=1$. Therefore, Assumption 2 holds for $s=1$. We used logarithmic least squares to fit the excess risk with respect to the sample size, resulting in the convergence rate $r$. The experimental results align well with our theoretical findings.

**Experiment 2:** We use most of the settings from Experiment 1, except that the regression function is changed to $f_{*}(x) = \sqrt{\mu_2^{s}N(d, 2)} P_2(\langle \xi, x \rangle)$ with $s=1.9$, $P_2(t) := (d t^2-1)/(d-1)$ the Gegenbauer polynomial, and $\xi \in \mathbb{S}^{d}$. Notice that the addition formula $P_2(\langle \xi, x \rangle) = \frac{1}{N(d, 2)}\sum_{j=1}^{N(d, 2)}Y_{2, j}(\xi)Y_{2, j}(x)$ implies that

$$
||f_{*}|| _ {[\mathcal{H}] ^ {s}}^2 = \frac{1}{N(d, 2)} \sum _ {j=1} ^ {N(d, 2)} Y_{2, j} ^ 2 (\xi) = P _ 2(1) = 1,
$$

hence $f_{*} \in [\mathcal{H}]^{s}$ and satisfies Assumption 2.

Our experiment settings are similar to those on page 30 of [5]. We choose the regularization parameter for KRR and kernel gradient flow as $\lambda=0.05 \cdot d^{-\theta}$. For KRR, since Corollary D.16 suggests that the optimal regularization parameter is $\lambda \asymp d^{-0.7}$, we set $\theta=0.7$. Similarly, based on Corollary D.16, we set $\theta=0.5$ for kernel gradient flow. Additionally, we set $\gamma = 1.8$. The results indicate that the best convergence rate of KRR is slower than that of kernel gradient flow, implying that KRR is inferior to kernel gradient flow when the regression function is sufficiently smooth.

### References

- Lu, Weihao, et al. "Optimal rate of kernel regression in large dimensions." arXiv preprint arXiv:2309.04268 (2023).

- Li, Yicheng, Haobo Zhang, and Qian Lin. "On the Saturation Effect of Kernel Ridge Regression." International Conference on Learning Representations. (2024)

---

### Decision · Program_Chairs · 2024-09-25

**Decision:**

Accept (poster)

**Comment:**

A very solid contribution on the saturation effect of spectral learning algorithms in high dimension (dimension grows as a power of the sample size), a setting that has generated a lot of attention and investigation in recent years, while the saturation effect had only been studied in fixed dimension. The concern of possibly only technically incremental results with respect to previous work raised by one reviewer has been answered convincingly by the authors in the discussion, pointing out novelties in results and techniques used. All reviews lauded the paper for its quality of writing and presentation.